# BRIDGING INPUT FEATURE SPACES TOWARDS GRAPH FOUNDATION MODELS

**Moshe Eliasof**
University of Cambridge
Ben-Gurion University of the Negev
me532@cam.ac.uk

**Krishna Sri Ipsit Mantri**
Purdue University
mantrik@purdue.edu

**Beatrice Bevilacqua**
Purdue University
bbevilac@purdue.edu

**Bruno Ribeiro**
Purdue University
ribeirob@purdue.edu

**Carola-Bibiane Schönlieb**
University of Cambridge
cbs31@cam.ac.uk

## ABSTRACT

Unlike vision and language domains, graph learning lacks a shared input space, as input features differ across graph datasets not only in semantics, but also in value ranges and dimensionality. This misalignment prevents graph models from generalizing across datasets, limiting their use as foundation models. In this work, we propose ALL-IN, a simple and theoretically grounded method that enables transferability across datasets with different input features. Our approach projects node features into a shared random space and constructs representations via covariance-based statistics, thus eliminating dependence on the original feature space. We show that the computed node-covariance operators and the resulting node representations are invariant in distribution to permutations of the input features. We further demonstrate that the expected operator exhibits invariance to general orthogonal transformations of the input features. Empirically, ALL-IN achieves strong performance across diverse node- and graph-level tasks on unseen datasets with new input features, without requiring architecture changes or retraining. These results point to a promising direction for input-agnostic, transferable graph models.

## 1 INTRODUCTION

Foundation models have shown remarkable success in domains such as language and vision, where large-scale pretraining enables strong performance across a wide range of downstream tasks. A similar goal has emerged for graph learning: to develop graph foundation models that generalize across tasks, domains, and datasets (Mao et al., 2024). However, a key obstacle in this direction is the lack of transferability across graphs, as knowledge learned from one graph is often difficult to apply to another due to fundamental differences in their structure and, critically, their input features.

Unlike vision or language data, graph datasets typically do not share a common input space. Node features often differ significantly not only in distribution and semantics but also in dimensionality from one graph to another. Furthermore, graphs themselves may vary in size, sparsity, and topological patterns. These mismatches break many of the assumptions that underlie successful generalization in other domains, making it difficult to define a common representation space or pretraining strategy.

Existing approaches to graph foundation models fall into two broad categories. The first integrates LLMs by serializing graph data into text or designing prompt-based mechanisms (Liu et al., 2024; Zhao et al., 2023; Chen et al., 2024b; Fatemi et al., 2024; Perozzi et al., 2024; Chen et al., 2024a; Zhao et al., 2023; He & Hooi, 2024; Huang et al., 2023; Tang et al., 2024; Kim et al., 2024; Zhao et al., 2024a; Gong et al., 2024; Sun et al., 2022; 2023), leveraging LLM capabilities but often discarding fine-grained graph properties. The second direction aims to explicitly align or adapt feature spaces across datasets using techniques like input projections (Xia & Huang, 2024; Yu et al., 2024; Zhao et al., 2024a), specialized encoders (Lachi et al., 2024), structuralization (Frasca et al., 2024), or order statistics (Shen et al., 2025). However, these methods often remain specialized to particular settings or tasks, or may require careful adaptation to new scenarios.

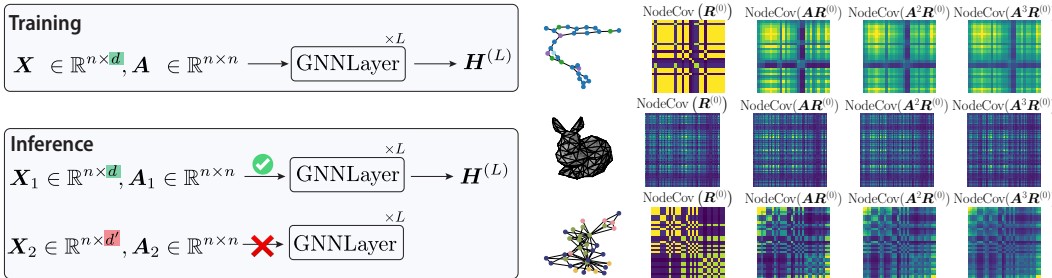

(a) Motivation: The challenge of varying input features     (b) ALL-IN consistent node-covariance operators

Figure 1: **Addressing feature heterogeneity with ALL-IN's node-covariance operators.** (a) When a GNN is trained on graph data with node features $X$ of dimension $d$, it cannot be directly applied on graphs with features of a different dimensionality $d'$. (b) ALL-IN computes $n \times n$ node-covariance operators, capturing node similarities, providing a common space that is independent of the original, heterogeneous, feature spaces. Different node colors indicate distinct node features.

In this work, we propose a novel approach, grounded in statistical principles, to overcome input feature heterogeneity (Figure 1). Our method first projects potentially disparate node features into a common, high-dimensional space using a stochastic projection matrix. We then leverage second-order statistics within this space using covariance operators. Specifically, we model feature dimensions as independent and identically distributed samples from an unknown distribution over the nodes, and compute the empirical node-covariance matrix based on these projected representations. This matrix captures pairwise node similarities based on how their projected features co-vary, providing a representation inherently robust to changes in feature semantics, value, and dimensionality.

We introduce ALL-IN (All Input spaces), a graph learning framework built upon this principle. Instead of directly processing raw node features in downstream layers, ALL-IN utilizes the computed stochastic node-covariance matrix (and its higher-order variants), as shown in Figure 1, as a graph operator within a graph neural network (GNN). This node-covariance matrix captures interactions between nodes, specifically, how similar two nodes are in terms of their feature activations across the feature dimensions. Our theoretical analysis reveals significant robustness properties: (a) The computed operators and, critically, the resulting node representations throughout the GNN are invariant in distribution to arbitrary permutations of the original input features; (b) The expected operator is invariant to general orthogonal transformations (basis changes) of the input features; (c) The overall method is inherently insensitive to dimensional mismatches across datasets. We further identify qualitative conditions under which covariance-based representations retain task-relevant information and enable transfer across datasets with different input features.

Our empirical results confirm the efficacy of this approach: ALL-IN achieves strong transfer performance to new datasets with new input features across diverse node- and graph-level tasks. As a result, ALL-IN offers a promising approach toward the development of graph foundation models.

## 2    RELATED WORK

**Graph Foundation Models (GFMs).** GFMs aim to learn representations that generalize across datasets and tasks, but achieving robust generalization remains challenging, especially when node features change. Some approaches integrate LLMs by converting graphs to text or embedding features through prompt-based designs (Liu et al., 2024; Zhao et al., 2023; Chen et al., 2024b; Fatemi et al., 2024; Perozzi et al., 2024; Chen et al., 2024a; Zhao et al., 2023; He & Hooi, 2024; Huang et al., 2023; Tang et al., 2024; Kim et al., 2024; Zhao et al., 2024a; Gong et al., 2024; Sun et al., 2022; 2023), or by generating or augmenting graphs with LLM guidance before training a graph encoder (Xia et al., 2024), but this can lead to loss of structural details. Text-attributed GFMs further learn transferable vocabularies or automatically search architectures on such LLM-derived features (Wang et al., 2024; Chen et al., 2025a), which improves transfer within TAGs but does not directly handle non-textual node attributes. Other works align feature spaces through projections (Xia & Huang, 2024; Yu et al., 2024; Zhao et al., 2024a; Fang et al., 2023) and multi-domain feature or structure aligners with

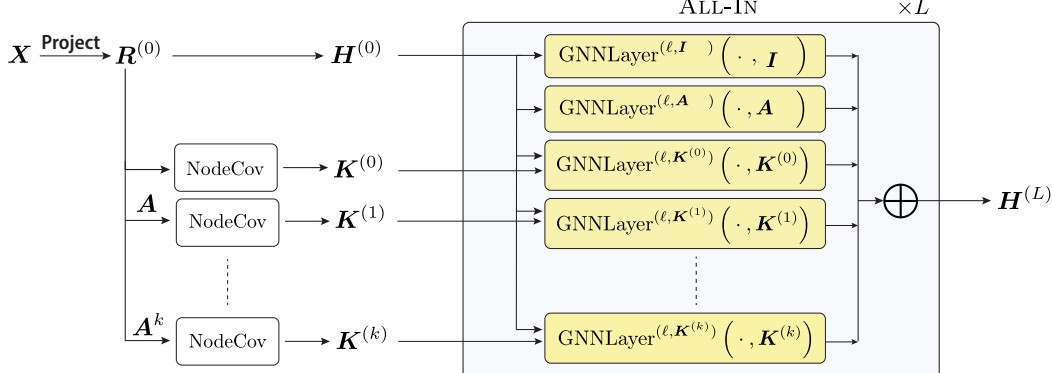

Figure 2: **The ALL-IN Architecture.** Input node features $X$ are first randomly projected into $R^{(0)}$. This $R^{(0)}$ serves as initial node representations $H^{(0)}$. Concurrently, $R^{(0)}$ and its propagated versions (e.g., $R^{(p)} = A^p R^{(0)}$) are used to compute a set of node-covariance matrices $\{K^{(p)}\}_{p=0}^k$ capturing diverse orders of feature-based node similarities. These matrices are used as operators within different GNN (sub-)layers, whose outputs are concatenated to form the updated node representation.

prompts or mixtures-of-experts (Yu et al., 2025; Yuan et al., 2025), perceiver-based encoders (Lachi et al., 2024), computing analytical solutions (in the case of node classification) (Zhao et al., 2024b), encoding features into the graph structure (Frasca et al., 2024; Galkin et al., 2024; Wang & Luo, 2024; Franks et al., 2025) or learning shared structural vocabularies in Riemannian spaces (Sun et al., 2025), or encoding feature relationships (Shen et al., 2025). While these methods advance GFM capabilities, they often require task-specific adaptations, leaving a gap for truly input-space-agnostic solutions. ALL-IN offers a distinct path: it creates transferable representations by processing arbitrary input features through stochastic projections and node-covariance operators, enabling frozen-encoder transfer without task- or domain-specific prompts or architectural changes.

**Structural and Positional Encodings.** Efforts to create universal graph representations include transferable structural and positional encodings (SPEs) (Rampášek et al., 2022; Cantürk et al., 2024; Chen et al., 2025b; Kim et al., 2024). SPEs aim to capture graph topology in a feature-agnostic manner, often within Graph Transformers or GNNs. While such SPEs can complement node features, ALL-IN directly addresses the challenge of heterogeneous node features themselves, transforming them into a robust, transferable format using their covariance structure, irrespective of any additional SPEs.

**Covariance networks.** Covariance matrices have also informed the design of neural networks. For instance, coVariance Neural Networks (VNNs) (Sihag et al., 2022) process $d \times d$ sample covariance matrices, with $d$ the input feature dimension, which describe feature inter-correlations, offering benefits like stability to varying sample sizes and inspiring extensions for fairness (Cavallo et al., 2025) and sparsity (Cavallo et al., 2024). Other related efforts focus on transferring principal components derived from data covariance (Hendy & Dar, 2024). While these methods analyze relationships between features using sample covariance matrices, ALL-IN constructs an $n \times n$ node-covariance matrix, with $n$ number of nodes. This operator quantifies similarities between pairs of nodes based on how their (randomly projected) features co-vary across dimensions. This distinct formulation is tailored to building transferable representations from graphs with heterogeneous node features, addressing a challenge different from that targeted by the aforementioned approaches.

## 3 METHOD

Our method, ALL-IN, replaces dataset-specific raw node features with covariance-based operators that are better suited for generalization across input feature spaces. The approach comprises three main stages: (1) Random Feature Projection to map input features to a shared space, (2) Node-Covariance Operator computation to capture robust node similarities, and (3) Operator-based Propagation to learn transferable node representations. An overview of ALL-IN can be found in Figure 2.

**Random Feature Projections.** Given a graph with $n$ nodes and node feature matrix $\boldsymbol{X} \in \mathbb{R}^{n \times d}$, where the input dimension $d$ may vary across graph datasets, we first apply a random linear transformation to project the features into a unified fixed-dimensional space $h$ that is shared across datasets:

$$\boldsymbol{R}^{(0)} = \boldsymbol{X}\boldsymbol{C}, \qquad \text{with } \text{vec}(\boldsymbol{C}) \sim \mathcal{N}(\boldsymbol{0}, \boldsymbol{I}_{dh}) \text{ sampled at each forward pass}, \qquad (1)$$

that is, $\boldsymbol{C} \in \mathbb{R}^{d \times h}$ is an isotropic Gaussian random weight matrix sampled independently at each forward pass. This step is key to ensuring that our approach is invariant (in distribution) to feature permutations, as we discuss in Section 4.

**Node-Covariance Operators.** We treat each column of $\boldsymbol{R}^{(0)}$ as an i.i.d. signal over the nodes and compute the node-covariance matrix to capture second-order relationships (node similarities) based on feature co-variation across the latent dimensions:

$$\boldsymbol{K}^{(0)} = \text{NodeCov}(\boldsymbol{R}^{(0)}) = \frac{1}{h}\boldsymbol{R}_c^{(0)}\boldsymbol{R}_c^{(0)^T} \in \mathbb{R}^{n \times n}, \qquad (2)$$

where $\boldsymbol{R}_c^{(0)} \in \mathbb{R}^{n \times h}$ is the centered projected feature matrix defined by $\boldsymbol{R}_c^{(0)} = \boldsymbol{R}^{(0)} - \boldsymbol{1}_n\bar{\boldsymbol{r}}$ with $\bar{\boldsymbol{r}} = \frac{1}{n}\sum_i^n \boldsymbol{R}_i^{(0)} \in \mathbb{R}^{1 \times h}$ the empirical mean of the projected node features, and $\boldsymbol{1}_n \in \mathbb{R}^{n \times 1}$ the all-ones vector. This centering operation is equivalent to pre-multiplying by the geometric centering matrix $\boldsymbol{\Pi}_c = \boldsymbol{I}_n - \frac{1}{n}\boldsymbol{1}_n\boldsymbol{1}_n^T$, i.e., $\boldsymbol{R}_c^{(0)} = \boldsymbol{\Pi}_c\boldsymbol{R}^{(0)}$. The resulting $\boldsymbol{K}^{(0)}$ is an $n \times n$ matrix reflecting node similarities in the projected feature space. An interesting property is that if we consider two nodes $u$ and $v$ with feature vectors $\boldsymbol{X}_u$ and $\boldsymbol{X}_v = -\boldsymbol{X}_u$, then their auto-covariance terms in $\boldsymbol{K}^{(p)}$ coincide, but their rows $\boldsymbol{K}_u^{(p)}$ and $\boldsymbol{K}_v^{(p)}$ differ in the cross-covariance entries with other nodes because their signs flip, so message passing based on $\boldsymbol{K}^{(p)}$ can still distinguish them.

To integrate structural information with feature similarities, we compute higher-order covariance matrices based on propagated features. Specifically, for each $p = 1, 2, \ldots, k$, we first perform message passing on the initial projected features $\boldsymbol{R}^{(0)}$ using the graph's adjacency matrix $\boldsymbol{A}$:

$$\boldsymbol{R}^{(p)} = \boldsymbol{A}^p\boldsymbol{R}^{(0)}.$$

Then, we compute the covariance matrix on these propagated, centered features $\boldsymbol{R}_c^{(p)} = \boldsymbol{\Pi}_c\boldsymbol{R}^{(p)}$:

$$\boldsymbol{K}^{(p)} = \text{NodeCov}(\boldsymbol{R}^{(p)}) = \frac{1}{h}\boldsymbol{R}_c^{(p)}\boldsymbol{R}_c^{(p)^T} \in \mathbb{R}^{n \times n}. \qquad (3)$$

The operator $\boldsymbol{K}^{(p)}$ captures node similarities based on features aggregated from neighborhoods up to $p$ hops away, thus encoding increasingly global structural context in the graph.

**Node Representations.** We collect a set of graph operators, which includes the identity matrix $\boldsymbol{I}$, the adjacency matrix $\boldsymbol{A}$, and the computed node-covariance matrices $\mathcal{K} = \{\boldsymbol{K}^{(p)}\}_{p=0}^k$:

$$\mathcal{O} = \{\boldsymbol{I}, \boldsymbol{A}, \boldsymbol{K}^{(0)}, \boldsymbol{K}^{(1)}, \ldots, \boldsymbol{K}^{(k)}\}. \qquad (4)$$

Instead of using the original node features, we rely on the random projections $\boldsymbol{R}^{(0)}$, potentially augmented with structural encodings, such as random-walk encodings (Dwivedi et al., 2022a). That is, we let $\boldsymbol{H}^{(0)}$ be

$$\boldsymbol{H}^{(0)} = \boldsymbol{R}^{(0)} \oplus \boldsymbol{S} \qquad (5)$$

where $\oplus$ indicates concatenation and $\boldsymbol{S} \in \mathbb{R}^{n \times h_s}$ is a structural encoding matrix. We note that, although the node-covariance operators $\boldsymbol{K}^{(p)}$ capture second-order statistics, ALL-IN maintains first-order information: the projected features $\boldsymbol{R}^{(0)}$ are used directly as part of the initial node representations $\boldsymbol{H}^{(0)}$ in Equation (5). For example, if two nodes $u$ and $v$ have feature vectors $\boldsymbol{X}_u$ and $\boldsymbol{X}_v = -\boldsymbol{X}_u$, then their projected features satisfy $\boldsymbol{R}_u^{(0)} = \boldsymbol{X}_u\boldsymbol{C}$ and $\boldsymbol{R}_v^{(0)} = -\boldsymbol{X}_u\boldsymbol{C}$, so they are distinguishable in $\boldsymbol{H}^{(0)}$.

At each layer $\ell = 1, \ldots, L$, we propagate the current node representations using every operator $\boldsymbol{O} \in \mathcal{O}$, and concatenate the outputs to obtain the updated representations:

$$\boldsymbol{H}^{(\ell)} = \bigoplus_{\boldsymbol{O} \in \mathcal{O}} \text{GNNLayer}^{(\ell, \boldsymbol{O})}(\boldsymbol{H}^{(\ell-1)}, \boldsymbol{O}), \qquad (6)$$

where $\text{GNNLayer}^{(\ell, \boldsymbol{O})}$ is the GNN layer associated with operator $\boldsymbol{O} \in \mathcal{O}$ in layer $\ell$, taking as input $\boldsymbol{H}^{(\ell-1)}$ and performing message passing using $\boldsymbol{O}$ as the operator, using learnable weights $\boldsymbol{W}^{(\ell, \boldsymbol{O})} \in \mathbb{R}^{h^{(\ell-1)} \times h^{(\ell)}}$ and $h^{(0)} = h + h_s$.

In the presence of edge features, which, similarly to node features, may vary across datasets, we employ an analogous strategy. Specifically, we first project the edge features into a fixed-dimensional space using an isotropic Gaussian random weight matrix, yielding edge representations that are independent of feature dimensionality. Then, we aggregate these projected edge features at the node level (e.g., by averaging features of incoming edges for each node) to obtain node-level representations $\boldsymbol{R}^{(0)}_{\text{edge}}$ derived from edges. We then compute node-covariance matrices $\boldsymbol{K}^{(p)}_{\text{edge}}$ based on these aggregated (and potentially propagated) features, similar to Equation (3). Finally, we add these edge-derived covariance operators $\mathcal{K}_{\text{edge}} = \{\boldsymbol{K}^{(0)}_{\text{edge}}, \ldots, \boldsymbol{K}^{(k)}_{\text{edge}}\}$ to the operator set $\mathcal{O}$ (Equation (4)). This allows the model (Equation (6)) to incorporate edge information while remaining compatible across datasets with differing edge feature spaces.

## 4 THEORETICAL INSIGHTS

This section establishes the theoretical foundations underpinning the ability of ALL-IN to handle heterogeneous input features and enable generalization across datasets. A core contribution is proving the method's robustness to variations in feature representation. We first demonstrate that the node-covariance operators and the resulting node representations are invariant *in distribution* to arbitrary permutations of the input features, providing robustness to feature re-ordering. We then show that the *expected node-covariance operator* is invariant to general orthogonal transformations, ensuring robustness to the choice of orthonormal basis (Section 4.1). Building on these properties, we validate the stochastic training procedure using Jensen's inequality under standard convexity assumptions (Section 4.2). Finally, we discuss conditions supporting transferability, analyzing scenarios where the operator remains stable across graphs with differing feature distributions and proving its consistency for large projection dimensions (Section 4.3). All proofs are provided in Appendix B.

### 4.1 INVARIANCE TO FEATURE SPACE TRANSFORMATIONS

A primary obstacle to cross-dataset transfer is the lack of feature standardization, leading to arbitrary differences in feature ordering and basis choice across datasets. Our approach, centered on node-covariance after random projection, inherently addresses these issues through invariance properties. First, the use of random isotropic Gaussian projections renders the process statistically insensitive to the order of input features. We formalize this by showing that the distribution of the projected feature matrix remains unchanged when the original features are permuted.

**Proposition 4.1** (Distributional Invariance of Projected Features to Feature Permutation). *Let $\boldsymbol{X} \in \mathbb{R}^{n \times d}$ be node features, $\boldsymbol{P} \in \mathbb{R}^{d \times d}$ be any permutation matrix, and $h$ be the projection dimension. Let $\boldsymbol{C} \in \mathbb{R}^{d \times h}$ be an isotropic Gaussian random matrix (i.e., $vec(\boldsymbol{C}) \sim \mathcal{N}(\boldsymbol{0}, \boldsymbol{I}_{dh})$). Define the projected features as $\boldsymbol{R}^{(0)} = \boldsymbol{XC}$ and the features projected after permutation as $\bar{\boldsymbol{R}}^{(0)} = (\boldsymbol{XP})\boldsymbol{C}$. Then $\boldsymbol{R}^{(0)}$ and $\bar{\boldsymbol{R}}^{(0)}$ are equal in distribution: $\boldsymbol{R}^{(0)} \stackrel{d}{=} \bar{\boldsymbol{R}}^{(0)}$.*

In essence, Proposition 4.1 establishes that random projections effectively "mix" features, rendering their original ordering statistically irrelevant after projection. More importantly, the permutation invariance is characterized *in distribution*, rather than pointwise: for a fixed random projection $\boldsymbol{C}$, the features in $\boldsymbol{R}^{(0)}$ retain sensitivity to input permutations, thereby enabling a neural network to better capture the relationships between node features and topology.

To illustrate this concept, consider three nodes $u, v, w \in V$ with features $\boldsymbol{X}_u = (0, 1)$, $\boldsymbol{X}_v = (0, 1)$, and $\boldsymbol{X}_w = (1, 0)$. Under strict (pointwise) permutation invariance, the embeddings of all nodes would be equivalent, obscuring the key distinction that $u$ and $v$ share identical features, whereas $w$ has a different feature. In contrast, distributional invariance ensures that the distributions of $\boldsymbol{R}^{(0)}_u$, $\boldsymbol{R}^{(0)}_v$, and $\boldsymbol{R}^{(0)}_w$ are identical, yet individual forward passes yield different outcomes: given $\boldsymbol{C}$, we have $\boldsymbol{R}^{(0)}_u = \boldsymbol{R}^{(0)}_v \neq \boldsymbol{R}^{(0)}_w$. This property preserves the model's ability to distinguish between nodes $u$ and $v$ (which share the same features) and node $w$ (which has a different feature), while maintaining

symmetry in the model's statistical behavior, thus striking a balance between permutation invariance and expressive power.

Next, we show that the NodeCov operators applied to the sequence $\{\boldsymbol{R}^{(p)}\}_{p=0}^k$ (as defined in Equation (3)) yield features that are also distributionally invariant.

**Corollary 4.2** (Distributional Invariance of Node Covariance Operators to Feature Permutation)**.** *Let $\boldsymbol{X} \in \mathbb{R}^{n \times d}$ be node features, and $\boldsymbol{P} \in \mathbb{R}^{d \times d}$ be any permutation matrix. Let $\boldsymbol{R}^{(0)} = \boldsymbol{X}\boldsymbol{C}$ be the initial projected features. Let $\mathcal{K} = \{\boldsymbol{K}^{(p)}\}_{p=0}^k$ be the set of node-covariance operators, where $\boldsymbol{K}^{(p)} = NodeCov(\boldsymbol{A}^p \boldsymbol{R}^{(0)})$ is computed using the deterministic function NodeCov (Equation (3)), and $\boldsymbol{A}$ is the adjacency matrix. It follows directly from the distributional invariance of $\boldsymbol{R}^{(0)}$ that the entire set of operators $\mathcal{K}$ is also invariant in distribution to permutations of the input features $\boldsymbol{X}$. That is, if $\bar{\mathcal{K}}$ is the set of operators computed using $\boldsymbol{X}\boldsymbol{P}$ instead of $\boldsymbol{X}$, then $\mathcal{K} \stackrel{d}{=} \bar{\mathcal{K}}$.*

The significance of Proposition 4.1 and Corollary 4.2 is substantial: it guarantees that the complete statistical behavior of $\boldsymbol{R}^{(0)}$ and the operators $\boldsymbol{K}^{(p)}$ central to ALL-IN is fundamentally robust to arbitrary input feature ordering, directly addressing a key source of heterogeneity across graph datasets. This distributional invariance also extends to the hidden representations $\boldsymbol{H}^{(\ell)}$, for all $\ell = 1 \ldots L$ derived from these operators, as shown in Theorem B.1 in Appendix B.

The stochastic projection matrix $\boldsymbol{C}$ plays a critical role beyond enabling the distributionally invariance properties discussed earlier; its use is intrinsically linked to the expressive capability of the learning framework. Training with node-covariance operators NodeCov$(\boldsymbol{R}^{(0)})$ derived from these stochastic projections offers advantages over relying on a single, deterministically computed covariance operator, such as NodeCov$(\boldsymbol{X})$. While NodeCov$(\boldsymbol{X})$ provides a stable, pointwise feature-permutation invariant view of node similarities, it can obscure subtle but important distinctions between nodes. In contrast, individual stochastic realizations NodeCov$(\boldsymbol{R}^{(0)}) = $ NodeCov$(\boldsymbol{X}\boldsymbol{C})$ (for a specific $\boldsymbol{C}$) can preserve these finer-grained distinctions, providing richer and more varied signals to the GNN. Theorem 4.3 formalizes this concept by demonstrating that there exist instances where the stochastic operator NodeCov$(\boldsymbol{X}\boldsymbol{C})$ can distinguish nodes that the deterministic operator NodeCov$(\boldsymbol{X})$ cannot.

**Theorem 4.3** (Distinguishability through $\boldsymbol{C}$)**.** *There exist node features $\boldsymbol{X} \in \mathbb{R}^{n \times d}$, nodes $u, v \in V$ with $\boldsymbol{X}_u \neq \boldsymbol{X}_v$ such that NodeCov$(\boldsymbol{X})$ makes $u$, $v$ indistinguishable (automorphic), but NodeCov$(\boldsymbol{X}\boldsymbol{C})$ (for a.s. all $\boldsymbol{C}$) makes $u$, $v$ distinguishable (not automorphic).*

Finally, while distributional invariance covers permutations, analyzing the expected operator reveals broader robustness to basis changes and identifies the structure captured on average, as we show next.

**Theorem 4.4** (Expected Invariance to Orthogonal Transformations)**.** *Let $\boldsymbol{X} \in \mathbb{R}^{n \times d}$ be node features, $\boldsymbol{Q} \in \mathbb{R}^{d \times d}$ be an orthogonal matrix, and $h$ be the projection dimension. Consider a random projection matrix $\boldsymbol{C} \in \mathbb{R}^{d \times h}$ with $vec(\boldsymbol{C}) \sim \mathcal{N}(\boldsymbol{0}, \boldsymbol{I}_{dh})$. Let NodeCov$(\boldsymbol{R}^{(0)}) = \frac{1}{h}(\boldsymbol{\Pi}_c \boldsymbol{R}^{(0)})(\boldsymbol{\Pi}_c \boldsymbol{R}^{(0)})^T$ be the Node Covariance operator (Equation (2)), where $\boldsymbol{\Pi}_c = \boldsymbol{I}_n - \frac{1}{n}\boldsymbol{1}_n \boldsymbol{1}_n^T$ is the centering matrix. Then, the expected Node Covariance computed from the stochastically projected features is invariant to the orthogonal transformation $\boldsymbol{Q}$:*

$$\mathbb{E}_{\boldsymbol{C}}[NodeCov(\boldsymbol{X}\boldsymbol{Q}\boldsymbol{C})] = \mathbb{E}_{\boldsymbol{C}}[NodeCov(\boldsymbol{X}\boldsymbol{C})] = \boldsymbol{\Pi}_c \boldsymbol{X}\boldsymbol{X}^T \boldsymbol{\Pi}_c \qquad (7)$$

*where the expectation $\mathbb{E}_{\boldsymbol{C}}[\cdot]$ is over the random sampling of $\boldsymbol{C}$, and $\boldsymbol{\Pi}_c \boldsymbol{X}\boldsymbol{X}^T \boldsymbol{\Pi}_c$ is the Gram matrix of the centered original features.*

Theorem 4.4 demonstrates that the expected operator is agnostic to any choice of orthonormal basis (rotations, reflections, permutations) for the input features. Furthermore, identifying this stable expectation as the Gram matrix of centered original features ($\boldsymbol{\Pi}_c \boldsymbol{X}\boldsymbol{X}^T \boldsymbol{\Pi}_c$) reveals that ALL-IN, on average, recovers intrinsic, basis-invariant pairwise node similarities directly reflecting the original data structure, irrespective of the specific random projection used.

## 4.2 TRAINING OBJECTIVE UPPER BOUND

ALL-IN computes the feature projection $\boldsymbol{R}^{(0)}$ and node-covariance operator $\boldsymbol{K}^{(0)} = $ NodeCov$(\boldsymbol{X}\boldsymbol{C})$ using a stochastic projection matrix $\boldsymbol{C}$ sampled in each forward pass. We now validate this practical training approach by showing its connection to performance on the stable, expected final representation $\mathbb{E}_{\boldsymbol{C}}[\boldsymbol{H}^{(L)}]$, assuming common convexity conditions for the final prediction layer.

Table 1: Performance of ALL-IN on pre-training datasets compared to ALL-IN-SPECIALIZED which is trained separately on each individual dataset. ALL-IN maintains highly competitive performance.

| Method | ZINC (MAE ↓) | MOLESOL (RMSE ↓) | MOLHIV (ROC-AUC ↑) | MOLTOX21 (ROC-AUC ↑) | MNIST (ACC ↑) | CIFAR10 (ACC ↑) | MODELNET (ACC ↑) | CUNEIFORM (ACC ↑) | MSRC 21 (ACC ↑) |
|---|---|---|---|---|---|---|---|---|---|
| **TRAINED PER DATASET** | | | | | | | | | |
| ALL-IN-SPECIALIZED (0 props) | 0.1480 | 1.22 | 72.65 | 69.37 | 94.03 | 39.96 | 37.24 | 85.19 | 91.65 |
| ALL-IN-SPECIALIZED | 0.1195 | 1.19 | 73.78 | 70.04 | 94.77 | 40.03 | 39.81 | 87.20 | 94.16 |
| **TRAINED ON ALL DATASETS** | | | | | | | | | |
| ALL-IN (0 props) | 0.1557 | 1.28 | 72.74 | 68.19 | 94.57 | 40.11 | 37.11 | 89.88 | 97.51 |
| ALL-IN | 0.1237 | 1.29 | 74.49 | 68.20 | 95.22 | 40.08 | 39.37 | 91.17 | 98.08 |

**Theorem 4.5** (Loss Upper Bound). *Let $\boldsymbol{H}^{(L)} \in \mathbb{R}^{n \times h^{(L)}}$ be the final node representations computed by* ALL-IN, *dependent on the initial random projection $\boldsymbol{C}$. Let $\phi : \mathbb{R}^{n \times h^{(L)}} \to \mathbb{R}^{n \times t}$ be the final prediction layer, and let $\mathcal{L}(\cdot, \boldsymbol{Y})$ be the loss function comparing predictions to ground truth labels $\boldsymbol{Y}$. Assume that the composite function $f(\boldsymbol{H}^{(L)}) = \mathcal{L}(\phi(\boldsymbol{H}^{(L)}), \boldsymbol{Y})$ is convex with respect to the final node representations $\boldsymbol{H}^{(L)}$. Then, our stochastic optimization objective provides an upper bound for the loss of the expected representation:*

$$\underbrace{\mathcal{L}(\phi(\mathbb{E}_{\boldsymbol{C}}[\boldsymbol{H}^{(L)}]), \boldsymbol{Y})}_{\text{Loss of Expected Representation}} \leq \underbrace{\mathbb{E}_{\boldsymbol{C}}[\mathcal{L}(\phi(\boldsymbol{H}^{(L)}), \boldsymbol{Y})]}_{\text{Expected Loss (Training Objective)}} \tag{8}$$

*where the expectation $\mathbb{E}_{\boldsymbol{C}}[\cdot]$ is taken over the random projection matrix $\boldsymbol{C}$.*

This holds, for instance, if $\phi$ is a linear map or linear plus softmax, and $\mathcal{L}$ is cross-entropy or mean squared error. Theorem 4.5 provides theoretical support for training with stochastic projections. Equation (8) establishes that the expected loss minimized during training (RHS) serves as an upper bound for the loss evaluated on the stable, expected final representation (LHS). Thus, minimizing the empirical average loss (approximating the RHS) acts as a theoretically sound surrogate objective, implicitly minimizing the loss associated with the expected representation, validating our stochastic approach.

### 4.3 CONDITIONS FOR TRANSFERABILITY AND OPERATOR CONSISTENCY

Beyond invariance, achieving transfer across graphs with fundamentally different feature distributions $(\boldsymbol{X}^{(1)}, \boldsymbol{X}^{(2)}$ for graphs $G_1, G_2)$ relies on the stability of the underlying structure captured by the expected operator, $\mathbb{E}_{\boldsymbol{C}}[\boldsymbol{K}^{(0)}] = \Pi_c \boldsymbol{X} \boldsymbol{X}^T \Pi_c$. We posit that such stability can arise when graphs share intrinsic properties. Plausible scenarios where such stability in the expected operator might arise include graphs exhibiting similar relational structures tied to node features (e.g., comparable label homophily if features reflect labels), originating from a shared underlying generative process (e.g., common SBM or graphon influencing features), or possessing similar distributions of node roles (e.g., hubs, bridges) if features are role-informative. In these cases, even if the specific feature realizations differ, the resulting $\Pi_c \boldsymbol{X}^{(i)} (\boldsymbol{X}^{(i)})^T \Pi_c$ matrices may capture analogous relational structures.

For this potential transfer to be practically realized, the stochastic operator $\boldsymbol{K}_h^{(0)}$ computed using a finite projection dimension $h$ must reliably estimate its expectation. This holds for large $h$.

**Proposition 4.6** (Consistency of Projected Node Covariance). *Let $\boldsymbol{X} \in \mathbb{R}^{n \times d}$ be node features. For a projection dimension $h$, let $\boldsymbol{C} \in \mathbb{R}^{d \times h}$ be such that $vec(\boldsymbol{C}) \sim \mathcal{N}(\boldsymbol{0}, \boldsymbol{I}_{dh})$. Define the stochastic node-covariance operator $\boldsymbol{K}_h^{(0)} = NodeCov(\boldsymbol{X} \boldsymbol{C}) = \frac{1}{h}(\Pi_c \boldsymbol{X} \boldsymbol{C})(\Pi_c \boldsymbol{X} \boldsymbol{C})^T$, where $\Pi_c$ is the centering matrix. Then, $\boldsymbol{K}_h^{(0)}$ converges in probability to its expected value as $h \to \infty$:*

$$\boldsymbol{K}_h^{(0)} \xrightarrow{p} \mathbb{E}_{\boldsymbol{C}}[\boldsymbol{K}_h^{(0)}] = \Pi_c \boldsymbol{X} \boldsymbol{X}^T \Pi_c \quad \text{as } h \to \infty. \tag{9}$$

This consistency connects theory to practice. It shows that for a sufficiently large $h$, the operator accurately reflects the stable expected operator $\Pi_c \boldsymbol{X} \boldsymbol{X}^T \Pi_c$. Therefore, if two graphs have aligned expected operators (due to shared properties), using a large enough $h$ allows ALL-IN to effectively leverage these shared underlying structures, facilitating transfer across disparate feature spaces.

Table 2: Performance on unseen node classification datasets with new input features. ALL-IN effectively transfers to new datasets with new features, often outperforming or matching SOTA.

| Method | CORA (ACC ↑) | CITESEER (ACC ↑) | PUBMED (ACC ↑) |
|---|---|---|---|
| **NON-PARAMETRIC BASELINES** | | | |
| LABEL PROPAGATION (Zhu & Ghahramani, 2002) | 69.20 ± 0.00 | 51.30 ± 0.00 | 71.40 ± 0.00 |
| **SUPERVISED BASELINES** | | | |
| MLP | 48.42 ± 0.63 | 48.56 ± 0.27 | 66.26 ± 1.53 |
| GCN (Kipf & Welling, 2017) | 78.86 ± 1.48 | 64.52 ± 0.89 | 74.49 ± 0.99 |
| GIN (Xu et al., 2019) | 67.10 ± 3.00 | 58.80 ± 2.20 | 68.40 ± 2.70 |
| **LLM-AUGMENTED GNNS** | | | |
| OFA (Liu et al., 2024) | 76.10 ± 4.11 | 73.04 ± 2.88 | 75.61 ± 5.06 |
| GLEM-LM (Chen et al., 2024b) | 67.55 ± 3.53 | 66.00 ± 5.66 | 62.12 ± 0.07 |
| **LLM-BASED** | | | |
| GRAPHTEXT (Zhao et al., 2023) | 75.41 ± 2.08 | 58.24 ± 0.26 | 63.70 ± 0.29 |
| RWNN-LLAMA3-8B (Kim et al., 2024) | 72.29 | N/A | N/A |
| **GNN-BASED** | | | |
| ANYGRAPH (Xia & Huang, 2024) | 62.60 ± 0.14 | 19.32 ± 0.37 | 70.73 ± 4.13 |
| GRAPHANY (Zhao et al., 2024b) | 79.36 ± 0.23 | 68.42 ± 0.39 | 76.30 ± 0.41 |
| MDGPT (Yu et al., 2024) | 43.36 ± 8.92 | 42.50 ± 9.78 | 51.91 ± 9.00 |
| GCOPE (Zhao et al., 2024a) | 35.54 ± 2.09 | 31.18 ± 4.35 | 32.87 ± 4.08 |
| GPPT (Sun et al., 2022) | 43.15 ± 9.44 | 37.26 ± 6.17 | 48.31 ± 17.72 |
| ALL-IN-ONE (Sun et al., 2023) | 52.39 ± 10.17 | 40.41 ± 2.80 | 45.17 ± 6.45 |
| GPROMPT (Gong et al., 2024) | 56.66 ± 11.22 | 53.21 ± 10.94 | 39.74 ± 15.35 |
| GPF (Fang et al., 2023) | 38.57 ± 5.41 | 31.16 ± 8.05 | 49.99 ± 8.86 |
| GPF-PLUS (Fang et al., 2023) | 55.77 ± 10.30 | 59.67 ± 11.87 | 46.64 ± 18.97 |
| ULTRA (3G) (Galkin et al., 2024) | 79.40 ± 0.00 | 67.40 ± 0.00 | 77.90 ± 0.00 |
| SCORE (Wang & Luo, 2024) | 81.80 ± 1.02 | 71.33 ± 0.27 | 82.93 ± 0.55 |
| OpenGraph (Xia et al., 2024) | N/A | 58.58 | 58.40 |
| RiemannGFM (Sun et al., 2025) | N/A | 66.38 | 76.20 |
| AutoGFM (Chen et al., 2025a) | 80.32 ± 1.12 | N/A | 78.28 ± 1.40 |
| ALL-IN (0 props) | 79.26 ± 1.08 | 65.96 ± 1.25 | 77.30 ± 0.47 |
| ALL-IN | 82.13 ± 0.97 | 69.12 ± 0.89 | 78.03 ± 0.82 |

## 5 EXPERIMENTS

In this section, we empirically evaluate the ability of ALL-IN to learn transferable representations from diverse graph datasets, and, critically, its capability to generalize to new datasets presenting entirely new input features. Our experiments are designed to answer two primary research questions:

(**Q1**) How does a single ALL-IN model, pre-trained jointly on a diverse collection of graph datasets (each with its own input features and task), perform on these individual source datasets compared to training a separate model for each dataset?

(**Q2**) How effectively do the representations learned by a pre-trained ALL-IN model transfer to new, unseen datasets that may have entirely different input features and downstream tasks?

Next, we report our main experiments and refer to Appendix C for additional results (including time complexity). Implementation details, dataset statistics, and hyperparameter configurations are in Appendices D and E.

### 5.1 PERFORMANCE ON PRE-TRAINING SOURCE DATASETS (A1)

In this subsection, we assess the ability of ALL-IN to learn from a wide array of source datasets simultaneously, without significant performance degradation on the individual datasets it was pre-trained on. This is needed for establishing its viability to obtain general-purpose pre-trained representations.

To test this, we pre-train a single ALL-IN encoder on a diverse corpus of nine graph datasets, encompassing molecular data (ZINC (Dwivedi et al., 2023), OGBG-MOLHIV (Hu et al., 2020a), OGBG-MOLESOL (Hu et al., 2020a), OGBG-MOLTOX21 (Hu et al., 2020a)), computer vision derived graphs (MNIST (Dwivedi et al., 2023), CIFAR10 (Dwivedi et al., 2023), CUNEIFORM (Morris et al., 2020), MSRC 21 (Morris et al., 2020)), and 3d shape (MODELNET (Wu et al., 2015)) with varying tasks (classification and regression) and heterogeneous input features (differing dimensionalities, types, value ranges and semantics). For each dataset-task pair, a dedicated prediction head is attached

to the shared ALL-IN component and trained to predict the corresponding target. We compare this single, jointly-trained model against its specialist counterparts: nine separate instances of the ALL-IN architecture, each trained from scratch on only one of the source datasets (ALL-IN-SPECIALIZED).

**Results and Discussion.** Table 1 confirms that ALL-IN not only successfully operates across datasets with heterogeneous features but is also highly effective, achieving performance competitive with, and at times superior to, specialized models. While the individually trained ALL-IN-SPECIALIZED holds a slight edge on ZINC, MOLESOL, MOLTOX21, and MODELNET, the jointly-trained ALL-IN demonstrates superior performance on the remaining 5 datasets. This advantage is particularly notable on CUNEIFORM (91.17% vs. 87.20%) and MSRC 21 (98.08% vs. 94.16%), while also outperforming ALL-IN-SPECIALIZED on MOLHIV, MNIST, and CIFAR10. We also observe a clear advantage to using propagated operators, as the full model generally outperforms the (0 props) variant (a version computed without propagated covariance operators) across the tasks.

Overall, these results strongly indicate that a single, jointly pre-trained ALL-IN encoder can learn general-purpose representations from diverse data that remain highly competitive with, and in several cases surpass, those obtained when learning on a single task.

## 5.2 TRANSFERABILITY TO UNSEEN DATASETS AND INPUT FEATURES (A2)

This subsection assesses the central hypothesis underlying our research: namely, that a single, pre-trained ALL-IN model can effectively generalize to novel datasets characterized by distinct input features. To evaluate this hypothesis, we maintain the pre-trained ALL-IN encoder frozen, thereby ensuring that its learned representations remain unchanged. For each new dataset, which encompasses a range of node and graph-level tasks and introduces previously unseen input features and target label schemas, we instantiate and train a new prediction head using the frozen representations extracted by ALL-IN. This approach enables us to isolate the generalizability of ALL-IN's pre-trained representations, providing a test of its ability to adapt to unfamiliar data distributions.

We compare ALL-IN against several categories of baselines: (1) the non-parametric baseline Label Propagation (Zhu & Ghahramani, 2002), on node classification tasks where it is applicable; (2) standard supervised GNNs trained from scratch on the target datasets; (3) LLM-augmented GNNs; (4) LLM-based methods; and (5) other GNN-based foundation models or transfer learning approaches. We adhere to their prescribed protocols for adaptation on new datasets. We refer the reader to Appendix C for this categorization.

**Results and Discussions.** ALL-IN demonstrates robust transferability across both node-level (Table 2) and graph-level (Table 3) tasks on unseen datasets with new input features. ALL-IN not only significantly surpasses the performance of standard supervised GNNs trained from scratch on these target datasets, but also outperforms recent state-of-the-art graph foundation

Table 3: Performance on unseen graph classification datasets with new input features. ALL-IN demonstrates strong transferability to graph-level tasks with new features, underscoring its versatility across different tasks and its ability to handle different features.

| Dataset | MUTAG (ACC ↑) | PROTEINS (ACC ↑) |
|---|---|---|
| **SUPERVISED BASELINES** | | |
| MLP | 67.20 ± 1.00 | 59.20 ± 1.00 |
| GIN (Xu et al., 2019) | 89.40 ± 5.60 | 76.20 ± 2.80 |
| **LLM-AUGMENTED GNNs** | | |
| OFA (Liu et al., 2024) | 61.04 ± 4.71 | 61.40 ± 2.99 |
| **GNN-BASED** | | |
| MDGPT (Yu et al., 2024) | 57.36 ±14.26 | 54.35 ±10.26 |
| GPPT (SUN ET AL., 2022) | 60.40 ±15.43 | 60.92 ± 2.47 |
| ALL-IN-ONE (Sun et al., 2023) | 79.87 ± 5.34 | 66.49 ± 6.26 |
| GPROMPT (Gong et al., 2024) | 73.60 ± 4.76 | 59.17 ±11.26 |
| GPF (Fang et al., 2023) | 68.40 ± 5.09 | 63.91 ± 3.26 |
| GPF-PLUS (Fang et al., 2023) | 65.20 ± 6.94 | 62.92 ± 2.78 |
| ULTRA(3G) (Galkin et al., 2024) | 63.33 ± 0.00 | 58.09 ± 0.00 |
| SCORE (Wang & Luo, 2024) | 85.33 ± 2.11 | 68.54 ± 1.47 |
| ALL-IN (0 props) | 92.50 ± 6.60 | 76.72 ± 3.19 |
| ALL-IN | 92.90 ± 6.34 | 78.20 ± 3.81 |

tion models. On node classification benchmarks (Table 2), ALL-IN consistently demonstrates strong transfer capabilities. For instance, on CORA (Kipf & Welling, 2017), it obtains an accuracy of 82.13% which not only surpasses standard supervised GCN (78.86%), but it also exceeds leading baselines like SCORE (Wang & Luo, 2024) (81.80%) and GRAPHANY (Zhao et al., 2024b) (79.36%). This strong performance extends to graph classification tasks (Table 3). On MUTAG (Morris et al., 2020), ALL-IN achieves 92.90% accuracy, exceeding both the supervised GIN baseline (89.40%) and state of the art methods like SCORE (85.33%) and ALL-IN-ONE (Sun et al., 2023) (79.87%). Furthermore, consistent with observations on the source datasets in Section 5.1, the inclusion of

propagated covariance operators in ALL-IN enhances transfer performance compared to ALL-IN (0 props).

These results provide evidence that a single pre-trained ALL-IN encoder produces effective, general-purpose representations. These representations readily adapt to both node and graph-level tasks on new datasets with new features, maintaining a versatility that provides a strong advantage over specialized models (GRAPHANY (Zhao et al., 2024b), GRAPHTEXT (Zhao et al., 2023), GCOPE (Zhao et al., 2024a), ANYGRAPH (Xia & Huang, 2024)), only supporting node classification.

## 6 CONCLUSION

Input feature heterogeneity critically limits the development of Graph Foundation Models (GFMs). Our ALL-IN offers a theoretically-grounded solution, processing arbitrary node features through stochastic projections and node-covariance operators to build robust representations independent of the original feature space. We prove that these representations achieve distributional invariance to input feature permutations, and their underlying expected operator is invariant to orthogonal basis changes, thereby helping capture robust intrinsic structures of the data. The empirical transfer performance of ALL-IN across new datasets with disparate features demonstrates its potential to mitigate the challenges posed by feature heterogeneity, contributing to the development of GFMs.

**Limitations and Future Work.** The scalability of ALL-IN on extremely large graphs may be constrained by its dense covariance operators, in case direct access to the covariance operators are required, similarly to graph transformers; developing sparse approximations presents a key avenue for future research. Another promising direction involves exploring structured or learnable input feature projections as alternatives to the random Gaussian projections. Notably, as discussed in Appendix C.17, in common GNNs, we can avoid the storage of dense covariance operators, thereby achieving improved scalability.

**Reproducibility Statement.** Our code is available at https://github.com/MosheEliasof/ALLIN. We carefully document dataset details in Appendix D and implementation details in Appendix E.

**Ethics Statement.** Our work is primarily methodological and presents minimal direct ethical concerns. All experiments are conducted on publicly available benchmark datasets widely used in the graph machine learning community, and we have used these datasets in accordance with their established licensing and terms of use. While our contribution is foundational, we advocate for the responsible application of transferable graph models. We caution against their use in analyzing sensitive social or personal data without appropriate safeguards and ethical oversight.

**Usage of Large Language Models in This Work.** LLMs were used in this work for text editing suggestions. All concepts, theoretical analysis, code development, and original writing were carried out by the authors.

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

## A  ADDITIONAL RELATED WORK

**Generalization Theory of MPNNs.**  Significant theoretical progress has advanced our understanding of generalization in Message Passing Neural Networks (MPNNs). As discussed in recent surveys (Vasileiou et al., 2025; Zhang et al., 2024a), these efforts often focus on how architectures and graph properties (such as maximum degree) influence the generalization gap, employing analytical tools like Rademacher complexity and PAC-Bayesian analysis to derive performance bounds (Garg et al., 2020; Liao et al., 2021). Other lines of work, leveraging concepts like covering numbers or graphon theory, investigate model stability and generalization under shifts in graph structure or topology, particularly in large-scale or evolving graph scenarios (Levie, 2023; Vasileiou et al., 2024). While these foundational theories provide important insights into GNN expressivity and their ability to generalize, especially concerning structural variations, they typically assume a consistent definition of the input feature space across different graphs. The cross-dataset generalization challenge that ALL-IN addresses is distinct: we specifically tackle scenarios where graphs present node features from entirely different feature spaces, potentially varying in both the number of available features (dimensionality) and their semantic meaning between train (source) and test (target) graphs. Our theoretical framework (Section 4) therefore focuses on establishing principles for robustness and transferability under such input feature space heterogeneity, aiming to complement existing generalization theories that predominantly address structural changes.

**Additional Efforts towards Graph Foundation Models.**  Another significant challenge in graph transfer learning arises in settings like heterogeneous knowledge graphs, where models must generalize to unseen entities and relation types. Approaches such as ISDEA+ (Gao et al., 2023) and MTDEA (Zhou et al., 2023) tackle this by employing set aggregation techniques over representations specific to edge types, aiming for equivariance to permutations of these types, supported by a "double equivariance" theoretical framework. Similarly, methods like InGram (Lee et al., 2023), ULTRA (Galkin et al., 2024), TRIX (Zhang et al., 2024b), and MOTIF (Huang et al., 2025) construct explicit "relation graphs" to model interactions among different edge types. These works provide valuable solutions for structural and relational heterogeneity. In contrast, ALL-IN primarily addresses the distinct challenge of heterogeneity in input features, that is, varying feature dimensionalities and semantics across graphs. While the aforementioned methods focus on generalizing over graph schema and relation types (often assuming node features are not present), ALL-IN directly processes arbitrary node features to derive transferable node-covariance operators and representations. Other efforts in graph representation learning aim for transferability across diverse graph tasks. For example, HoloGNN (Bevilacqua et al., 2025) proposes a framework to learn node representations that can be applied to various downstream tasks on a given graph or graphs. However, such approaches typically assume that the underlying node feature space remains consistent across these tasks. ALL-IN, conversely, is specifically designed to address the challenge of generalizing to new and unseen datasets where the node features themselves can differ fundamentally in dimensionality and semantics, a problem distinct from task-level transfer within a fixed feature domain.

## B  ADDITIONAL THEORETICAL CONSIDERATIONS AND PROOFS

**Proposition 4.1** (Distributional Invariance of Projected Features to Feature Permutation)**.** *Let* $\boldsymbol{X} \in \mathbb{R}^{n \times d}$ *be node features,* $\boldsymbol{P} \in \mathbb{R}^{d \times d}$ *be any permutation matrix, and* $h$ *be the projection dimension. Let* $\boldsymbol{C} \in \mathbb{R}^{d \times h}$ *be an isotropic Gaussian random matrix (i.e.,* $vec(\boldsymbol{C}) \sim \mathcal{N}(\boldsymbol{0}, \boldsymbol{I}_{dh})$*). Define the projected features as* $\boldsymbol{R}^{(0)} = \boldsymbol{X}\boldsymbol{C}$ *and the features projected after permutation as* $\bar{\boldsymbol{R}}^{(0)} = (\boldsymbol{X}\boldsymbol{P})\boldsymbol{C}$*. Then* $\boldsymbol{R}^{(0)}$ *and* $\bar{\boldsymbol{R}}^{(0)}$ *are equal in distribution:* $\boldsymbol{R}^{(0)} \stackrel{d}{=} \bar{\boldsymbol{R}}^{(0)}$*.*

*Proof.* Let $\boldsymbol{C}$ have columns $\boldsymbol{c}_1, \ldots, \boldsymbol{c}_h$. Since the entries $C_{ik}$ are i.i.d $\mathcal{N}(0, 1)$, each column $\boldsymbol{c}_j \sim \mathcal{N}(\boldsymbol{0}, \boldsymbol{I}_d)$ and the columns are mutually independent.

Consider the matrix $\bar{\boldsymbol{C}} = \boldsymbol{P}^T \boldsymbol{C}$. Since $\boldsymbol{P}$ is a permutation matrix, $\boldsymbol{P}^T$ is also a permutation matrix and is orthogonal, that is $\boldsymbol{P}^T (\boldsymbol{P}^T)^T = \boldsymbol{P}^T \boldsymbol{P} = \boldsymbol{I}_d$.

The columns of $\bar{\boldsymbol{C}}$ are $\bar{\boldsymbol{c}}_j = \boldsymbol{P}^T \boldsymbol{c}_j$. Since $\boldsymbol{c}_j \sim \mathcal{N}(\boldsymbol{0}, \boldsymbol{I}_d)$ and $\boldsymbol{P}^T$ is orthogonal, then

$$\bar{\boldsymbol{c}}_j \sim \mathcal{N}(\boldsymbol{P}^T \boldsymbol{0}, \boldsymbol{P}^T \boldsymbol{I}_d (\boldsymbol{P}^T)^T) = \mathcal{N}(\boldsymbol{0}, \boldsymbol{P}^T \boldsymbol{P}) = \mathcal{N}(\boldsymbol{0}, \boldsymbol{I}_d) \tag{10}$$

Furthermore, since $c_1, \ldots, c_h$ are independent, the transformed columns $\bar{c}_1, \ldots, \bar{c}_h$ are also independent. Thus, the matrix $\bar{C}$ has the same distribution as $C$, i.e., $\bar{C} \stackrel{d}{=} C$.

Now consider $\bar{R}^{(0)} = (XP)C$. Since $C \stackrel{d}{=} \bar{C}$, we can write:

$$\bar{R}^{(0)} \stackrel{d}{=} (XP)\bar{C}$$

Substitute $\bar{C} = P^T C$:

$$\bar{R}^{(0)} \stackrel{d}{=} (XP)(P^T C) = X(PP^T)C$$

Since $P$ is orthogonal, $PP^T = I_d$.

$$\bar{R}^{(0)} \stackrel{d}{=} X I_d C = XC = R$$

Thus, $R$ and $\bar{R}^{(0)}$ are equal in distribution. $\qquad\square$

**Corollary 4.2** (Distributional Invariance of Node Covariance Operators to Feature Permutation). *Let $X \in \mathbb{R}^{n \times d}$ be node features, and $P \in \mathbb{R}^{d \times d}$ be any permutation matrix. Let $R^{(0)} = XC$ be the initial projected features. Let $\mathcal{K} = \{K^{(p)}\}_{p=0}^{k}$ be the set of node-covariance operators, where $K^{(p)} = NodeCov(A^p R^{(0)})$ is computed using the deterministic function NodeCov (Equation (3)), and $A$ is the adjacency matrix. It follows directly from the distributional invariance of $R^{(0)}$ that the entire set of operators $\mathcal{K}$ is also invariant in distribution to permutations of the input features $X$. That is, if $\bar{\mathcal{K}}$ is the set of operators computed using $XP$ instead of $X$, then $\mathcal{K} \stackrel{d}{=} \bar{\mathcal{K}}$.*

*Proof.* Let $g_p(R^{(0)}) = \text{NodeCov}(A^p R^{(0)})$ be the deterministic function that computes the p-th order node covariance operator from the initial projected features $R^{(0)}$. From Proposition 4.1, we have $R^{(0)} \stackrel{d}{=} \bar{R}^{(0)}$. Since applying a deterministic function $g_p$ to random variables that are equal in distribution results in outputs that are equal in distribution, we have $g_p(R^{(0)}) \stackrel{d}{=} g_p(\bar{R}^{(0)})$, which means $K^{(p)} \stackrel{d}{=} \bar{K}^{(p)}$ for each $p = 0 \ldots k$. Furthermore, since all operators $K^{(p)}$ in $\mathcal{K}$ are derived from the same $R^{(0)}$, and all operators $\bar{K}^{(p)}$ in $\bar{\mathcal{K}}$ are derived from $\bar{R}^{(0)}$, the distributional equality extends to the joint distribution of the sets: $\mathcal{K} \stackrel{d}{=} \bar{\mathcal{K}}$. $\qquad\square$

**Theorem B.1** (Distributional Invariance of Hidden Representations to Input Permutation). *Let $X \in \mathbb{R}^{n \times d}$ be node features, and $P \in \mathbb{R}^{d \times d}$ be any permutation matrix. Let $R^{(0)} = XC$ be the initial projected features, and $\mathcal{K} = \{K^{(p)}\}_{p=0}^{k}$ be the set of node-covariance operators. Let the initial hidden representation be $H^{(0)} = R^{(0)} \oplus S$, where $S$ is a structural encoding matrix independent of $X$. Subsequent hidden representations $H^{(\ell)}$ for $\ell = 1, \ldots, L$ are computed by a deterministic GNN layer function.*

*The initial hidden representation $H^{(0)}$ and all subsequent hidden representations $H^{(\ell)}$ for $\ell = 1, \ldots, L$ are invariant in distribution to permutations of the input features $X$. That is, if $\bar{H}^{(\ell)}$ are the representations computed using $XP$ instead of $X$, then $H^{(\ell)} \stackrel{d}{=} \bar{H}^{(\ell)}$ for all $\ell$.*

*Proof.* We proceed by induction on the layer index $\ell$.

**Base Case ($\ell = 0$).** Let $R^{(0)} = XC$ and $\bar{R}^{(0)} = (XP)C$. The initial hidden representations are $H^{(0)} = R^{(0)} \oplus S$ and $\bar{H}^{(0)} = \bar{R}^{(0)} \oplus S$. From Proposition 4.1, we know that $R^{(0)} \stackrel{d}{=} \bar{R}^{(0)}$. Since the structural encoding $S$ is assumed independent of $X$ (and thus fixed with respect to the permutation $P$), and the concatenation operation $\oplus$ is a deterministic function, applying this function preserves the distributional equality. Therefore, $H^{(0)} = R^{(0)} \oplus S \stackrel{d}{=} \bar{R}^{(0)} \oplus S = \bar{H}^{(0)}$. The base case holds.

**Inductive Hypothesis.** Assume that for some layer $\ell - 1 \geq 0$, the hidden representations are equal in distribution: $H^{(\ell-1)} \stackrel{d}{=} \bar{H}^{(\ell-1)}$.

**Inductive Step (Layer $\ell$).** The hidden representations at layer $\ell$ are computed as:

$$H^{(\ell)} = F_\ell(H^{(\ell-1)}, \mathcal{O})$$

$$\bar{H}^{(\ell)} = F_\ell(\bar{H}^{(\ell-1)}, \bar{\mathcal{O}})$$

where $F_\ell$ represents the deterministic computation performed by the $\ell$-th GNN layer (given fixed learned weights), $\mathcal{O} = \{I, A\} \cup \mathcal{K}$ with $\mathcal{K} = \{\text{NodeCov}(A^p R^{(0)})\}_{p=0}^k$, and $\bar{\mathcal{O}} = \{I, A\} \cup \bar{\mathcal{K}}$ with $\bar{\mathcal{K}} = \{\text{NodeCov}(A^p \bar{R}^{(0)})\}_{p=0}^k$.

From Corollary 4.2, we know that the set of random operators $\mathcal{K}$ is equal in distribution to $\bar{\mathcal{K}}$, i.e., $\mathcal{K} \overset{d}{=} \bar{\mathcal{K}}$. Since $I$ and $A$ are fixed, the full set of operators used by the layer also satisfies $\mathcal{O} \overset{d}{=} \bar{\mathcal{O}}$.

Now consider the inputs to the function $F_\ell$. The pair $(H^{(\ell-1)}, \mathcal{O})$ determines $H^{(\ell)}$, and the pair $(\bar{H}^{(\ell-1)}, \bar{\mathcal{O}})$ determines $\bar{H}^{(\ell)}$. Both $H^{(\ell-1)}$ and $\mathcal{O}$ are deterministic functions of the initial projection $R^{(0)}$ (and fixed elements $S, A, I$, and layer weights). Let $J$ be the function representing the computation up to layer $\ell - 1$ and the computation of operators, such that $(H^{(\ell-1)}, \mathcal{O}) = J(R^{(0)}, S, A, I, \text{Weights})$ Similarly, $(\bar{H}^{(\ell-1)}, \bar{\mathcal{O}}) = J(\bar{R}^{(0)}, S, A, I, \text{Weights})$.

Since $R^{(0)} \overset{d}{=} \bar{R}^{(0)}$ (Proposition 4.1) and $J$ is a deterministic function, it follows that the joint distribution of the outputs is preserved:

$$(H^{(\ell-1)}, \mathcal{O}) \overset{d}{=} (\bar{H}^{(\ell-1)}, \bar{\mathcal{O}})$$

This establishes that the inputs to the deterministic layer function $F_\ell$ are equal in distribution. Applying the deterministic function $F_\ell$ preserves this equality:

$$H^{(\ell)} = F_\ell(H^{(\ell-1)}, \mathcal{O}) \overset{d}{=} F_\ell(\bar{H}^{(\ell-1)}, \bar{\mathcal{O}}) = \bar{H}^{(\ell)}$$

Thus, the inductive step holds. $\qquad\qquad\square$

**Theorem 4.3** (Distinguishability through $C$). *There exist node features $X \in \mathbb{R}^{n \times d}$, nodes $u, v \in V$ with $X_u \neq X_v$ such that $NodeCov(X)$ makes $u, v$ indistinguishable (automorphic), but $NodeCov(XC)$ (for a.s. all $C$) makes $u, v$ distinguishable (not automorphic).*

*Proof.* We will show that there exists $X$, $u$, $v$ such that (1) nodes $u$ and $v$ are automorphic within $\text{NodeCov}(X)$, and consequently, the GNN, when using $\text{NodeCov}(X)$ as the operator and identical initial embeddings, produces identical final representations for these nodes. (2) For the same $X$, with probability 1 (over the draw of $C$), nodes $u$ and $v$ are **not** automorphic and therefore distinguishable in $\text{NodeCov}(XC)$. We provide a constructive example. Let $n = 3$ nodes $\{u, v, w\}$ and $d = 3$ features. Consider the feature matrix $X$:

$$X = \begin{pmatrix} X_u^T \\ X_v^T \\ X_w^T \end{pmatrix} = \begin{pmatrix} 1 & 0 & 1 \\ 0 & 1 & 1 \\ 1 & 1 & 0 \end{pmatrix}$$

Here, $X_u = (1, 0, 1)^T$, $X_v = (0, 1, 1)^T$, and $X_w = (1, 1, 0)^T$. Clearly, $X_u \neq X_v$.

**Proof for item (1).** The column means of $X$ are $\bar{X}_{\text{col}} = (2/3, 2/3, 2/3)^T$. The centered feature matrix $X_c = \Pi_c X = X - \mathbf{1}_3 \bar{X}_{\text{col}}^T$ is:

$$X_c = \begin{pmatrix} 1/3 & -2/3 & 1/3 \\ -2/3 & 1/3 & 1/3 \\ 1/3 & 1/3 & -2/3 \end{pmatrix}$$

Then

$$\text{NodeCov}(X) = \begin{pmatrix} 2/9 & -1/9 & -1/9 \\ -1/9 & 2/9 & -1/9 \\ -1/9 & -1/9 & 2/9 \end{pmatrix}.$$

In the weighted graph defined by $\text{NodeCov}(X)$, all nodes are automorphic to each other. If a GNN uses $\text{NodeCov}(X)$ as its feature-derived operator and starts with identical initial embeddings for all nodes, standard message passing layers will preserve this symmetry, leading to identical final representations $H_u^{(L)} = H_v^{(L)} = H_w^{(L)}$. Thus, such a GNN cannot distinguish $u$ from $v$.

**Proof for item (2).** Let $R^{(0)} = XC$. The rows of $R^{(0)}$ are $R_u^{(0)} = X_u^T C$, $R_v^{(0)} = X_v^T C$, $R_w^{(0)} = X_w^T C$. Since $X_u \neq X_v$ and $C$ is drawn from a continuous distribution (Gaussian entries), $X_u^T C \neq$

$\boldsymbol{X}_v^T \boldsymbol{C}$ with probability 1. Thus, $\boldsymbol{R}_u^{(0)} \neq \boldsymbol{R}_v^{(0)}$ almost surely. Let $\boldsymbol{R}_c^{(0)} = \boldsymbol{\Pi}_c \boldsymbol{R}^{(0)}$. The rows of $\boldsymbol{R}_c$ are $\boldsymbol{R}_{c,u}^{(0)}, \boldsymbol{R}_{c,v}^{(0)}, \boldsymbol{R}_{c,w}^{(0)}$. Since $\boldsymbol{R}_u^{(0)} \neq \boldsymbol{R}_v^{(0)}$, it follows that $\boldsymbol{R}_{c,u}^{(0)} \neq \boldsymbol{R}_{c,v}^{(0)}$ almost surely (unless $\boldsymbol{\Pi}_c$ projects their difference to zero, which is a measure zero event for a fixed $\boldsymbol{X}$ and random $\boldsymbol{C}$). The operator is $\boldsymbol{K}^{(0)} = \text{NodeConv}(\boldsymbol{XC}) = \frac{1}{h}\boldsymbol{R}_c \boldsymbol{R}_c^T$. An element $(\boldsymbol{K}^{(0)})_{ij} = \frac{1}{h}\boldsymbol{R}_{c,i}^{(0)} \cdot \boldsymbol{R}_{c,j}^{(0)}$. Consider the specific symmetry that existed for $\text{NodeConv}(\boldsymbol{X})$, e.g., $(\text{NodeConv}(\boldsymbol{X}))_{uw} = (\text{NodeConv}(\boldsymbol{X}))_{vw} = -1/9$. For $\boldsymbol{K}^{(0)}$, we compare $(\boldsymbol{K}^{(0)})_{uw} = \frac{1}{h}\boldsymbol{R}_{c,u}^{(0)} \cdot \boldsymbol{R}_{c,w}^{(0)}$ and $(\boldsymbol{K}^{(0)})_{vw} = \frac{1}{h}\boldsymbol{R}_{c,v}^{(0)} \cdot \boldsymbol{R}_{c,w}^{(0)}$. These are equal if $(\boldsymbol{R}_{c,u}^{(0)} - \boldsymbol{R}_{c,v}^{(0)}) \cdot \boldsymbol{R}_{c,w}^{(0)} = 0$. Since $\boldsymbol{R}_{c,u}^{(0)} - \boldsymbol{R}_{c,v}^{(0)} \neq \boldsymbol{0}$ almost surely, and $\boldsymbol{R}_{c,w}^{(0)}$ is a random vector (whose distribution depends on $\boldsymbol{C}$), the event that their dot product is exactly zero has probability 0 for continuous distributions unless one of them is deterministically zero (which is not the case here a.s.). Therefore, with probability 1, $(\boldsymbol{K}^{(0)})_{uw} \neq (\boldsymbol{K}^{(0)})_{vw}$. This breaks the specific symmetry that made node u and node v have equivalent relational profiles to node w in $\text{NodeCov}(\boldsymbol{X})$. More generally, the matrix $\boldsymbol{K}^{(0)}$ will not, with probability 1, exhibit the high degree of symmetry found in $\text{NodeCov}(\boldsymbol{X})$ for this specific $\boldsymbol{X}$. Thus, nodes $u$ and $v$ will generally not be automorphic with respect to $\boldsymbol{K}^{(0)}$ in the same way they were for $\text{NodeCov}(\boldsymbol{X})$. A GNN using this specific realization $\boldsymbol{K}^{(0)}$ (and identical initial embeddings, can now potentially produce $\boldsymbol{H}_u^{(L)} \neq \boldsymbol{H}_v^{(L)}$ because the operator $\boldsymbol{K}^{(0)}$ provides different relational information for $u$ and $v$.

$\square$

**Theorem 4.4** (Expected Invariance to Orthogonal Transformations). *Let $\boldsymbol{X} \in \mathbb{R}^{n \times d}$ be node features, $\boldsymbol{Q} \in \mathbb{R}^{d \times d}$ be an orthogonal matrix, and h be the projection dimension. Consider a random projection matrix $\boldsymbol{C} \in \mathbb{R}^{d \times h}$ with $vec(\boldsymbol{C}) \sim \mathcal{N}(\boldsymbol{0}, \boldsymbol{I}_{dh})$. Let $NodeCov(\boldsymbol{R}^{(0)}) = \frac{1}{h}(\boldsymbol{\Pi}_c \boldsymbol{R}^{(0)})(\boldsymbol{\Pi}_c \boldsymbol{R}^{(0)})^T$ be the Node Covariance operator (Equation (2)), where $\boldsymbol{\Pi}_c = \boldsymbol{I}_n - \frac{1}{n}\boldsymbol{1}_n \boldsymbol{1}_n^T$ is the centering matrix. Then, the expected Node Covariance computed from the stochastically projected features is invariant to the orthogonal transformation $\boldsymbol{Q}$:*

$$\mathbb{E}_{\boldsymbol{C}}[NodeCov(\boldsymbol{XQC})] = \mathbb{E}_{\boldsymbol{C}}[NodeCov(\boldsymbol{XC})] = \boldsymbol{\Pi}_c \boldsymbol{X}\boldsymbol{X}^T \boldsymbol{\Pi}_c \quad (7)$$

*where the expectation $\mathbb{E}_{\boldsymbol{C}}[\cdot]$ is over the random sampling of $\boldsymbol{C}$, and $\boldsymbol{\Pi}_c \boldsymbol{X}\boldsymbol{X}^T \boldsymbol{\Pi}_c$ is the Gram matrix of the centered original features.*

*Proof.* Let $\boldsymbol{R}^{(0)} = \boldsymbol{XC}$. Using the definition of the NodeCov operator and properties of the centering matrix $\boldsymbol{\Pi}_c$:

$$\begin{aligned}
\text{NodeCov}(\boldsymbol{R}^{(0)}) &= \frac{1}{h}(\boldsymbol{\Pi}_c \boldsymbol{R}^{(0)})(\boldsymbol{\Pi}_c \boldsymbol{R}^{(0)})^T \\
&= \frac{1}{h}\boldsymbol{\Pi}_c(\boldsymbol{XC})(\boldsymbol{XC})^T \boldsymbol{\Pi}_c^T \\
&= \frac{1}{h}\boldsymbol{\Pi}_c \boldsymbol{X}\boldsymbol{C}\boldsymbol{C}^T \boldsymbol{X}^T \boldsymbol{\Pi}_c
\end{aligned}$$

Taking the expectation over $\boldsymbol{C}$:

$$\begin{aligned}
\mathbb{E}_{\boldsymbol{C}}[\text{NodeCov}(\boldsymbol{XC})] &= \mathbb{E}_{\boldsymbol{C}}\left[\frac{1}{h}\boldsymbol{\Pi}_c \boldsymbol{X}\boldsymbol{C}\boldsymbol{C}^T \boldsymbol{X}^T \boldsymbol{\Pi}_c\right] \\
&= \frac{1}{h}\boldsymbol{\Pi}_c \boldsymbol{X}\mathbb{E}_{\boldsymbol{C}}[\boldsymbol{C}\boldsymbol{C}^T]\boldsymbol{X}^T \boldsymbol{\Pi}_c \quad \text{(by linearity of expectation)}
\end{aligned}$$

We evaluate $\mathbb{E}_{\boldsymbol{C}}[\boldsymbol{C}\boldsymbol{C}^T]$. Let $\boldsymbol{c}_j \in \mathbb{R}^d$ be the j-th column of $\boldsymbol{C}$. Since the entries of $\boldsymbol{C}$ are i.i.d. $\mathcal{N}(0, 1)$, each column vector $\boldsymbol{c}_j$ follows $\boldsymbol{c}_j \sim \mathcal{N}(\boldsymbol{0}, \boldsymbol{I}_d)$. Therefore, $E[\boldsymbol{c}_j \boldsymbol{c}_j^T] = \boldsymbol{I}_d$. Using linearity of expectation:

$$\mathbb{E}_{\boldsymbol{C}}[\boldsymbol{C}\boldsymbol{C}^T] = \mathbb{E}_{\boldsymbol{C}}\left[\sum_{j=1}^h \boldsymbol{c}_j \boldsymbol{c}_j^T\right] = \sum_{j=1}^h \mathbb{E}_{\boldsymbol{C}}[\boldsymbol{c}_j \boldsymbol{c}_j^T] = \sum_{j=1}^h \boldsymbol{I}_d = h\boldsymbol{I}_d$$

Substituting this back:

$$\mathbb{E}_{\boldsymbol{C}}[\text{NodeCov}(\boldsymbol{XC})] = \frac{1}{h}\boldsymbol{\Pi}_c \boldsymbol{X}(h\boldsymbol{I}_d)\boldsymbol{X}^T \boldsymbol{\Pi}_c = \boldsymbol{\Pi}_c \boldsymbol{X}\boldsymbol{X}^T \boldsymbol{\Pi}_c$$

Now consider the transformed features $\bar{\boldsymbol{X}} = \boldsymbol{X}\boldsymbol{Q}$. Let $\bar{\boldsymbol{R}}^{(0)} = \bar{\boldsymbol{X}}\boldsymbol{C} = \boldsymbol{X}\boldsymbol{Q}\boldsymbol{C}$. We compute $\mathbb{E}_{\boldsymbol{C}}[\mathrm{NodeCov}(\bar{\boldsymbol{R}}^{(0)})]$:

$$\mathrm{NodeCov}(\bar{\boldsymbol{R}}^{(0)}) = \frac{1}{h}(\boldsymbol{\Pi}_c \bar{\boldsymbol{R}}^{(0)})(\boldsymbol{\Pi}_c \bar{\boldsymbol{R}}^{(0)})^T$$
$$= \frac{1}{h}\boldsymbol{\Pi}_c(\boldsymbol{X}\boldsymbol{Q}\boldsymbol{C})(\boldsymbol{X}\boldsymbol{Q}\boldsymbol{C})^T\boldsymbol{\Pi}_c$$
$$= \frac{1}{h}\boldsymbol{\Pi}_c\boldsymbol{X}\boldsymbol{Q}\boldsymbol{C}\boldsymbol{C}^T\boldsymbol{Q}^T\boldsymbol{X}^T\boldsymbol{\Pi}_c$$

Taking the expectation over $\boldsymbol{C}$:

$$\mathbb{E}_{\boldsymbol{C}}[\mathrm{NodeCov}(\boldsymbol{X}\boldsymbol{Q}\boldsymbol{C})] = \frac{1}{h}\boldsymbol{\Pi}_c\boldsymbol{X}\boldsymbol{Q}\mathbb{E}_{\boldsymbol{C}}[\boldsymbol{C}\boldsymbol{C}^T]\boldsymbol{Q}^T\boldsymbol{X}^T\boldsymbol{\Pi}_c$$
$$= \frac{1}{h}\boldsymbol{\Pi}_c\boldsymbol{X}\boldsymbol{Q}(h\boldsymbol{I}_d)\boldsymbol{Q}^T\boldsymbol{X}^T\boldsymbol{\Pi}_c \quad \text{(using } E[\boldsymbol{C}\boldsymbol{C}^T] = h\boldsymbol{I}_d)$$
$$= \boldsymbol{\Pi}_c\boldsymbol{X}\boldsymbol{Q}\boldsymbol{I}_d\boldsymbol{Q}^T\boldsymbol{X}^T\boldsymbol{\Pi}_c$$
$$= \boldsymbol{\Pi}_c\boldsymbol{X}(\boldsymbol{Q}\boldsymbol{Q}^T)\boldsymbol{X}^T\boldsymbol{\Pi}_c$$
$$= \boldsymbol{\Pi}_c\boldsymbol{X}\boldsymbol{I}_d\boldsymbol{X}^T\boldsymbol{\Pi}_c \quad \text{(since } \boldsymbol{Q} \text{ is orthogonal, } \boldsymbol{Q}\boldsymbol{Q}^T = \boldsymbol{I}_d)$$
$$= \boldsymbol{\Pi}_c\boldsymbol{X}\boldsymbol{X}^T\boldsymbol{\Pi}_c$$

Thus, $\mathbb{E}_{\boldsymbol{C}}[\mathrm{NodeCov}(\boldsymbol{X}\boldsymbol{Q}\boldsymbol{C})] = \mathbb{E}_{\boldsymbol{C}}[\mathrm{NodeCov}(\boldsymbol{X}\boldsymbol{C})] = \boldsymbol{\Pi}_c\boldsymbol{X}\boldsymbol{X}^T\boldsymbol{\Pi}_c$. □

**Theorem 4.5** (Loss Upper Bound). *Let $\boldsymbol{H}^{(L)} \in \mathbb{R}^{n \times h^{(L)}}$ be the final node representations computed by* ALL-IN*, dependent on the initial random projection $\boldsymbol{C}$. Let $\phi : \mathbb{R}^{n \times h^{(L)}} \to \mathbb{R}^{n \times t}$ be the final prediction layer, and let $\mathcal{L}(\cdot, \boldsymbol{Y})$ be the loss function comparing predictions to ground truth labels $\boldsymbol{Y}$. Assume that the composite function $f(\boldsymbol{H}^{(L)}) = \mathcal{L}(\phi(\boldsymbol{H}^{(L)}), \boldsymbol{Y})$ is convex with respect to the final node representations $\boldsymbol{H}^{(L)}$. Then, our stochastic optimization objective provides an upper bound for the loss of the expected representation:*

$$\underbrace{\mathcal{L}(\phi(\mathbb{E}_{\boldsymbol{C}}[\boldsymbol{H}^{(L)}]), \boldsymbol{Y})}_{\textit{Loss of Expected Representation}} \leq \underbrace{\mathbb{E}_{\boldsymbol{C}}[\mathcal{L}(\phi(\boldsymbol{H}^{(L)}), \boldsymbol{Y})]}_{\textit{Expected Loss (Training Objective)}} \tag{8}$$

*where the expectation $\mathbb{E}_{\boldsymbol{C}}[\cdot]$ is taken over the random projection matrix $\boldsymbol{C}$.*

*Proof.* The proof follows directly from Jensen's inequality for vector- or matrix-valued random variables.

Let the random variable be the final hidden representation $Z = \boldsymbol{H}^{(L)}$, which is a function of the random projection matrix $\boldsymbol{C}$.

By assumption, the function $f$ is convex with respect to its input argument $\boldsymbol{H}^{(L)}$. Jensen's inequality states that for a convex function $f$ and a random variable $Z$ with finite expectation, $f(\mathbb{E}[Z]) \leq \mathbb{E}[f(Z)]$. Applying this with $Z = \boldsymbol{H}^{(L)}$ and the defined function $f$, we get:

$$\mathcal{L}(\phi(\mathbb{E}_{\boldsymbol{C}}[\boldsymbol{H}^{(L)}]), \boldsymbol{Y}) \leq \mathbb{E}_{\boldsymbol{C}}[\mathcal{L}(\phi(\boldsymbol{H}^{(L)}), \boldsymbol{Y})]$$

which is the desired result. □

**Proposition 4.6** (Consistency of Projected Node Covariance). *Let $\boldsymbol{X} \in \mathbb{R}^{n \times d}$ be node features. For a projection dimension $h$, let $\boldsymbol{C} \in \mathbb{R}^{d \times h}$ be such that $\mathrm{vec}(\boldsymbol{C}) \sim \mathcal{N}(\boldsymbol{0}, \boldsymbol{I}_{dh})$. Define the stochastic node-covariance operator $\boldsymbol{K}_h^{(0)} = \mathrm{NodeCov}(\boldsymbol{X}\boldsymbol{C}) = \frac{1}{h}(\boldsymbol{\Pi}_c\boldsymbol{X}\boldsymbol{C})(\boldsymbol{\Pi}_c\boldsymbol{X}\boldsymbol{C})^T$, where $\boldsymbol{\Pi}_c$ is the centering matrix. Then, $\boldsymbol{K}_h^{(0)}$ converges in probability to its expected value as $h \to \infty$:*

$$\boldsymbol{K}_h^{(0)} \xrightarrow{p} \mathbb{E}_{\boldsymbol{C}}[\boldsymbol{K}_h^{(0)}] = \boldsymbol{\Pi}_c\boldsymbol{X}\boldsymbol{X}^T\boldsymbol{\Pi}_c \quad \textit{as } h \to \infty. \tag{9}$$

*Proof.* Let $\boldsymbol{C} = [\boldsymbol{c}_1, \ldots, \boldsymbol{c}_h]$ denote the random projection matrix, where each column $\boldsymbol{c}_j \in \mathbb{R}^d$ is a random vector. Since the entries of $\boldsymbol{C}$ are sampled i.i.d. from $\mathcal{N}(0, 1)$, the columns $\boldsymbol{c}_j$ are independent and identically distributed according to $\boldsymbol{c}_j \sim \mathcal{N}(\boldsymbol{0}, \boldsymbol{I}_d)$.

The stochastic node-covariance operator $\boldsymbol{K}_h^{(0)}$ (Equation (2)) can be rewritten as:

$$
\begin{aligned}
\boldsymbol{K}_h^{(0)} &= \frac{1}{h}(\boldsymbol{\Pi}_c \boldsymbol{X} \boldsymbol{C})(\boldsymbol{\Pi}_c \boldsymbol{X} \boldsymbol{C})^T \\
&= \frac{1}{h}(\boldsymbol{\Pi}_c \boldsymbol{X}[\boldsymbol{c}_1, \ldots, \boldsymbol{c}_h])(\boldsymbol{\Pi}_c \boldsymbol{X}[\boldsymbol{c}_1, \ldots, \boldsymbol{c}_h])^T \\
&= \frac{1}{h}([\boldsymbol{\Pi}_c \boldsymbol{X} \boldsymbol{c}_1, \ldots, \boldsymbol{\Pi}_c \boldsymbol{X} \boldsymbol{c}_h])([\boldsymbol{\Pi}_c \boldsymbol{X} \boldsymbol{c}_1, \ldots, \boldsymbol{\Pi}_c \boldsymbol{X} \boldsymbol{c}_h])^T \\
&= \frac{1}{h} \sum_{j=1}^{h} (\boldsymbol{\Pi}_c \boldsymbol{X} \boldsymbol{c}_j)(\boldsymbol{\Pi}_c \boldsymbol{X} \boldsymbol{c}_j)^T \quad \text{(using block matrix multiplication definition)}
\end{aligned}
$$

Let us define the random matrix $\boldsymbol{Y}_j \in \mathbb{R}^{n \times n}$ as:

$$
\boldsymbol{Y}_j = (\boldsymbol{\Pi}_c \boldsymbol{X} \boldsymbol{c}_j)(\boldsymbol{\Pi}_c \boldsymbol{X} \boldsymbol{c}_j)^T
$$

Since the columns $\boldsymbol{c}_j$ are i.i.d. and $\boldsymbol{Y}_j$ is a fixed function of $\boldsymbol{c}_j$ (given the fixed matrices $\boldsymbol{X}$ and $\boldsymbol{\Pi}_c$), the random matrices $\boldsymbol{Y}_1, \boldsymbol{Y}_2, \ldots, \boldsymbol{Y}_h$ are also independent and identically distributed (i.i.d.).

The operator $\boldsymbol{K}_h^{(0)}$ can thus be written as the sample mean of these i.i.d. random matrices:

$$
\boldsymbol{K}_h^{(0)} = \frac{1}{h} \sum_{j=1}^{h} \boldsymbol{Y}_j
$$

Now, we compute the expected value of $\boldsymbol{Y}_j$. Using the linearity of expectation and the property that $\boldsymbol{\Pi}_c$ and $\boldsymbol{X}$ are constant with respect to the expectation over $\boldsymbol{C}$ (and $\boldsymbol{\Pi}_c = \boldsymbol{\Pi}_c^T$):

$$
\begin{aligned}
\mathbb{E}[\boldsymbol{Y}_j] &= \mathbb{E}[(\boldsymbol{\Pi}_c \boldsymbol{X} \boldsymbol{c}_j)(\boldsymbol{\Pi}_c \boldsymbol{X} \boldsymbol{c}_j)^T] \\
&= \mathbb{E}[\boldsymbol{\Pi}_c \boldsymbol{X} \boldsymbol{c}_j \boldsymbol{c}_j^T \boldsymbol{X}^T \boldsymbol{\Pi}_c^T] \\
&= \boldsymbol{\Pi}_c \boldsymbol{X} \mathbb{E}[\boldsymbol{c}_j \boldsymbol{c}_j^T] \boldsymbol{X}^T \boldsymbol{\Pi}_c
\end{aligned}
$$

Since $\boldsymbol{c}_j \sim \mathcal{N}(\boldsymbol{0}, \boldsymbol{I}_d)$, we know that $\mathbb{E}[\boldsymbol{c}_j \boldsymbol{c}_j^T] = \text{Cov}(\boldsymbol{c}_j) + \mathbb{E}[\boldsymbol{c}_j]\mathbb{E}[\boldsymbol{c}_j]^T = \boldsymbol{I}_d + \boldsymbol{0}\boldsymbol{0}^T = \boldsymbol{I}_d$. Substituting this in:

$$
\mathbb{E}[\boldsymbol{Y}_j] = \boldsymbol{\Pi}_c \boldsymbol{X} \boldsymbol{I}_d \boldsymbol{X}^T \boldsymbol{\Pi}_c = \boldsymbol{\Pi}_c \boldsymbol{X} \boldsymbol{X}^T \boldsymbol{\Pi}_c
$$

Let $\boldsymbol{K}_{\exp} = \boldsymbol{\Pi}_c \boldsymbol{X} \boldsymbol{X}^T \boldsymbol{\Pi}_c$. We have shown that $\mathbb{E}[\boldsymbol{Y}_j] = \boldsymbol{K}_{\exp}$. Since $\boldsymbol{X}$ is a fixed finite matrix, and the moments of Gaussian variables are finite, the expectation $\mathbb{E}[\boldsymbol{Y}_j]$ exists and is finite.

We have $\boldsymbol{K}_h^{(0)}$ as the sample mean of $h$ i.i.d. random matrices $\boldsymbol{Y}_j$, each with finite expectation $\boldsymbol{K}_{\exp}$. By the Weak Law of Large Numbers, applicable to sums of i.i.d. random vectors or matrices (considering convergence element-wise or in matrix norm), the sample mean converges in probability to the expected value as the number of samples $h$ goes to infinity. Therefore, for each entry $(a, b)$ of the matrices:

$$
(\boldsymbol{K}_h^{(0)})_{ab} = \frac{1}{h} \sum_{j=1}^{h} (\boldsymbol{Y}_j)_{ab} \xrightarrow{p} \mathbb{E}[(\boldsymbol{Y}_j)_{ab}] = (\boldsymbol{K}_{\exp})_{ab} \quad \text{as } h \to \infty
$$

This element-wise convergence implies convergence in probability for the matrix:

$$
\boldsymbol{K}_h^{(0)} \xrightarrow{p} \boldsymbol{K}_{\exp} = \boldsymbol{\Pi}_c \boldsymbol{X} \boldsymbol{X}^T \boldsymbol{\Pi}_c \quad \text{as } h \to \infty.
$$

This completes the proof. $\qquad \square$

## C   ADDITIONAL RESULTS

### C.1   CATEGORIZATION AND DESCRIPTION OF BASELINES

Table 2 compares our approach against diverse families of baselines evaluated on node classification benchmarks. We group methods into four primary categories: (i) SUPERVISED GNNS that are trained

Table 4: Performance of ALL-IN on pre-training source datasets compared to specialized supervised baselines trained individually per dataset (including our ALL-IN-SPECIALIZED which is trained separately on each individual dataset). ALL-IN maintains highly competitive performance.

| Method | ZINC (MAE ↓) | MOLHIV (ROC-AUC ↑) | MOLESOL (RMSE ↓) | MOLTOX21 (ROC-AUC ↑) | MNIST (ACC ↑) | CIFAR10 (ACC ↑) | MODELNET (ACC ↑) | CUNEIFORM (ACC ↑) | MSRC 21 (ACC ↑) |
|---|---|---|---|---|---|---|---|---|---|
| **TRAINED PER DATASET** | | | | | | | | | |
| GCN (Kipf & Welling, 2017) | 0.3674 | 76.06 | 1.11 | 75.29 | 90.120 | 54.142 | 17.18 | 45.67 | 89.53 |
| GAT (Veličković et al., 2018) | 0.3842 | 76.00 | 1.05 | 75.21 | 95.535 | 64.223 | 65.20 | 78.60 | 82.10 |
| GIN (Xu et al., 2019) | 0.1630 | 75.58 | 1.17 | 74.91 | 96.485 | 55.255 | 73.13 | 79.05 | 86.31 |
| ALL-IN-SPECIALIZED (0 props) | 0.1480 | 72.65 | 1.22 | 69.37 | 94.03 | 39.96 | 37.24 | 85.19 | 91.65 |
| ALL-IN-SPECIALIZED | 0.1195 | 73.78 | 1.19 | 70.04 | 94.77 | 40.03 | 39.81 | 87.20 | 94.16 |
| **TRAINED ON ALL DATASETS** | | | | | | | | | |
| ALL-IN (0 props) | 0.1557 | 72.74 | 1.28 | 68.19 | 94.57 | 40.11 | 37.11 | 89.88 | 97.51 |
| ALL-IN | 0.1237 | 74.49 | 1.29 | 68.20 | 95.22 | 40.08 | 39.37 | 91.17 | 98.08 |

from scratch on each dataset, (ii) LLM-AUGMENTED GNNs where the node features are enhanced using language models, (iii) LLM-BASED REASONING that converts the graph into a compatible input to pre-trained LLMs, and (iv) GNN-BASED methods.

**SUPERVISED BASELINES** include (a) MLP: a multi-layer perceptron directly on the target dataset features without using graph structure; serves as a non-graph baseline. (b) GCN (Kipf & Welling, 2017): trained from scratch on the target dataset (c) GIN (Xu et al., 2019) trained from scratch, included to represent expressive message-passing GNNs in supervised settings. These fall under supervised baselines as they do not perform pretraining or transfer, and rely solely on training from scratch on each dataset.

**LLM-AUGMENTED GNNs** include (a) OFA (Liu et al., 2024): constructs a prompt-augmented graph using text nodes and pretrains an RGCN to enable in-context transfer across node/link/graph tasks; falls here for fusing text prompts with GNN structure and relying on LLM embeddings. (b) GLEM-LM (Chen et al., 2024b): Enhances GNNs using sentence-level text embeddings from a frozen LLM; categorized here due to its augmentation of GNN input via LLM-derived features. These are classified as LLM-Augmented GNNs since they incorporate LLMs to enrich graph inputs or guide GNN training, but retain a GNN backbone.

**LLM-BASED** methods include (a) GRAPHTEXT (Zhao et al., 2023) that transforms $k$-hop neighborhoods into textual prompts and performs zero/few-shot classification using frozen LLMs and (b) RWNN (Kim et al., 2024) that converts random walks on graphs to node label anonymized sequences and uses frozen LLMs for prediction. belong to this category due to their reliance on prompt-based inference using LLMs without any GNNs.

**GNN-BASED** methods include (a) ANYGRAPH (Xia & Huang, 2024) that pretrains a graph mixture-of-experts model using link prediction objective on diverse graphs that allows transfer to unseen datasets, (b) GRAPHANY (Zhao et al., 2024b) that learns permutation-invariant attention over a bank of pretrained LinearGNNs; (c) MDGPT (Yu et al., 2024) pretrains a GCN on multiple datasets with SVD-projected features and prompt vectors; (d) GCOPE (Zhao et al., 2024a) constructs a universal pretraining graph with virtual nodes and uses contrastive learning to train a shared GNN; (e) GPPT (Sun et al., 2022) introduces task-specific graph prompts for node task and link-prediction alignment; (f) GPROMPT (Gong et al., 2024) utilizes prompt vectors into graph pooling via element–wise multiplication (g) ALL-IN-ONE (Sun et al., 2023) combines token graphs with original graph as prompts (h) GPF (Fang et al., 2023) introduces prompt tokens and GPF-PLUS trains multiple independent basis vectors and combines them using attention (i) ULTRA (Galkin et al., 2024) learns transferable graph representations by conditioning on relational interactions. (j) SCORE (Wang & Luo, 2024) introduces zero-shot reasoning on knowledge graphs using graph topology. All of these are grouped under GNN-BASED baselines as they rely on pretraining GNNs (often with auxiliary components like prompts or experts) to enable generalization to new graphs.

## C.2 COMPARISON TO METHODS TRAINED ON EACH INDIVIDUAL DATASET

In this section, we compare the performance of ALL-IN to that of standard supervised GNN baselines (GCN (Kipf & Welling, 2017), GAT (Veličković et al., 2018), GIN (Xu et al., 2019)), trained individually for each dataset, using their original, dataset-specific input features. Contrary to ALL-IN,

which is trained jointly on all datasets, these supervised baselines are thus specialized for each respective dataset.

When evaluating on the pre-training datasets, it is generally expected that supervised models trained on each individual dataset would achieve strong, if not optimal, performance, particularly as each dataset provides sufficient data for dedicated task-specific learning. The goal for ALL-IN here is therefore to show that its jointly pre-trained shared encoder can support task-specific heads that remain competitive against individually trained models, indicating its ability to learn general-purpose representations without substantial performance degradation on each task.

Table 4 summarizes the performance of ALL-IN (with and without propagated covariance operators, obtained by setting $k = 0$ in Equation (4), denoted as ALL-IN and ALL-IN (0 props) respectively) against the specialized version of our model (ALL-IN-SPECIALIZED and ALL-IN-SPECIALIZED (0 props)), as well as specialized supervised baselines on the pre-training datasets. Our findings indicate that, while specialized baselines maintain an edge on certain datasets (e.g., CIFAR10 and MODELNET), ALL-IN is broadly competitive. For instance, on the ZINC regression task, ALL-IN achieves a MAE of 0.1237, surpassing all listed specialized baselines, including GIN (0.1630). Similarly, ALL-IN demonstrates higher accuracy on CUNEIFORM (91.17% vs. GIN 79.05%) and MSRC 21 (98.08% vs. GIN 86.31%). Finally, we highlight the general advantage of the full ALL-IN (which utilizes propagated operators) over ALL-IN (0 props), and similarly of ALL-IN-SPECIALIZED over ALL-IN-SPECIALIZED (0 props), suggesting that the richer relational information from propagated operators contributes to more effective representation learning during this phase.

Overall, these results indicate that a single, pre-trained ALL-IN encoder can maintain strong, often competitive, performance across a diverse set of source datasets and tasks.

### C.3   USING SVM ON THE PRE-TRAINED REPRESENTATIONS

To assess the linear separability and structural quality of the learned graph representations from ALL-IN, we evaluate downstream graph classification accuracy using support vector machines (SVMs) with both linear and radial basis function (RBF) kernels (Table 5). This setup allows us to probe how well the learned representations support simple (linear) versus more expressive (nonlinear) decision boundaries.

We compare against several non-learnable baselines that do not involve any representation learning:

    (a) Input Features ($X$): Raw input features of each graph, computed by averaging node features.

    (b) Propagated Input Features ($AX$): Features after one round of neighborhood propagation, capturing local graph structure.

    (c) Input Features along with random walk structural encodings ($X \oplus S$): Concatenates the raw features with random walk structural encoding (RWSE) (Dwivedi et al., 2022a), which encodes graph structure based on transition probabilities of random walks.

These baselines serve as direct input replacements for ALL-IN and are shared across both kernel settings. They provide a strong reference for understanding the inherent structure in the input space, independent of any learning or pretraining.

For ALL-IN, we report results both with and without concatenation of the input features to assess the added value of structural information in the learned embeddings.

Under the RBF kernel, ALL-IN combined with input features achieves the best performance on four out of six datasets, including PTC, NCI1, NCI109, and ENZYMES, highlighting its ability to encode discriminative patterns suitable for nonlinear classification. In contrast, performance under the linear kernel is more mixed, with RWSE showing strong results on datasets like PROTEINS, indicating some inherent linear separability in the structural baseline. Overall, these results demonstrate that ALL-IN learns representations that are expressive and transferable across diverse graph datasets, especially when paired with nonlinear classifiers.

Table 5: Graph classification accuracy (%) using SVMs with Linear and RBF kernels. Baselines are shared across both kernels. Results are reported as mean $\pm$ standard deviation over 10 runs.

| Method | MUTAG (ACC ↑) | PTC (ACC ↑) | PROTEINS (ACC ↑) | NCI1 (ACC ↑) | NCI109 (ACC ↑) | ENZYMES (ACC ↑) |
|---|---|---|---|---|---|---|
| **LINEAR SVM** | | | | | | |
| Input Features | 81.87 ± 7.25 | 60.88 ± 1.83 | **72.68 ± 0.58** | 64.59 ± 1.24 | 63.36 ± 2.22 | 22.00 ± 4.46 |
| Propagated Input Features | 69.64 ±14.21 | 57.34 ±10.89 | 59.56 ± 3.94 | 64.16 ± 1.22 | 63.26 ± 1.63 | 14.33 ± 5.01 |
| Input Features + RWSE | 80.96 ± 0.89 | 60.14 ± 1.15 | 65.74 ± 0.43 | 64.30 ± 0.16 | 63.45 ± 0.20 | 27.00 ± 4.63 |
| ALL-IN | 74.47 ± 7.70 | 53.12 ± 9.09 | 60.91 ± 4.25 | 63.26 ± 1.36 | 63.19 ± 1.89 | 21.16 ± 6.28 |
| ALL-IN + Input Features | 74.47 ± 7.70 | 52.84 ± 9.03 | 62.00 ± 4.29 | 64.45 ± 1.48 | 63.72 ± 1.67 | 21.50 ± 5.18 |
| **RBF SVM** | | | | | | |
| Input Features | 72.73 ±14.29 | 55.88 ±11.58 | 71.06 ± 2.93 | 66.44 ± 1.43 | 66.80 ± 1.35 | 33.33 ± 4.77 |
| Propagated Input Features | 79.70 ±11.03 | 54.10 ±10.25 | 72.05 ± 4.70 | 55.66 ± 5.80 | 58.05 ± 5.42 | 33.16 ± 4.43 |
| Input Features + RWSE | 79.21 ±10.99 | 58.71 ± 8.76 | 67.21 ± 6.22 | 70.68 ± 2.60 | 67.82 ± 2.79 | 36.66 ± 5.96 |
| ALL-IN | 82.98 ± 7.76 | 59.28 ± 9.13 | 70.62 ± 4.53 | 65.88 ± 1.62 | 65.68 ± 1.90 | 28.83 ± 5.87 |
| ALL-IN + Input Features | **84.06 ± 6.61** | **59.88 ± 7.72** | 71.42 ± 4.29 | **67.54 ± 1.33** | **67.34 ± 1.51** | **32.16 ± 6.71** |

## C.4 ADDITIONAL RESULTS ON TRANSFERABILITY TO UNSEEN DATASETS

In Table 6, we present comparison with more baselines on our graph classification datasets MUTAG and PROTEINS. We describe below the changes we make to the following baselines to make them applicable to this setting:

- **GLEM-LM** (Chen et al., 2024b): This is a method that only supports tasks on text-attributed graphs. Since the TU Datasets (Morris et al., 2020) do not have node text attributes, we describe the input node features and pass them to ChatGPT.

- **GCOPE** (Zhao et al., 2024a): This method introduces one virtual node for each node classification dataset, connecting it to all the nodes within the dataset. To perform graph classification, we introduce one virtual node for each graph classification dataset and connect it to all the nodes in all the graphs within the dataset.

- **ANYGRAPH** (Xia & Huang, 2024): This method performs node classification by adding one node per class and connecting each training node to its corresponding class node. Classification of unlabeled nodes is performed by computing the dot product between the node's embedding and each class node embedding to rank the classes. To extend this paradigm to graph classification, we introduce a virtual node that connects to all nodes in the graph and add one class node per category. For classifying new graphs, we compute the dot product between the virtual node embedding and each class node embedding to rank the classes.

We leave out the following methods and provide justification below:

- **GRAPHTEXT** (Zhao et al., 2023): While the authors mention that GRAPHTEXT is applicable for graph classification, they do not provide a way to construct a graph syntax tree for an entire graph, which can be ambiguous as it could involve introducing a virtual node or averaging results from syntax trees of multiple nodes.

- **GRAPHANY** (Zhao et al., 2024b): This method is explicitly only designed for node classification on arbitrary graphs, as it relies on an analytical solution that is not directly applicable to graph-level tasks.

The results in Table 6 further substantiate ALL-IN's strong performance. These findings reinforce the observations made in the main paper (Table 3): ALL-IN, with its frozen pre-trained encoder and a retrained head, effectively generalizes to new graph classification datasets with novel input features, surpassing a wide variety of adapted baselines.

## C.5 THE IMPORTANCE OF SPES AND RANDOM PROJECTIONS IN EQUATION (5)

In this section, we conduct an ablation study to investigate the importance of SPEs and random projections within ALL-IN. We compare our ALL-IN with several additional models having the same backbone, loss, and training datasets, namely:

Table 6: Performance on unseen graph-classification datasets with new input features. ALL-IN demonstrates strong transferability, underscoring its versatility and ability to handle different feature spaces. [†] indicates these methods were modified to work on these datasets, as explained in Appendix C.4

| Dataset | MUTAG (ACC ↑) | PROTEINS (ACC ↑) |
|---|---|---|
| **SUPERVISED BASELINES** | | |
| MLP | 67.20 ± 1.00 | 59.20 ± 1.00 |
| GIN (Xu et al., 2019) | 89.40 ± 5.60 | 76.20 ± 2.80 |
| **LLM-AUGMENTED GNNs** | | |
| OFA (Liu et al., 2024) | 61.04 ± 4.71 | 61.40 ± 2.99 |
| GLEM-LM[†] (Chen et al., 2024b) | 72.97 ± 0.00 | 43.22 ±12.01 |
| **LLM-BASED** | | |
| RWNN-DEBERTA (Kim et al., 2024) | 58.22 ± 0.24 | 67.85 ± 0.53 |
| **GNN-BASED** | | |
| GCOPE[†] (Zhao et al., 2024a) | 81.87 ± 7.26 | 71.84 ± 3.48 |
| ANYGRAPH[†] (Xia & Huang, 2024) | 75.61 ± 6.94 | 72.23 ± 4.63 |
| MDGPT (Yu et al., 2024) | 57.36 ±14.26 | 54.35 ±10.26 |
| GPPT (Sun et al., 2022) | 60.40 ±15.43 | 60.92 ±12.47 |
| ALL-IN-ONE (Sun et al., 2023) | 79.87 ± 5.34 | 66.49 ± 6.26 |
| GPROMPT (Gong et al., 2024) | 73.60 ± 4.76 | 59.17 ±11.26 |
| GPF (Fang et al., 2023) | 68.40 ± 5.09 | 63.91 ± 3.26 |
| GPF-PLUS (Fang et al., 2023) | 65.20 ± 6.94 | 62.92 ± 2.78 |
| ULTRA(3G) (Galkin et al., 2024) | 63.33 ± 0.00 | 58.09 ± 0.00 |
| SCORE (Wang & Luo, 2024) | 85.33 ± 2.11 | 68.54 ± 1.47 |
| ALL-IN (0 props) | 92.50 ± 6.60 | 76.72 ± 3.19 |
| ALL-IN | 92.90 ± 6.34 | 78.20 ± 3.81 |

Table 7: The impact of SPEs and random projections in Equation (5). ALL-IN with SPEs performs best, while using only SPEs leads to a significant drop in performance, highlighting the importance of random feature projections, which cannot be compensated by using SVD.

| Method | ZINC (MAE ↓) | MOLESOL (RMSE ↓) | MOLHIV (ROC-AUC ↑) | MOLTOX21 (ROC-AUC ↑) | MNIST (ACC ↑) | CIFAR10 (ACC ↑) | MODELNET (ACC ↑) | CUNEIFORM (ACC ↑) | MSRC 21 (ACC ↑) |
|---|---|---|---|---|---|---|---|---|---|
| ALL-IN (SVD) | 0.1445 | 1.43 | 71.82 | 65.55 | 92.97 | 37.12 | 36.51 | 87.28 | 95.84 |
| SPEs-only | 0.1396 | 1.45 | 71.95 | 64.10 | 91.01 | 35.22 | 30.65 | 85.89 | 95.13 |
| ALL-IN (SVD + SPEs) | 0.1318 | 1.41 | 72.06 | 66.13 | 93.40 | 37.74 | 36.95 | 88.56 | 96.91 |
| ALL-IN (no SPEs) | 0.1251 | 1.31 | 74.02 | 67.62 | 94.88 | 39.45 | 38.72 | 90.61 | 97.93 |
| ALL-IN | 0.1237 | 1.29 | 74.49 | 68.20 | 95.22 | 40.08 | 39.37 | 91.17 | 98.08 |

- ALL-IN (SVD), where we replace Equation (5) with $\mathbf{H}^{(0)} = \text{SVD}(\mathbf{X}^{(0)})$, thus removing both random projections and SPEs, and replacing them with SVD of the input features;

- SPEs-only variant, where we replace Equation (5) with $\mathbf{H}^{(0)} = \mathbf{S}$, while keeping the same covariance operator set and head, therefore removing $\mathbf{R}^{(0)}$ only from Equation (5), but still using $\mathbf{R}^{(0)}$ to define the covariance operators.

- ALL-IN (SVD + SPEs), where we replace Equation (5) with $\mathbf{H}^{(0)} = \text{SVD}(\mathbf{X}^{(0)}) \oplus \mathbf{S}$, thus removing random projections and replacing them with SVD of the input features (while keeping SPEs);

- ALL-IN (no SPEs), where we replace Equation (5) with $\mathbf{H}^{(0)} = \mathbf{R}^{(0)}$, thus removing SPEs;

The results in Table 7 support our claim: the full ALL-IN with SPEs performs best, but only slightly better than the version without SPEs. In contrast, using only SPEs leads to a significant drop in performance, highlighting the importance of random feature projections, which provides improved performance also when compared with SVD.

## C.6 THE IMPORTANCE OF RANDOM PROJECTIONS

In this section, we demonstrate the impact of random projections by comparing ALL-IN with the baseline obtained by removing random projections from Equation (5) (thus, setting $\mathbf{H}^{(0)} = \mathbf{S}$) and from Equation (2), thus replacing NodeCov($\mathbf{X}\mathbf{C}$) with NodeCov($\mathbf{X}$).

Table 8: The impact of using random projections within ALL-IN, obtained by comparing ALL-IN to its counterpart that has no random projections in either Equation (5) or Equation (2).

| Method | ZINC (MAE ↓) | MOLESOL (RMSE ↓) | MOLHIV (ROC-AUC ↑) | MOLTOX21 (ROC-AUC ↑) | MNIST (ACC ↑) | CIFAR10 (ACC ↑) | MODELNET (ACC ↑) | CUNEIFORM (ACC ↑) | MSRC 21 (ACC ↑) |
|---|---|---|---|---|---|---|---|---|---|
| ALL-IN (no random) | 0.1475 | 1.51 | 71.40 | 62.85 | 91.10 | 35.42 | 33.91 | 85.47 | 95.02 |
| ALL-IN | 0.1237 | 1.29 | 74.49 | 68.20 | 95.22 | 40.08 | 39.37 | 91.17 | 98.08 |

Table 9: The impact of the operators in the operator set (Equation (4)). Results improve when considering covariance operators compared to graph (adjacency) only, highlighting their importance in ALL-IN.

| Method | ZINC (MAE ↓) | MOLESOL (RMSE ↓) | MOLHIV (ROC-AUC ↑) | MOLTOX21 (ROC-AUC ↑) | MNIST (ACC ↑) | CIFAR10 (ACC ↑) | MODELNET (ACC ↑) | CUNEIFORM (ACC ↑) | MSRC 21 (ACC ↑) |
|---|---|---|---|---|---|---|---|---|---|
| Identity Only | 0.1535 | 1.65 | 67.10 | 60.33 | 86.22 | 29.34 | 25.15 | 81.49 | 91.78 |
| Adjacency Only | 0.1378 | 1.46 | 71.75 | 65.17 | 92.78 | 35.22 | 31.40 | 87.25 | 95.10 |
| Covariance Only | 0.1282 | 1.34 | 73.80 | 67.93 | 94.30 | 38.50 | 36.85 | 89.44 | 97.13 |
| ALL-IN | 0.1237 | 1.29 | 74.49 | 68.20 | 95.22 | 40.08 | 39.37 | 91.17 | 98.08 |

The results in Table 8 suggest that random projection is critical to bridge input feature spaces. This aligns with our results in Theorem 4.3, which demonstrates the theoretical benefit of using random projections in the covariance operators.

## C.7 ABLATION STUDY ON THE OPERATOR SET

In this section, we perform an ablation study isolating the contribution of different operators in ALL-IN. Table 9 reports the performance of ALL-IN (which uses the operators defined in Equation (4)) and compares it with Identity Only, obtained by setting $\mathcal{O} = \{\boldsymbol{I}\}$ in Equation (4), Adjacency Only, obtained by setting $\mathcal{O} = \{\boldsymbol{A}\}$ in Equation (4), and Covariance Only, obtained by setting $\mathcal{O} = \{\boldsymbol{K}^{(0)}\}$ in Equation (4).

Covariance operators enable the neural network to learn shared characteristics in input feature spaces and graph structures, as results improve when considering covariance operators compared to graph (adjacency) only operators.

## C.8 THE ROLE OF THE FEATURE DIMENSIONALITY $h$

We next evaluate the performance of ALL-IN when varying the hidden dimension $h$. Results are reported in Table 10.

Across datasets, performance improves from very small $h$ and then plateaus at 256, and gains beyond that are marginal. This trend aligns with Proposition 4.6: as $h$ grows, the stochastic operator concentrates around its expectation. In practice, a moderate $h$ achieves near-saturated accuracy with a better compute/memory trade-off than a very large $h$. Therefore, model performance stabilizes at moderate $h$ and larger $h$ primarily improves stability, matching the proposition's claim.

## C.9 ADDITIONAL DATASETS

We further evaluate ALL-IN on the larger node-level dataset ogbn-arxiv (Hu et al., 2020a) (169,343 nodes, 1,166,243 edges), on heterophilic benchmarks Actor, Chameleon, Squirrel using the splits in Pei et al. (2020), and on the AmzRating, Minesweep, Tolokers datasets (Platonov et al., 2023). All results, which are reported in Tables 11 to 13, respectively, show that ALL-IN offers consistently better performance. We also investigate the behavior of our covariance operators on heterophilous graphs. Intuitively, the node-covariance matrix computed from the projected input features captures feature similarity across all node pairs, not just along edges. For heterophilous graphs, the base covariance operator $\boldsymbol{K}^{(0)}$ can therefore highlight similarities between non-adjacent nodes in the original input graph or dissimilarities between adjacent nodes, which can help GNNs with heterophily. In addition, the propagated operators $\boldsymbol{K}^{(p)}$ for $p > 0$ further help in this setting, because their availability to the GNN allows it to view and mix information from multiple neighborhoods, in line

Table 10: Performance of ALL-IN with varying hidden dimension $h$.

| Method | ZINC (MAE ↓) | MOLESOL (RMSE ↓) | MOLHIV (ROC-AUC ↑) | MOLTOX21 (ROC-AUC ↑) | MNIST (ACC ↑) | CIFAR10 (ACC ↑) | MODELNET (ACC ↑) | CUNEIFORM (ACC ↑) | MSRC 21 (ACC ↑) |
|--------|------|---------|--------|----------|-------|---------|----------|-----------|--------|
| ALL-IN ($h = 64$) | 0.1316 | 1.43 | 72.20 | 65.75 | 92.14 | 36.15 | 34.26 | 87.89 | 96.12 |
| ALL-IN ($h = 128$) | 0.1264 | 1.34 | 73.46 | 67.91 | 94.41 | 38.92 | 37.88 | 90.21 | 97.56 |
| ALL-IN ($h = 256$) | 0.1239 | 1.30 | 74.38 | 68.14 | 95.03 | 39.85 | 39.19 | 91.05 | 98.01 |
| ALL-IN ($h = 512$) | 0.1237 | 1.29 | 74.49 | 68.20 | 95.22 | 40.08 | 39.37 | 91.17 | 98.08 |
| ALL-IN ($h = 1024$) | 0.1236 | 1.28 | 74.58 | 68.18 | 95.24 | 40.11 | 39.40 | 91.22 | 98.10 |

Table 11: Performance on the ogbn-arxiv (Hu et al., 2020a).

| Method | ogbn-arxiv (↑) |
|--------|--------|
| **NON-PARAMETRIC BASELINES** | |
| LABEL PROPAGATION (Zhu & Ghahramani, 2002) | 61.04 |
| **SUPERVISED BASELINES** | |
| GCN (Kipf & Welling, 2017) | 71.74 |
| GAT (Veličković et al., 2018) | 71.95 |
| GraphGPS (Rampášek et al., 2022) | 70.97 |
| **LLM-AUGMENTED GNNS** | |
| OFA (Liu et al., 2024) | 73.22 |
| **LLM-BASED** | |
| GraphText (Zhao et al., 2023) | 49.47 |
| **GNN-BASED** | |
| AnyGraph (Xia & Huang, 2024) | 62.33 |
| GraphAny (Zhao et al., 2024b) | 58.38 |
| ALL-IN | 75.27 |

with understandings from literature on heterophily in graphs (Zhu et al., 2020; Chien et al., 2021). Motivated by this discussion, we conduct an ablation study where we vary the number of propagation orders $k \in \{0, 1, 2\}$ used in the covariance operators and evaluate downstream performance on Actor, Chameleon, and Squirrel. As reported in Table 14, adding propagated operators consistently improves performance.

## C.10 FIXED RANDOM PROJECTIONS

In the main experiments, the projection matrix $C$ is sampled at each forward pass, which yields the distributional invariance guarantees in Section 4. To isolate the empirical effect of this stochasticity, we consider a variant where $C$ is sampled once and kept fixed for all subsequent training and inference steps denoted *ALL-IN (Fixed C)*. We keep all other settings, including the backbone and training budget, identical. Table 15 reports performance on the pre-training source datasets, and Table 16 reports transfer results on representative downstream tasks. Across all pre-training datasets in Table 15, fixing $C$ leads to a consistent but moderate degradation compared to the stochastic variant. A similar pattern holds on the downstream tasks in Table 16, where ALL-IN (Fixed $C$) underperforms the stochastic version on both node and graph classification. These results empirically support the beneficial role of stochastic projections in our framework, while showing that the model remains competitive also when the projection matrix is fixed.

## C.11 EDGE FEATURES ABLATION

For datasets with edge features such as ZINC, we follow the strategy described in Section 3, where edge features are first randomly projected, then aggregated to nodes, and used to construct additional node-covariance operators that are added to the operator set. Concretely, the aggregated edge features are converted into an $n \times n$ edge covariance operator $\boldsymbol{K}_{\text{edge}}$, whose entries compare the aggregated edge-feature environments of all node pairs, and, the backbone GNN uses the projected edge features. To quantify the empirical contribution of this design, we perform an ablation on ZINC that compares: (i) a standard GIN without edge features, (ii) GINE (GIN with edge features), (iii) ALL-IN with edge features removed (ALL-IN (no edge features)), and (iv) the full ALL-IN using the edge-derived covariance operator as described above. Results are reported in Table 17. From Table 17, we observe that including the edge-based covariance operator yields substantially better performance

Table 12: Performance on heterophilic datasets, using the splits in Pei et al. (2020).

| Method | Actor (ACC ↑) | Chameleon (ACC ↑) | Squirrel (ACC ↑) |
|---|---|---|---|
| **NON-PARAMETRIC BASELINES** | | | |
| LABEL PROPAGATION (Zhu & Ghahramani, 2002) | $18.83 \pm 0.00$ | $40.89 \pm 0.00$ | $33.42 \pm 0.00$ |
| **SUPERVISED BASELINES** | | | |
| GCN (Kipf & Welling, 2017) | $28.55 \pm 0.68$ | $64.69 \pm 2.21$ | $47.07 \pm 0.71$ |
| **GNN-BASED** | | | |
| GraphAny (Zhao et al., 2024b) | $28.60 \pm 0.21$ | $62.59 \pm 0.86$ | $49.70 \pm 0.95$ |
| ULTRA (Galkin et al., 2024) | $22.61 \pm 0.00$ | N/A | N/A |
| SCORE (Wang & Luo, 2024) | $23.26 \pm 0.56$ | N/A | N/A |
| ALL-IN | $29.47 \pm 0.38$ | $67.40 \pm 1.29$ | $49.98 \pm 0.73$ |

Table 13: Performance on the AmzRating, Minesweep, Tolokers datasets (Platonov et al., 2023).

| Method | AmzRatings (ACC ↑) | Minesweeper (ACC ↑) | Tolokers (ACC ↑) |
|---|---|---|---|
| GCN (Kipf & Welling, 2017) | $47.35 \pm 0.26$ | $81.12 \pm 0.37$ | $79.93 \pm 0.10$ |
| GraphAny (Zhao et al., 2024b) | $42.84 \pm 0.04$ | $80.46 \pm 0.15$ | $78.24 \pm 0.03$ |
| ALL-IN | $49.02 \pm 0.11$ | $82.93 \pm 0.26$ | $81.43 \pm 0.07$ |

than omitting edge features entirely, and that ALL-IN with edge features not only recovers but surpasses the behavior of an edge-aware GNN such as GINE. In contrast, removing edge features in ALL-IN leads to performance closer to a standard GIN, consistent with observations from the supervised GNN literature. This ablation indicates that the aggregation scheme in Section 3 retains and effectively utilizes edge information.

## C.12 PRE-TRAINING WITH CITATION NETWORKS

In the main experiments, citation networks are excluded from the pre-training corpus to act as out-of-distribution targets with very high-dimensional, sparse features and large graph sizes. We now show that ALL-IN can also benefit from citation networks during pre-training, and consider an extended setting where Cora and CiteSeer are added to the pre-training mix. We keep the architecture and training budget fixed, and compare (i) the original pre-training corpus (no citation networks) and (ii) the extended corpus (original + Cora + CiteSeer). Table 18 reports pre-training performance on all source datasets, including Cora and CiteSeer for the extended setting. The results show that adding citation networks leaves performance on the original pre-training corpus stable, further indicating the ability of ALL-IN in learning from multiple sources acting as an input feature space bridge. Table 19 reports downstream performance on OGBN-ARXIV, MUTAG, and PROTEINS for both pre-training regimes, indicating that including citation networks in pre-training maintains or improves downstream performance.

## C.13 TRANSFER TO ADDITIONAL DOMAINS

To further evaluate the generality of ALL-IN beyond citation and bioinformatics datasets, we consider two downstream tasks from distinct domains: (i) 3D shape segmentation on ShapeNet, and (ii) social-network classification on IMDB-BINARY (IMDB-B). For ShapeNet, we use knn graphs over point clouds as is standard with this dataset (Wang et al., 2019) and report mean Intersection-over-Union (mIoU); for IMDB-B, we report classification accuracy. In both cases, we use the same ALL-IN encoder as in the main experiments and compare with two GNN baselines (GIN and GPS). As shown in Table 20, ALL-IN consistently outperforms the GIN and GPS baselines on both ShapeNet and IMDB-B. This indicates that the input-space bridge from ALL-IN yields representations that are beneficial also in 3D shape graphs and social networks, further highlighting its effectiveness.

## C.14 DOWNSTREAM REGRESSION TRANSFER

Our pre-training stage for ALL-IN uses a supervised multi-task objective over several graph-level datasets, including both graph classification and graph regression. This design choice reflects our

Table 14: Effect of the number of propagation orders $k$ on heterophilous benchmarks.

| Number of propagation orders $k$ | Actor (ACC ↑) | Chameleon (ACC ↑) | Squirrel (ACC ↑) |
|---|---|---|---|
| 0 | $28.62 \pm 0.45$ | $65.12 \pm 1.44$ | $47.89 \pm 0.80$ |
| 1 | $29.00 \pm 0.42$ | $66.37 \pm 1.35$ | $49.14 \pm 0.76$ |
| 2 | $29.47 \pm 0.38$ | $67.40 \pm 1.29$ | $49.98 \pm 0.73$ |

Table 15: Effect of fixing the projection matrix $C$ during pre-training.

| Method | ZINC (MAE ↓) | MOLESOL (RMSE ↓) | MOLHIV (ROC-AUC ↑) | MOLTOX21 (ROC-AUC ↑) | MNIST (ACC ↑) | CIFAR10 (ACC ↑) | MODELNET (ACC ↑) | CUNEIFORM (ACC ↑) | MSRC21 (ACC ↑) |
|---|---|---|---|---|---|---|---|---|---|
| ALL-IN (Fixed $C$) | 0.1369 | 1.38 | 74.12 | 66.72 | 93.97 | 39.84 | 39.02 | 90.11 | 96.27 |
| ALL-IN (stochastic $C$) | 0.1237 | 1.29 | 74.49 | 68.20 | 95.22 | 40.08 | 39.37 | 91.17 | 98.08 |

goal of learning a single encoder that learns across diverse graph modalities and objectives. The motivation for including regression tasks such as ZINC in the pre-training mix is inspired by the broader multi-task and foundation-model literature: training a shared encoder on a diverse collection of tasks and objectives is widely used to encourage more general-purpose representations (Zhang & Yang, 2021; Raffel et al., 2020). To directly demonstrate regression-style transfer, we additionally evaluate ALL-IN on a held-out graph-level regression benchmark not used during pre-training, PEPTIDES-STRUCT from LRGB (Dwivedi et al., 2022b). We compare GNN baselines (GINE and a GPS) with ALL-IN. As shown in Table 21, ALL-IN achieves the lowest mean absolute error on PEPTIDES-STRUCT, demonstrating the effectiveness of ALL-IN also in a regression downstream task.

### C.15 SUPERVISED VS. UNSUPERVISED PRE-TRAINING OF ALL-IN

Our main experiments adopt a supervised multi-task pre-training objective for ALL-IN, combining graph-level classification (e.g., OGBG-MOLHIV, MODELNET) and regression tasks (e.g., ZINC). This design leverages the availability of labels on diverse source datasets to learn input-space agnostic representations that are directly aligned with downstream prediction objectives. Prior work on graph representation learning has shown that, when labels are available, supervised pre-training can yield stronger and more task-discriminative representations than purely self-supervised approaches (Hu et al., 2020b), and similar observations hold in large-scale vision studies (He et al., 2022).

To provide an empirical comparison between supervised and unsupervised pre-training on top of our input-space bridge, we construct an unsupervised variant in which we replace all supervised losses on the pre-training datasets with a masked-feature reconstruction objective of masked graph autoencoders (Hou et al., 2022). Concretely, as in Hou et al. (2022) we randomly mask node features and train ALL-IN to reconstruct the original feature values from the node embeddings. The encoder architecture and training budget are kept identical to the supervised setting. Then, we benchmark the downstream performance on Cora and MUTAG. As shown in Table 22, both pre-training approaches achieve similar downstream performance, where the supervised variant slightly outperforms the unsupervised. This is consistent with prior observations that supervised objectives can provide particularly strong graph representations when labels are available, and it supports our choice to adopt supervised multi-task pre-training for ALL-IN in the setting considered in this work.

### C.16 EFFECT OF THE NUMBER OF PROPAGATION ORDERS

ALL-IN constructs node-covariance operators not only on the original projected features, but also on features that have been propagated through the graph up to $k$ times, as discussed in Section 3. Intuitively, increasing the number of propagation orders $k$ allows the covariance operators to incorporate multi-hop information coupled with the input features, at the cost of additional computations and operators. In the main experiments we set $k = 0, 2$ as a default choice. Here, we provide an extended ablation over $k \in \{0, 1, 2, 3, 4\}$. In this study we vary the number of propagation orders $k$, while keeping all other hyperparameters and training settings unchanged. The case $k = 0$ uses only the input features node-covariance operators, whereas larger $k$ progressively add operators built from 1-hop, 2-hop, and higher-order propagated features. Table 23 reports the pre-training performance of ALL-IN across all source datasets for different values of $k$. As can be seen, moving from $k = 0$ to

Table 16: Effect of fixing the projection matrix $C$ on downstream transfer performance.

| Method | CORA (ACC ↑) | CITESEER (ACC ↑) | MUTAG (ACC ↑) | PROTEINS (ACC ↑) |
|---|---|---|---|---|
| ALL-IN (Fixed $C$) | $81.93 \pm 0.85$ | $68.43 \pm 0.92$ | $91.26 \pm 5.59$ | $75.86 \pm 4.05$ |
| ALL-IN (stochastic $C$) | $82.13 \pm 0.97$ | $69.12 \pm 0.89$ | $92.90 \pm 6.34$ | $78.20 \pm 3.81$ |

Table 17: Effect of using edge features and edge-based covariance operators on ZINC (MAE ↓).

| Method | ZINC (MAE ↓) |
|---|---|
| GIN | 0.3870 |
| GINE (GIN with edge features) | 0.1630 |
| ALL-IN (no edge features) | 0.2583 |
| ALL-IN (with edge features) | 0.1195 |

small positive values of $k$ yields consistent improvements, confirming the benefit of incorporating multi-hop feature information into the covariance operators. Performance largely saturates around 2-3 hops. Thus, we choose to work with $k = 2$ in the main experiments as a good balance between accuracy and efficiency.

### C.17 ASYMPTOTIC COMPUTATIONAL COMPLEXITY

For a graph with $n$ nodes and $m$ edges, with node feature matrix $X \in \mathbb{R}^{n \times d}$, projecting features using a random linear transformation takes $\mathcal{O}(ndh)$ time and $\mathcal{O}(nh)$ memory, where $h$ is the projection dimension. Computing $\{R^{(p)}\}_{p=1}^k$ takes $\mathcal{O}(k(m+n))$ time, as this is equivalent to $k$ message-passing layers propagating $R^{(0)}$. The centering operation takes $\mathcal{O}(knh)$ time. Notably, when explicitly constructing the node-covariance operators $K^{(p)} = \frac{1}{h} R_c^{(p)} (R_c^{(p)})^\top \in \mathbb{R}^{n \times n}$, the computational complexity is $\mathcal{O}(kn^2 h)$ and memory complexity is $\mathcal{O}(kn^2)$ (as $p = 1, \cdots, k$), resulting in quadratic complexity with respect to the number of nodes. This explicit construction may be necessary in certain scenarios such as subgraph GNNs where the full pairwise similarity matrix is required as the graph structure itself (Bevilacqua et al., 2025). However, for standard message passing operations in most MPNNs (Kipf & Welling, 2017; Xu et al., 2019; Rampášek et al., 2022), we can avoid explicitly constructing the covariance matrix. Because message passing can be written as a left-hand multiplication by a propagation matrix (our covariance operator $K$), and by substituting the definition $K = RR^\top$, we can compute $R(R^\top H^{(\ell-1)})$ instead of $(RR^\top) H^{(\ell-1)}$. This way, at no point do we need to hold the full covariance matrix in memory. This approach has computational complexity $\mathcal{O}(k(mh + nhh^{(\ell-1)}))$ and memory complexity $\mathcal{O}(n(h + h^{(\ell-1)}))$ for the entire layer computation, where $h^{(\ell-1)}$ is the feature dimension of $H^{(\ell-1)}$, avoiding the $\mathcal{O}(n^2)$ memory bottleneck while producing mathematically identical results. Therefore, the computational complexity of ALL-IN assuming the covariance matrix does not to be stored, which is the case in our experiments, is $\mathcal{O}(k(mh + nhh^{(\ell-1)}))$ time and $\mathcal{O}(n(h + h^{(\ell-1)}))$ space.

## D  DATASET INFORMATION

In this section, we describe the datasets used in our experiments. We categorize them based on their use in pretraining and task transferability.

### D.1  PRE-TRAINING SOURCE DATASETS (A1)

For pretraining ALL-IN, we use 10 diverse datasets covering molecular graphs, drugs, computer vision, and 3D shapes. The statistics for each dataset are summarized in Table 24. The detailed information is as follows:

- **ZINC** (Dwivedi et al., 2023) is a molecular property prediction dataset where the task is regressing the constrained solubility values of molecules. We report mean absolute error (MAE) as the evaluation metric.

Table 18: ALL-IN pre-training performance on different pre-training corpus, with and without citation networks.

| Pre-training corpus | ZINC (MAE ↓) | MOLESOL (RMSE ↓) | MOLHIV (ROC-AUC ↑) | MOLTOX21 (ROC-AUC ↑) | MNIST (ACC ↑) | CIFAR10 (ACC ↑) | MODELNET (ACC ↑) | CUNEIFORM (ACC ↑) | MSRC21 (ACC ↑) | CORA (ACC ↑) | CITESEER (ACC ↑) |
|---|---|---|---|---|---|---|---|---|---|---|---|
| Original | 0.1237 | 1.29 | 74.49 | 68.20 | 95.22 | 40.08 | 39.37 | 91.17 | 98.08 | – | – |
| Original + Cora + CiteSeer | 0.1253 | 1.30 | 74.52 | 67.99 | 95.18 | 40.12 | 39.21 | 91.08 | 98.23 | 82.89 | 69.33 |

Table 19: Downstream performance of ALL-IN with and without citation-network in pre-training corpus.

| Pre-training corpus | OGBN-ARXIV (ACC ↑) | MUTAG (ACC ↑) | PROTEINS (ACC ↑) |
|---|---|---|---|
| Original (no citation networks) | 75.27 | $92.90 \pm 6.34$ | $78.20 \pm 3.81$ |
| Original + Cora + CiteSeer | 75.61 | $92.68 \pm 6.07$ | $78.24 \pm 3.77$ |

- **MOLHIV, MOLESOL, MOLTOX21** (Hu et al., 2020a) is a collection of molecular graphs from the OGB benchmark covering drug discovery and toxicity prediction tasks. Depending on the dataset, we perform binary classification (MOLHIV), regression (MOLESOL), or multi-label classification (MOLTOX21). Performance is measured using ROC-AUC or RMSE, as appropriate.

- **MNIST, CIFAR10** (Dwivedi et al., 2023) are computer vision datasets converted into graph-structured superpixels. Each image is modeled as a fixed-structure graph, with 1-dimensional input features and a 10-way classification objective.

- **MODELNET** (Wu et al., 2015) is a 3D object classification benchmark where shapes are represented as fixed-size point cloud graphs. We use the 10-class subset.

- **CUNEIFORM** Morris et al. (2020) is a graph-based OCR dataset derived from ancient script symbols, consisting of 62-node graphs with 150 edges on average and a 30-class prediction target.

- **MSRC-21** Morris et al. (2020) is an image segmentation dataset where region adjacency graphs are constructed from visual scenes. Each graph has approximately 212 nodes and 336 edges, with 4-dimensional node features and 21 semantic class labels.

## D.2 TRANSFERABILITY TO UNSEEN DATASETS AND INPUT FEATURES (A2)

To evaluate the transferability of ALL-IN to unseen input features, we choose the following datasets summarized in Table 25 and explained below:

- **CORA, CITESEER, PUBMED** Yang et al. (2016): In these datasets, nodes represent academic papers and edges denote citation links. Each node is assigned a class label corresponding to a subject area. The task is to predict the category of a paper based on its content features and citation graph. Models are evaluated under transductive learning settings using fixed splits Yang et al. (2016).

- **MUTAG** Morris et al. (2020): A binary classification dataset of small molecule graphs. Nodes represent atoms with categorical features, and graphs are labeled based on mutagenic effect on a bacterium.

- **PROTEINS** Morris et al. (2020): A dataset of protein structures modeled as graphs where nodes represent secondary structure elements and edges reflect neighborhood in the amino acid sequence. Each graph is labeled as enzyme or non-enzyme.

## E IMPLEMENTATION DETAILS

We implement ALL-IN using PyTorch (Paszke et al., 2019) (BSD-3 Clause license) and PyTorch Geometric (Fey & Lenssen, 2019) (MIT license). For experiment tracking and hyperparameter logging, we utilize the Weights and Biases framework (Biewald, 2020). Experiments were conducted with NVIDIA RTX A6000, RTX 4090, and NVIDIA A100 GPUs.

Table 20: Transfer to 3D shapes (ShapeNet) and social networks (IMDB-B) with ALL-IN. Higher is better for both mIoU and accuracy.

| Method | ShapeNet (MIOU ↑) | IMDB-B (ACC ↑) |
|---|---|---|
| GIN | 83.6 ± 0.4 | 75.1 ± 5.1 |
| GraphGPS | 84.9 ± 0.2 | 76.3 ± 5.4 |
| ALL-IN | 85.4 ± 0.3 | 77.2 ± 5.0 |

Table 21: Downstream regression transfer on PEPTIDES-STRUCT (MAE ↓).

| Method | PEPTIDES-STRUCT (MAE ↓) |
|---|---|
| GINE | 0.3547 ± 0.0045 |
| GPS | 0.2500 ± 0.0005 |
| ALL-IN | 0.2449 ± 0.0012 |

For all experiments, we use the GPS framework (Rampášek et al., 2022) with the GIN message passing layer (Xu et al., 2019) for $\{\text{GNNLayer}^{(\ell, A)(\cdot, A)}\}_{\ell=0}^{L}$, and we use standard message passing layer for other operators.

### E.1 PRE-TRAINING ON DIFFERENT SOURCE DATASETS (Q1)

To evaluate large-scale transfer, we pretrain ALL-IN on a diverse set of 10 graph datasets spanning multiple domains, as described in Appendix D. Each training epoch cycles through all datasets once, optimizing dataset-specific objectives. We train for 500 epochs and checkpoint every 25 epochs. Hyperparameters are detailed in Table 26. To accelerate training, (1) we use `DataParallel` to support multi-GPU runs, (2) cache the random projection matrix $C$ and refresh every 100 steps, (3) sample 10,000 graphs randomly at each epoch for MNIST and CIFAR10, and (4) sample 128 nodes with 6-nearest neighbors as edges for MODELNET in each graph.

### E.2 EVALUATION ON UNSEEN DATASETS AND INPUT SPACES (Q2)

To evaluate the transferability of ALL-IN to unseen datasets with novel input features, we freeze the pretrained encoder and evaluate its representations by training lightweight classifiers on new target datasets. These datasets span both node-level and graph-level classification tasks, with input feature spaces and labels disjoint from those used during pretraining.

For each target dataset, we instantiate a prediction head using one of the following: (1) a **multi-layer perceptron (MLP)** for both node and graph classification tasks; (2) a **2-layer GCN** Kipf & Welling (2017) applied to node classification benchmarks (CORA, CITESEER, PUBMED); and (3) a **2-layer GIN** Xu et al. (2019) for graph classification benchmarks (MUTAG, PROTEINS). All prediction heads are trained with frozen ALL-IN features as input. No gradients are backpropagated into the encoder during this stage.

For MLPs, we use a single hidden layer of size 128 with ReLU activation, followed by a softmax or sigmoid output layer, depending on whether the task is single-label or multi-label. We train all classifiers using the Adam optimizer with a learning rate of 0.001 and early stopping based on validation loss. Node classification models are trained on the standard 20/30/50 splits Yang et al. (2016) and evaluated using accuracy. For graph classification, we perform 10-fold stratified cross-validation and report the mean and standard deviation of classification accuracy.

All transfer experiments are implemented in PyTorch and PyTorch Geometric. Environment and optimization settings match those described in Appendix E.1.

Table 22: Comparison of supervised vs. unsupervised pre-training of ALL-IN.

| Pre-training approach | Cora (ACC ↑) | MUTAG (ACC ↑) |
|---|---|---|
| ALL-IN (unsupervised) | $82.05 \pm 0.89$ | $91.96 \pm 6.24$ |
| ALL-IN (supervised) | $82.13 \pm 0.97$ | $92.90 \pm 6.31$ |

Table 23: Effect of the number of propagations $k$ on pre-training performance.

| $k$ | ZINC (MAE ↓) | MOLESOL (RMSE ↓) | MOLHIV (ROC-AUC ↑) | MOLTOX21 (ROC-AUC ↑) | MNIST (ACC ↑) | CIFAR10 (ACC ↑) | MODELNET (ACC ↑) | CUNEIFORM (ACC ↑) | MSRC21 (ACC ↑) |
|---|---|---|---|---|---|---|---|---|---|
| 0 | 0.1557 | 1.28 | 72.74 | 68.19 | 94.57 | 40.11 | 37.11 | 89.88 | 97.51 |
| 1 | 0.1415 | 1.29 | 73.60 | 68.30 | 94.95 | 40.20 | 38.20 | 90.40 | 97.85 |
| 2 | 0.1237 | 1.29 | 74.49 | 68.20 | 95.22 | 40.08 | 39.37 | 91.17 | 98.08 |
| 3 | 0.1232 | 1.30 | 74.70 | 68.25 | 95.30 | 40.25 | 39.45 | 91.25 | 98.12 |
| 4 | 0.1239 | 1.30 | 74.65 | 68.22 | 95.28 | 40.18 | 39.30 | 91.10 | 98.05 |

Table 24: Statistics of pre-training datasets used in ALL-IN. The datasets span molecules, drugs, computer vision-derived graphs and 3D shape point clouds. Our pretraining corpus contains up to 200,558 graphs.

| Dataset | # Nodes | # Edges | # Features | # Classes | Domain / Category |
|---|---|---|---|---|---|
| ZINC | 23.2 (avg) | 24.9 (avg) | 28 | - | Molecular Graph Regression |
| OGBG-MOLESOL | 13.3 (avg) | 13.6 (avg) | 9 | - | Solubility Prediction |
| OGBG-MOLHIV | 25.5 (avg) | 27.5 (avg) | 9 | 2 | Drug Discovery |
| OGBG-MOLTOX21 | 18.6 (avg) | 19.4 (avg) | 9 | 12 (multi-label) | Toxicology |
| MNIST (SUPERPIXELS) | 75 | 142 | 1 | 10 | Vision (Digits) |
| CIFAR10 (SUPERPIXELS) | 85 | 170 | 1 | 10 | Vision (Objects) |
| MODELNET | 100 (fixed) | 150 (fixed) | 3 | 40 | 3D Shape Classification |
| CUNEIFORM | 62 (avg) | 150 (avg) | 1 | 30 | Archaeology / OCR |
| MSRC 21 | 212 (avg) | 336 (avg) | 4 | 21 | Image Segmentation |

Table 25: Statistics of finetuning datasets used in our experiments. For node classification datasets (citation networks), we report the total number of nodes and edges. For graph classification datasets (bioinformatics), we report the number of graphs and average graph sizes.

| Dataset | # Graphs / Nodes | # Edges | # Features | # Classes | Domain / Task |
|---|---|---|---|---|---|
| CORA | 2,708 nodes | 5,429 | 1,433 | 7 | Citation Network / Node Classification |
| CITESEER | 3,327 nodes | 4,732 | 3,703 | 6 | Citation Network / Node Classification |
| PUBMED | 19,717 nodes | 44,338 | 500 | 3 | Citation Network / Node Classification |
| MUTAG | 188 graphs | 17.9 (avg) | 7 | 2 | Bioinformatics / Graph Classification |
| PROTEINS | 1,113 graphs | 39.1 (avg) | 3 | 2 | Bioinformatics / Graph Classification |

Table 26: Hyperparameter Configuration for Pretraining Stage.

| Category | Hyperparameter (Value) |
|---|---|
| **Architecture** | |
| Activation Function | ReLU |
| Attention Type in GPS | PerformerAttention |
| GPS Heads | 4 |
| Channels $h^{(\ell)}$ | 256 |
| Random Projection Dim $h$ | 512 |
| Backbone GNNLayer | gps_gine |
| Number of Layers $L$ | 6 |
| Input PE Dim $h_s$ | 20 |
| Use Random Projections | True |
| # Node-Covariance Operators $k$ | 0, 2 |
| **Training Setup** | |
| Pretraining Epochs | 500 |
| Batch Size | 64 |
| Dropout | 0.0 |
| Learning Rate | 0.0001 |
| Weight Decay | 0.0 |
| Normalization Type | batchnorm |

