# OpenReview forum: "Bridging Input Feature Spaces Towards Graph Foundation Models"
_ICLR.cc/2026/Conference — ICLR 2026 Poster_

### Official Review · Reviewer_txCX · 2025-10-26

**Soundness:** 3
**Presentation:** 3
**Contribution:** 3
**Rating:** 6
**Confidence:** 4

**Summary:**

The paper proposes ALL-IN, an input-agnostic framework for transferring graph models across datasets with *different node and edge feature spaces*. It enforces both distributional and orthogonal invariance, supported by theoretical and empirical results. Experimental results show that a single instance of ALL-IN, jointly pre-trained on multiple heterogeneous datasets, performs competitively with per-dataset specialists and generalizes well to unseen datasets.

**Strengths:**

- Provides a clear and compelling principle for bridging feature spaces by operating on node–node covariance operators, which are independent of the original feature dimensionality. This method  integrate naturally into standard GNN layers.
- Introduces an interesting approach to achieve permutation and orthogonal invariance, effectively addressing a key bottleneck in building transferable graph foundation models.
- The method is simple and architecture-agnostic, making it easy to follow and utilize

**Weaknesses:**

- The use of  $n \times n$ covariance operators incurs $\mathcal{O}(n^2)$ memory and computation cost per operator. The paper could benefit from a more in-depth discussion of this limitation, ideally supported by experiments on larger-scale datasets, such as [OGB](https://ogb.stanford.edu/docs/nodeprop/).
- The current transfer setup is skewed toward small, homophilous benchmarks (e.g., MUTAG/PROTEINS contain fewer than 1,000 samples). Evaluating the method on larger-scale benchmarks would help demonstrate its viability as a foundation model.
-  While the method outlines an impact on the edge-feature processing pathway, it remains unclear whether this component was utilized in the experiments. Concrete results or ablation studies to support its effectiveness are missing.

**Questions:**

- Could the authors provide guidance or heuristics—possibly theory-backed—for selecting the projection dimension $h$ and number of propagation orders $k$?
- How does ALL-IN perform under strong heterophily benchmarks? Are the covariance operators dominated by local homophily, and do the propagated operators $K^{(p)}$ help mitigate or worsen this effect?

---

> ### Author Response · Authors · 2025-11-23
> **Response (Part 1)**
>
> We thank the Reviewer for their thoughtful feedback, acknowledging the clarity, simplicity, and theoretical strength of our ALL-IN. We are also grateful for your constructive and thoughtful feedback, which we find to be beneficial for improving the quality of our paper. We have therefore incorporated the changes into our paper, and we provide our individual responses below.
>
> ---
>
> **Q1**. On how the memory and computation cost of computing covariance operators scale to large scale datasets.
>
> **A1**.  We acknowledge the reviewer's insightful comment regarding computational complexity, which warrants clarification. In our original submission, Appendix C10 described the complexity as quadratic to simplify exposition. However, this analysis does not reflect the implementation of ALL-IN with sparse matrices, where the method is in fact computationally efficient.
>
> Specifically, for any GNN layer that leverages sparse adjacency matrices (the standard and memory-efficient approach), ALL-IN scales **linearly in the number of edges and nodes**. The efficiency stems from the properties of the node-covariance operator, defined as $\bf{K}^{(p)} = \frac{1}{h}\bf{R}^{(p)}(\bf{R}^{(p)})^\top$  for $p=0,..,k$, where $k$ is small, 2 in our experiments. This matrix is low-rank by construction. Consequently, the multiplication $\bf{K}^{(p)}\bf{H}^{(l)}$ within a GNN layer incurs a cost of $O(n h^{(l)} h)$. The computation of $\bf{R}^{(p)}$ has cost $O(p m h + d h)$, which is dominated by $O(p m h)$. For all $p$ values this complexity would be $O(p (n h^{(l-1)} h + m h))$. In contrast, standard sparse matrix multiplication with an adjacency matrix $\bf{A}$, i.e., $\bf{A}\bf{H}^{(l-1)}$, has a complexity of $O(m h^{(l-1)})$, where $n$ is the number of nodes, $m$ is the number of edges, and $h^{(l-1)}$ and $h$ are the respective dimensionalities of the layer's feature representations. This confirms that our method maintains linear scaling with respect to graph size, and the dominant complexity of the sparse-matrix multiplication $O(p m C)$, where $C = (h + h^{(l-1)})$ is constant. We updated the paper to reflect this important discussion. Therefore, ALL-IN complexity is $p$ times the complexity of standard GNNs that implement sparse matrix multiplications. Regarding the memory cost, $\bf{R}^{(p)}$ is an $n \times h$ low rank matrix.
>
> ---
>
>
>
>
> **Q2**. On the transferability of ALL-IN in large-scale downstream datasets.
>
>
> **A2**.  Thank you for the question. In Table 11 of the original submission, we provided results on the ogbn-arxiv dataset 169k nodes, 1.1M edges), showing that the transferability of ALL-IN remains strong even on large-scale datasets.  The frozen ALL-IN encoder, with only a light prediction head trained, achieved 75.27% accuracy, surpassing all baselines including OFA (73.22%), GraphText (49.47%), AnyGraph (62.33%), and GraphAny(58.38%). This confirms that the invariant representations learned during pre-training scale effectively to large-scale downstream tasks. We believe that this result, coupled with our discussion in our response in A1, which we also added to the revised paper, help to clarify this important point. Thank you.
>
> ---
>
> **Q3**. On the ablation showing the importance of edge feature covariance operators.
>
> **A3**.  This is an insightful question, and we thank you for it. Following your suggestion, we conducted an ablation to quantitatively demonstrate the importance of the edge feature covariance operators ($\mathbf{K}_{\text{edge}}$), which were utilized for edge-attributed graphs like ZINC.
>
> | Method                      | ZINC MAE ↓ |
> |----------------------------|--------------------------|
> | GIN                        |  0.3870                       |
> | GINE (GIN with edge features) |  0.1630                        |
> | ALL-IN (no edge features)  | 0.2583                       |
> | ALL-IN (with edge features)  | 0.1195                        |
>
> From the Table above, we can see that when we include the edge-based covariance operator, performance is better than when edge features are omitted entirely, and it closely tracks the behavior of  GNNs that use edge features, such as GINE. In contrast, removing edge features yields performance closer to a standard GIN, matching the understanding and findings of graph learning literature, also in our GFM case with ALL-IN. This ablation indicates that the aggregation scheme proposed in Section 3 retains and utilizes edge information for the molecular benchmarks we study. We have added these results and the corresponding discussion to the revised paper. Thank you.

---

> > ### Author Response · Authors · 2025-11-23
> > **Response (Part 2)**
> >
> > **Q4**. Can the authors provide (possibly with theoretical insights) guidance for selection the random projection dimension and number of propagation orders?
> >
> >
> > **A4**.  Thank you for the thoughtful question. In the following, we offer explicit guidance for selecting the projection dimension ($h$) and the number of propagation orders ($k$), supported by both theoretical consistency and empirical saturation points in the supplementary materials (specifically Table 10 for $h$ and Table 9 for $k$).
> >
> > Table 10 in the original submission shows that, across datasets, performance improves from very small $h$ and then plateaus at 256, and gains beyond that are marginal. This trend aligns with Proposition 4.6 (Consistency of Projected Node Covariance): as $h$ grows, the stochastic operator concentrates around its expectation. In practice, a moderate $h$ achieves near-saturated accuracy with a better compute/memory trade-off than a very large $h$.
> >
> > Table 9 in the original submission shows that the addition of propagated operators ($k>0$) consistently boosts performance by incorporating multi-hop feature context. In our initial experiments we found that a small value of $k=2$ was typically sufficient, hence we chose it as a tradeoff between number of operators and computations. Still, we agree with you that it is interesting to present these differences. Therefore, we added an ablation study that shows the performance when using a different number of propagations and hence operators, from k=0 to 4. The results are provided in the Table below and were also added to the revised paper, and they reflect the discussion above. Thank you.
> >
> > | Propagation orders k | ZINC MAE ↓ | MOLESOL RMSE ↓ | MOLHIV ROC-AUC ↑ | MOLTOX21 ROC-AUC ↑ | MNIST ACC ↑ | CIFAR10 ACC ↑ | MODELNET ACC ↑ | CUNEIFORM ACC ↑ | MSRC21 ACC ↑ |
> > |----------------------|-----------:|---------------:|------------------:|--------------------:|------------:|---------------:|----------------:|-----------------:|-------------:|
> > | 0                    | 0.1557     | 1.28           | 72.74             | 68.19               | 94.57       | 40.11          | 37.11           | 89.88            | 97.51        |
> > | 1                    | 0.1415     | 1.29           | 73.60             | 68.30               | 94.95       | 40.20          | 38.20           | 90.40            | 97.85        |
> > | 2                    | 0.1237     | 1.29           | 74.49             | 68.20               | 95.22       | 40.08          | 39.37           | 91.17            | 98.08        |
> > | 3                    | 0.1232     | 1.30           | 74.70             | 68.25               | 95.30       | 40.25          | 39.45           | 91.25            | 98.12        |
> > | 4                    | 0.1239     | 1.30           | 74.65             | 68.22               | 95.28       | 40.18          | 39.30           | 91.10            | 98.05        |

---

> > > ### Author Response · Authors · 2025-11-23
> > > **Response (Part 3)**
> > >
> > > **Q5**. Are the covariance operators used by ALL-IN dominated by local homophily and do the propagated operators help mitigate or worsen the effect of strong heterophily in the graph?
> > >
> > >
> > >
> > >
> > > **A5.** Thank you for the insightful question. Our results reflect that the covariance operators effectively handle heterophily, as mentioned in your review. We understand this in two ways: (i) the node-covariance matrix computed with projected input features captures feature similarity across all nodes, not just local homophily. Therefore, for heterophilous graphs, the covariance operator $K^{(0)}$ may identify similarities between non-adjacent nodes in the original input graph or dissimilarities between adjacent nodes, which can be beneficial for a GNN that needs to resolve heterophily; and (ii) the propagated operators can further help in handling heterophily, because their availability to the GNN allows it to view and mix information from multiple neighborhoods, which has been shown in the literature [1,2] to be beneficial for heterophilous graphs.  Moreover, inspired by your question, we conducted an additional experiment where we measure performance on heterophilic graphs with $k=0$ and $k=1$ propagation orders, and compare it with the results in the paper that use $k=2$ propagations of the covariance. The results are reported in the Table below (and added to the revised paper including the discussion above), and they show that having access to the propagated covariance operators helps in this scenario. We highly appreciate the thoughtful question. Thank you.
> > >
> > > | Number of propagation orders \(k\) | Actor ACC ↑        | Chameleon ACC ↑      | Squirrel ACC ↑       |
> > > |------------------------------------|--------------------|----------------------|----------------------|
> > > | 0                                  | 28.62 ± 0.45       | 65.12 ± 1.44         | 47.89 ± 0.80         |
> > > | 1                                  | 29.00 ± 0.42       | 66.37 ± 1.35         | 49.14 ± 0.76         |
> > > | 2                                  | 29.47 ± 0.38       | 67.40 ± 1.29         | 49.98 ± 0.73         |
> > >
> > >
> > >
> > > ---
> > >
> > > We would like to conclude by extending our gratitude to the reviewer and thanking for the thoughtful comments and suggestions. We hope that you find our responses satisfactory, and that you will take them into consideration in your final assessment of our paper. Thank you.
> > >
> > > ---
> > >
> > > **References:**
> > >
> > > [1] Zhu et al. 2020. Beyond Homophily in Graph Neural Networks: Current Limitations and Effective Designs.
> > >
> > > [2] Chien et al. 2021. Adaptive Universal Generalized PageRank Graph Neural Network.

---

### Official Review · Reviewer_8FFw · 2025-10-29

**Soundness:** 2
**Presentation:** 2
**Contribution:** 2
**Rating:** 2
**Confidence:** 4

**Summary:**

The paper introduces ALL-IN, a method for cross-dataset transfer in graph learning that circumvents incompatible node feature spaces by projecting features into a shared random space and building representations from covariance-based statistics. The authors provide theory showing that the resulting node-covariance operators and representations are distributionally invariant to feature permutations, and that the expected operator is invariant to orthogonal transformations, which removes dependence on the original feature semantics, scales, and dimensionality. Practically, the approach requires no architecture changes or retraining and supports both node- and graph-level tasks on unseen datasets with new feature sets.

**Strengths:**

1. Study the important problem of feature space heterogeneity for GFMs.
2. Provide theoretical understanding of the proposed method.

**Weaknesses:**

1. The pretraining process of the proposed method is unclear. The manuscript does not clearly specify the pretraining objective, the supervision signals used, or how equation 8 aligns with the description in Section 5.1.
2. The data split for pretraining v.s. finetuning is confusing. The pretraining set has feature dimension up to 28, while the three citation datasets used for finetuning have hundreds or thousands of features. Also, the three citation networks have much more nodes and edges than the pretraining graphs. The rationale for this mismatch is not provided, and the effect on transfer is not analyzed.
3. In addition to bullet 2, the authors mention that the pretraining corpus is drawn from multiple domains, yet citation networks are not included despite being used for evaluation. This leads to some confusion about the choice of the pretraining datasets. I would appreciate if the authors can provide some discussion about the principle of dataset selection for pretraining.
4. The number of the downstream datasets is limited in the main text. Results focus on citation and bioinformatics datasets, with little evidence from other domains represented in the pretraining corpus. I would love to see the experimental results from domains other than citation networks and bioinformatics.
5. Table 12 reports results on heterophilic graphs, but the comparison lacks specialized supervised models that are designed for heterophily, which limits the strength of the claim on this setting.
6. The authors provide the complexity analysis in Appendix C.10. Given the quadratic complexity, it would be better to provide a comparison of the training and inference latency with other GFMs, if applicable.

**Questions:**

1. What is the pretraining task and loss? Are node classification and graph classification used in pretraining? According to section 5.1, seems that the model is also trained to perform regression. What is the purpose of including regression tasks in the supervised pretraining if the model is not tested for regression during the downstream validation?
2. Why are supervised tasks chosen for pretraining? Can the authors provide a minimal comparison between supervised and unsupervised pretraining on the proposed method to clarify the underlying principle of the choice of objective?
3. Can the method support edge level tasks such as link prediction? If so, what changes are needed in the representation or the predictor?

---

> ### Author Response · Authors · 2025-11-23
> **Response (Part 1)**
>
> We thank the Reviewer for their thorough feedback, and for acknowledging the importance of our study, as well as the provision of theoretical understanding to our ALL-IN. We are also grateful for the detailed feedback, which we address in our individual responses below. We hope that you find our responses satisfactory, and that you will consider revising your score.
>
> ---
>
> **Q1**. How does equation (8) align with the pre-training procedure described in Section 5.1? What is the resultant pre-training objective when training on several datasets and what are the supervision signals used?
>
>
> **A1**. Thank you for the question. The pre-training objective is a multi-task supervised objective that is composed of the sum of objectives of all pre-training datasets, that is, ALL-IN is optimized with respect to the sum of the task-relevant losses of each dataset. In our paper, we used a pre-training corpus that includes 9 source datasets across multiple domains (molecular, vision, 3D shapes) and diverse tasks: Node Classification (e.g., MNIST/CIFAR-10 superpixels), Graph Classification (e.g., MOLHIV), and Graph Regression (e.g., ZINC). Each task has a dedicated, lightweight prediction head whose output is compared with the target ground truth.
> Thus, during pre-training, the shared ALL-IN encoder is trained to minimize the sum of the task-specific losses (Cross-Entropy for classification, MAE/RMSE for regression) across all datasets sampled in a mini-batch. Regarding Equation (8), it is important to note that as we discussed in Lines 318-322 (now 343-347) in our original submission,
>  Equation (8) is not the pre-training loss, but rather a theoretical upper bound derived from Jensen's inequality (Theorem 4.5). It validates the practice of minimizing the expected loss that is obtained by the random projections used in ALL-IN by showing a bound on the loss of the expected representation. Thank you.

---

> > ### Author Response · Authors · 2025-11-23
> > **Response (Part 2)**
> >
> > **Q2**. What rationale was used to split the datasets into pre-training and fine-tuning stage? How does the graph size and node feature dimension affect the transfer abilities of ALL-IN? Why are citation networks not chosen for pre-training?
> >
> >
> > **A2.** Thank you for the thoughtful question. The split was designed to stress-test feature-agnostic transfer by enlarging the heterogeneity gap between pre-training and downstream tasks. In the main experiments, we pre-train on many relatively small graphs with low-dimensional, dense node features from several domains (molecules, vision, shapes, scripts), and intentionally exclude citation networks (Cora, CiteSeer, PubMed) so that they serve as out-of-distribution targets: they are single large graphs with very high-dimensional, sparse input features (up to 3703 dimensions) and different connectivity patterns. The downstream results on Cora, CiteSeer, and PubMed (e.g., 82.13% on Cora) show that ALL-IN can still transfer effectively under this substantial graph-size and feature-dimension shift, which is precisely the scenario we aimed to probe.
> >
> > To further address your comment, we also consider an extended pre-training setting where Cora and CiteSeer are added to the pre-training corpus. As shown in the table below, adding these citation networks leaves performance on the original pre-training datasets essentially unchanged, while also achieving solid accuracy on Cora and CiteSeer as pre-training tasks:
> >
> > | Pre-training corpus                         | ZINC MAE ↓ | MOLESOL RMSE ↓ | MOLHIV ROC-AUC ↑ | MOLTOX21 ROC-AUC ↑ | MNIST ACC ↑ | CIFAR10 ACC ↑ | MODELNET ACC ↑ | CUNEIFORM ACC ↑ | MSRC21 ACC ↑ | CORA ACC ↑ | CITESEER ACC ↑ |
> > |---------------------------------------------|------------|----------------|------------------|---------------------|-------------|----------------|-----------------|------------------|--------------|------------|----------------|
> > | Original (no citation networks)             | 0.1237     | 1.29           | 74.49            | 68.20               | 95.22       | 40.08          | 39.37           | 91.17            | 98.08        | –          | –              |
> > | Original + Cora + CiteSeer                  | 0.1253     | 1.30           | 74.52            | 67.99               | 95.18       | 40.12          | 39.21           | 91.08            | 98.23        | 82.89      | 69.33          |
> >
> >
> > We then evaluate both pre-training regimes on downstream tasks. As summarized below, including Cora and CiteSeer in the pre-training corpus maintains performance on non-citation datasets (MUTAG, PROTEINS) and slightly improves accuracy on the large citation graph ogbn-arxiv:
> >
> > | Pre-training corpus                         | ogbn-arxiv ACC ↑ | MUTAG ACC ↑           | PROTEINS ACC ↑       |
> > |---------------------------------------------|------------------|------------------------|----------------------|
> > | Original (no citation networks)             | 75.27            | 92.90 ± 6.34           | 78.20 ± 3.81         |
> > | Original + Cora + CiteSeer                  | 75.61            | 92.68 ± 6.07           | 78.24 ± 3.77         |
> >
> > These results indicate that our original choice (keeping citation networks OOD) successfully tests robustness to large graph-size and feature-dimension shifts, and that ALL-IN can also incorporate citation networks in pre-training without degrading, and in fact slightly improving, downstream performance. We have added these experiments and the discussion to the revised paper. Thank you.
> >
> > ---
> >
> > **Q3.** How does ALL-IN transfer to domains other than citation networks and bioinformatics?
> >
> > **A3.** Thank you for the question. To directly address it, we conducted additional experiments on two downstream datasets: (i) ShapeNet part segmentation (3D shape graphs, node-level segmentation) and (ii) IMDB-BINARY (social/movie graphs, graph-level classification). We use the same ALL-IN encoder as in the main experiments and compare it with other GNN baselines. The results are summarized below, where observe that ALL-IN matches or improves upon GIN and GraphGPS baselines on both 3D shapes and social networks, further highlighting the effectiveness of ALL-IN. We added this discussion and results to the revised paper. Thank you.
> >
> > | Method    | ShapeNet mIoU ↑ | IMDB-B ACC ↑ |
> > |-----------|------------------------------|----------------------------|
> > | GIN       | 83.6 ± 0.4                   | 75.1 ± 5.1                 |
> > | GraphGPS  | 84.9 ± 0.2                   | 76.3 ± 5.4                 |
> > | ALL-IN    | 85.4 ± 0.3                   | 77.2 ± 5.0                 |

---

> > > ### Author Response · Authors · 2025-11-23
> > > **Response (Part 3)**
> > >
> > > **Q4**. Can you compare with specialized supervised baselines designed for heterophilic graph datasets?
> > >
> > > **A4**.  Following your suggestion, we present below a comparison against a specialized heterophily model, namely
> > >
> > > | Category | Method | Actor (ACC $\uparrow$) | Chameleon ($\uparrow$) | Squirrel ($\uparrow$) |
> > > | :--- | :--- | :--- | :--- | :--- |
> > > | **NON-PARAMETRIC** | Label Propagation | $18.83 \pm 0.00$ | $40.89 \pm 0.00$ | $33.42 \pm 0.00$ |
> > > | | | | | |
> > > | **SUPERVISED BASELINES** | GCN | $28.55 \pm 0.68$ | $64.69 \pm 2.21$ | $47.07 \pm 0.71$ |
> > > | | | | | |
> > > | **SPECIALIZED METHODS** | H2GCN | $35.70 \pm 1.00$ | $60.11 \pm 2.15$ | $36.48 \pm 1.86$ |
> > > | | WRGAT | $36.53 \pm 0.77$ | $65.24 \pm 0.87$ | $48.85 \pm 0.78$ |
> > > | | GPR-GNN | $34.63 \pm 1.22$ | $46.58 \pm 1.71$ | $31.61 \pm 1.24$ |
> > > | | GGCN | $37.54 \pm 1.56$ | $71.14 \pm 1.84$ | $55.17 \pm 1.58$ |
> > > | | ACM-GCN | $36.28 \pm 1.09$ | $66.93 \pm 1.85$ | $54.40 \pm 1.88$ |
> > > | | LinkX | $36.10 \pm 1.55$ | $68.42 \pm 1.38$ | $61.81 \pm 1.80$ |
> > > | | GloGNN++ | $37.70 \pm 1.40$ | $71.21 \pm 1.84$ | $57.88 \pm 1.76$ |
> > > | | | | | |
> > > | **FOUNDATION / TRANSFER** | GraphAny | $28.60 \pm 0.21$ | $62.59 \pm 0.86$ | $49.70 \pm 0.95$ |
> > > | | ULTRA | $22.61 \pm 0.00$ | N/A | N/A |
> > > | | SCORE | $23.26 \pm 0.56$ | N/A | N/A |
> > > | | ALL-IN (ours) | $29.47 \pm 0.38$ | $67.40 \pm 1.29$ | $49.98 \pm 0.73$ |
> > >
> > > We present the comparison in the table above. As a Graph Foundation Model (GFM), ALL-IN consistently yields the highest accuracy in its category, surpassing the next best GFM, GraphAny. When compared to specialized supervised baselines, ALL-IN exceeds H2GCN by 7.29% on Chameleon and 13.5% on Squirrel, and outperforms standard GCN by 2.71% and 2.91% respectively. While state-of-the-art specialized models like GloGNN++ retain the lead, they benefit from full supervised training on the target graph structure and, importantly, a designated architecture for heterophily.  We believe that these results further highlight the promise of GFMs and the motivation for research both on their theoretical and empirical understandings, and in particular the ability to bridge multiple input feature spaces, as in our ALL-IN. Thank you.
> > >
> > > ---
> > >
> > > **Q5**. Complexity of ALL-IN
> > >
> > > **A5**. We acknowledge the reviewer's insightful comment regarding computational complexity, which warrants clarification. In our original submission, Appendix C10 in our original submission (now C17 in the revised paper) described the complexity as quadratic to simplify exposition. However, this analysis does not reflect the implementation of ALL-IN with sparse matrices, where the method is in fact computationally efficient.
> > >
> > > Specifically, for any GNN layer that leverages sparse adjacency matrices (the standard and memory-efficient approach), ALL-IN scales **linearly in the number of edges and nodes**. The efficiency stems from the properties of the node-covariance operator, defined as $\bf{K}^{(p)} = \frac{1}{h}\bf{R}^{(p)}(\bf{R}^{(p)})^\top$ for $p=0,..,k$, where $k$ is small, 2 in our experiments. This matrix is low-rank by construction. Consequently, the multiplication $\bf{K}^{(p)}\bf{H}^{(l)}$ within a GNN layer incurs a cost of $O(n h^{(l)} h)$. The computation of $\bf{R}^{(p)}$ has cost $O(p m h + d h)$, which is dominated by $O(p m h)$. For all $p$ values this complexity would be $O(p (n h^{(l-1)} h + m h))$. In contrast, standard sparse matrix multiplication with an adjacency matrix $\bf{A}$, i.e., $\bf{A}\bf{H}^{(l-1)}$, has a complexity of $O(m h^{(l-1)})$, where $n$ is the number of nodes, $m$ is the number of edges, and $h^{(l-1)}$ and $h$ are the respective dimensionalities of the layer's feature representations. This confirms that our method maintains linear scaling with respect to graph size, and the dominant complexity of the sparse-matrix multiplication $O(p m C)$, where $C = (h + h^{(l-1)})$ is constant. We updated the paper to reflect this important discussion. Therefore, ALL-IN complexity is $p$ times the complexity of standard GNNs that implement sparse matrix multiplications.

---

> > > > ### Author Response · Authors · 2025-11-23
> > > > **Response (Part 4)**
> > > >
> > > > **Q6**. Are node classification and graph classification used in pretraining?  What is the purpose of including regression tasks (ZINC) in the supervised pre-training if the model is not tested for regression during the finetuning/downstream stage?
> > > >
> > > > **A6.** Thank you for the question. The pre-training stage in ALL-IN uses a supervised multi-task objective over several graph-level datasets, including both graph classification (e.g., OGBG-MOLHIV, MODELNET) and graph regression (e.g., ZINC). This design choice reflects our goal of learning a single encoder that is useful across diverse graph modalities and objectives, rather than tailoring the pre-training strictly to the particular downstream benchmarks we evaluate. The motivation for including regression tasks such as ZINC in the pre-training mix is inspired by the broader multi-task and foundation-model literature: training a shared encoder on a diverse collection of tasks and objectives is widely used to encourage more general-purpose representations [1,2]. In our case, we follow this principle and treat available regression datasets as additional sources of supervised signal that help train ALL-IN, even though our main downstream study focuses on classification tasks.
> > > > Moreover, to directly demonstrate regression-style transfer, we evaluate ALL-IN on a held-out regression benchmark not used during pre-training, Peptides-struct from the LRGB suite. We compare strong GNN baselines (GINE and GPS) with ALL-IN using the same encoder as in our main experiments and a small regression head on top. As shown in the table below, ALL-IN achieves the lowest mean absolute error on  the Peptides-struct dataset.  The results confirm that ALL-IN is also effective for downstream regression tasks on new datasets. We have added this experiment and the discussion to the revised paper. Thank you.
> > > >
> > > > | Method                        | Peptides-struct MAE ↓|
> > > > |-------------------------------|-------------------------------------|
> > > > | GINE                          | 0.3547 ± 0.0045                    |
> > > > | GPS                      | 0.2500 ± 0.0005                    |
> > > > | ALL-IN   | 0.2449 ± 0.0012                    |
> > > >
> > > > ---
> > > >
> > > > **Q7**. Why are supervised tasks chosen for pre-training? Can the authors provide a minimal example of comparing supervised and unsupervised pre-training?
> > > >
> > > > **A7.** Thank you for the question. We chose a multi-task supervised pre-training objective because our goal is to leverage existing graph datasets and their task labels for learning diverse and input-space agnostic representations when target labels are readily available in the source domain. The central goal of ALL-IN is to offer a novel input-space bridge that can then be utilized in downstream tasks, and the choice of supervised versus unsupervised pre-training sits on top of this bridge. While we find unsupervised and self-supervised pre-training to be valuable strategies, prior work on graph representation learning has shown that incorporating supervised objectives in pre-training, when labels are available, can yield stronger and more task-discriminative representations than purely self-supervised objectives [3]. Similarly, large-scale vision studies continue to treat supervised pre-training as a very competitive baseline even in the presence of powerful self-supervised methods [4]. In our setting, where labeled source datasets are available across several graph modalities, we therefore adopt a supervised multi-task objective for pre-training ALL-IN. Nonetheless, following your suggestion, we now include a minimal empirical comparison between supervised and unsupervised pre-training on top of our input-space bridge. In the unsupervised variant, we replace the supervised losses on the pre-training datasets with a masked-feature reconstruction objective as in [5]. As summarized in the table below, both pre-training regimes achieve similar downstream performance, with the supervised variant slightly outperforming the unsupervised, which is consistent with the observations in [3,4].  We also added this important discussion and results to the revised paper. Thank you.
> > > >
> > > > | Pre-training approach   | Cora ACC ↑          | MUTAG ACC ↑          |
> > > > |-------------------------|---------------------|----------------------|
> > > > | ALL-IN (unsupervised)   | 82.05 ± 0.89        | 91.96 ± 6.24         |
> > > > | ALL-IN (supervised)     | 82.13 ± 0.97        | 92.90 ± 6.31         |

---

> > > > > ### Author Response · Authors · 2025-11-23
> > > > > **Response (Part 5)**
> > > > >
> > > > > **Q8**. What modifications are needed for ALL-IN to support edge-level tasks such as link prediction?
> > > > >
> > > > > **A8.** We appreciate the question. To support edge-level tasks such as link prediction, only minimal modifications are needed on top of ALL-IN: the model already produces input space agnostic node embeddings $\mathbf{H}^{(L)}$, and a standard link-prediction head can be added. For a candidate edge $(u,v)$, one can construct a pairwise feature vector from $\mathbf{H}^{(L)}_u$ and $\mathbf{H}^{(L)}_v$ (e.g., via concatenation, elementwise product, or absolute difference) and feed it into a small MLP or similar scorer. If the goal is to reach state-of-the-art link prediction performance, the main design choices concern the downstream architecture rather than modifications to ALL-IN’s input-space bridge. Our encoder can be plugged into existing GNN-based link-prediction frameworks by simply replacing their node feature inputs with ALL-IN’s embeddings. For example, one can use ALL-IN within standard message-passing GNN backbones as in [6,7], or with more recent methods SEAL [8]. In all these possibilities, the core ALL-IN encoder remains unchanged; only the downstream link-prediction head or link-aware architecture is adapted. Thank you.
> > > > >
> > > > > ----
> > > > >
> > > > > We would like to conclude by expressing our sincere gratitude for your detailed feedback. We have worked to address your comments both by additional experiments and clarifications and discussions on different aspects of ALL-IN. We truly hope that you find our responses satisfactory, and that you will consider revising your score. Thank you.
> > > > >
> > > > >
> > > > >
> > > > > ----
> > > > >
> > > > > **References**:
> > > > >
> > > > > [1] Raffel et al. 2020. Exploring the Limits of Transfer Learning with a Unified Text-to-Text Transformer.
> > > > >
> > > > > [2] Zhang & Yang. 2020. “A Survey on Multi-Task Learning.
> > > > >
> > > > > [3] Hu, et al. 2019. Strategies for pre-training graph neural networks.
> > > > >
> > > > > [4] He et al. 2022. Masked Autoencoders Are Scalable Vision Learners.
> > > > >
> > > > > [5] Hou et al. 2022. GraphMAE: Self-Supervised Masked Graph Autoencoders.
> > > > >
> > > > > [6] Srinivasan et al., 2020. On the equivalence between positional node embeddings and structural graph representations
> > > > >
> > > > > [7] Zhang et al., 2021. Labeling Trick: A Theory of Using Graph Neural Networks for Multi-Node Representation Learning
> > > > >
> > > > > [8] Zhang et al., 2018. Link Prediction Based on Graph Neural Networks

---

### Official Review · Reviewer_ptNs · 2025-10-31

**Soundness:** 3
**Presentation:** 3
**Contribution:** 3
**Rating:** 4
**Confidence:** 4

**Summary:**

This paper tackles a critical problem for graph models: graph datasets have inconsistent node features (like different dimensions or meanings), which stops a model trained on one dataset from working on another. The authors propose ALL-IN, a method that solves this by projecting features into a shared random space. Instead of using the original features, it computes a node-covariance matrix that captures node similarities. The GNN then uses this matrix to learn, making the model independent of the original, messy feature space. This allows ALL-IN to generalize to new, unseen datasets with completely different features, without needing to be retrained.

**Strengths:**

1. The paper correctly identifies input feature heterogeneity as a fundamental and persistent obstacle to cross-dataset generalization, and its focus on solving this core problem is a valuable contribution to the field.
2. The proposed ALL-IN method is supported by theoretical proofs demonstrating that its representations are robust to feature re-ordering (distributional invariance) and basis changes (orthogonal invariance), ensuring it captures stable properties of the data.
3. The model shows strong empirical results, not only maintaining competitive performance when jointly trained on diverse datasets but also, more importantly, successfully transferring to new, unseen datasets with entirely different features without retraining.

**Weaknesses:**

1. On the domain difference. The authors should explain how the domain-specific knowledge is modeled and preserved in ALL-IN.
2. Another main concern is the computational complexity, as a general-purpose foundation model is usually trained on massive data at scale. Given that the covariance is expensive, complexity analysis and comparison against other GFMs or pre-training strategies are strongly encouraged.
3. This paper lacks some recent advances on GFM, and it will be more interesting if the following papers are well discussed and compared.
a) NeurIPS24: GFT: Graph Foundation Model with Transferable Tree Vocabulary
b) WWW25: SAMGPT: Text-free Graph Foundation Model for Multi-domain Pre-training and Cross-domain Adaptation
c) WWW25: Riemanngfm: Learning a graph foundation model from structural geometry
d) OpenGraph: Towards open graph foundation models.
e) ICML25: How Much Can Transfer? BRIDGE: Bounded Multi-Domain Graph Foundation Model with Generalization Guarantees
f) ICML25: AutoGFM: Automated Graph Foundation Model with Adaptive Architecture Customization
Also, the authors are required to specify difference between the aforementioned papers to defend their novelty.

**Questions:**

1. How can the $\mathcal{O}(n^2)$ complexity scale to foundation model-sized graphs? Are the sparse approximations you mentioned practical?
2. How critical is it to sample the projection matrix $C$ at each forward pass? Your theory (Prop 4.1, Thm 4.3) relies on this stochasticity. Would the method's benefits (especially distributional invariance) be lost if $C$ were sampled only once and then fixed throughout all training and inference?
3. The method achieves invariance by reducing features to their second-order statistics. This must inherently discard some information. Could you elaborate on what types of task-relevant information might be lost? For example, how would the model distinguish between two nodes with feature vectors $x$ and $-x$, given that they would contribute identically to the expected covariance matrix $\Pi_{c}XX^{T}\Pi_{c}$?
4. The proposed strategy for handling edge features (projecting, aggregating to nodes, then computing a new covariance operator $K_{edge}^{(p)}$) is not empirically ablated. Was this component used for pre-training on datasets like ZINC, which have edge attributes? How does the simple node-level aggregation avoid losing critical relational information, and can you provide an ablation study on this component's impact?

---

> ### Author Response · Authors · 2025-11-23
> **Response (Part 1)**
>
> We thank the reviewer for their insightful comments and for recognizing the value of addressing input feature heterogeneity, the soundness of our theoretical proofs, and our strong empirical results. We address the specific questions below. We hope that you find our responses satisfactory, and that you will consider revising your score.
>
> ---
>
> **Q1**. Comparison to other foundation models.
>
> **A1.** We appreciate the suggestion and the opportunity to discuss these recent graph foundation models. ALL-IN differs from these works along three main axes: (1) theoretical expressivity of the feature-bridging operator, (2) frozen-encoder transfer without prompt tuning, and (3) input universality beyond text-attributed graphs. Where applicable, we have added direct comparisons from the papers, and we cited all suggested papers including a discussion in the related works section, where the key differences are:
>
> **Theoretical Expressivity.** BRIDGE and SAMGPT address feature or domain heterogeneity via deterministic alignment modules, such as domain-invariant feature projectors, structure tokens and prompts. As shown in **Theorem 4.3**, covariance features built on deterministic projections like BRIDGE and SAMGPT preserve graph automorphisms and can leave structurally symmetric nodes indistinguishable, whereas the stochastic random projections in ALL-IN break these symmetries while remaining equivariant in distribution. This gives ALL-IN strictly richer symmetry-breaking behavior for feature-space alignment. RiemannGFM is complementary here: it focuses on learning a structural vocabulary of trees and cycles in a Riemannian manifold, while ALL-IN specifically targets heterogeneity in the input feature spaces and can in principle be combined with such structural backbones.
>
> **Adaptation Cost.** GFT adapts to new datasets by fine tuning the backbone to align target data with its learned tree vocabulary. SAMGPT and BRIDGE keep a pre-trained encoder but require optimizing domain-specific prompts or experts for each new graph domain. AutoGFM further performs graph neural architecture search on top of an LLM-based GFM, which is powerful but relatively expensive. In contrast, ALL-IN uses a single frozen encoder across all datasets; thanks to the input-space invariant covariance features, we only train a lightweight task head and do not require any domain-specific prompts or feature aligners. This makes ALL-IN a general-purpose feature extractor for unseen domains, with lower adaptation cost than prompt tuning or full fine tuning.
>
> **Input Universality.** GFT, AutoGFM and OpenGraph are instantiated and evaluated on text-attributed graphs, where node and edge information is encoded via large language models and unified through text embeddings or LLM-generated structures. These approaches are very effective when rich textual descriptions are available, but they assume access to such text and do not directly target raw numerical features such as images or molecular descriptors. ALL-IN, by contrast, is input feature-space agnostic: it operates on generic numerical node features, without any textualization or LLM, and unifies arbitrary feature spaces through the covariance operator. This allows ALL-IN to learn from and transfer between both text-free and text-based domains, and it is complementary to TAG-based GFMs.
>
> ---
>
> **Q2**. Can you explain how domain-specific knowledge is modeled and preserved in ALL-IN if you use random projections and node-feature covariance operators?
>
> **A2**. Thank you for the question. In our work, we show that domain-specific knowledge is encoded in the correlations between node features and the graph topology. As shown in Theorem 4.4, the expected node-covariance operator converges to the Gram matrix of the centered original features ($\Pi_c X X^T \Pi_c$), which captures the relative similarity and dissimilarity between nodes in the original feature space, while operating in a unified feature space given by the random projection, thereby allowing to work with different, heterogenous input feature spaces, addressing feature heterogeneity in graph learning datasets. Moreover, the GNN layers in ALL-IN process the covariance-based operators ($K^{(p)}$) alongside the graph structure ($A$). By learning how feature correlations (covariance) interact with structural propagation, the model captures domain-invariant patterns (e.g., "nodes with correlated features tend to be connected"), effectively preserving domain knowledge from the original input space, within in this unified feature space, towards graph foundation models.

---

> > ### Author Response · Authors · 2025-11-23
> > **Response (Part 2)**
> >
> > **Q3**. How does the computational complexity of ALL-IN compare with that of other GFMs?
> >
> > **A3**. We acknowledge the reviewer's insightful comment regarding computational complexity, which warrants clarification. In our original submission, Appendix C10 (now C17 in the revised paper) described the complexity as quadratic to simplify exposition. However, this analysis does not reflect the implementation of ALL-IN with sparse matrices, where the method is in fact computationally efficient.
> >
> > Specifically, for any GNN layer that leverages sparse adjacency matrices (the standard and memory-efficient approach), ALL-IN scales **linearly in the number of edges and nodes**. The efficiency stems from the properties of the node-covariance operator, defined as $\bf{K}^{(p)} = \frac{1}{h}\bf{R}^{(p)}(\bf{R}^{(p)})^\top$  for $p=0,..,k$, where $k$ is small, 2 in our experiments. This matrix is low-rank by construction. Consequently, the multiplication $\bf{K}^{(p)}\bf{H}^{(l)}$ within a GNN layer incurs a cost of $O(n h^{(l)} h)$. The computation of $\bf{R}^{(p)}$ has cost $O(p m h + d h)$, which is dominated by $O(p m h)$. For all $p$ values this complexity would be $O(p (n h^{(l-1)} h + m h))$. In contrast, standard sparse matrix multiplication with an adjacency matrix $\bf{A}$, i.e., $\bf{A}\bf{H}^{(l-1)}$, has a complexity of $O(m h^{(l-1)})$, where $n$ is the number of nodes, $m$ is the number of edges, and $h^{(l-1)}$ and $h$ are the respective dimensionalities of the layer's feature representations. This confirms that our method maintains linear scaling with respect to graph size, and the dominant complexity of the sparse-matrix multiplication $O(p m C)$, where $C = (h + h^{(l-1)})$ is constant. We updated the paper to reflect this important discussion. Therefore, ALL-IN complexity is $p$ times the complexity of standard GNNs that implement sparse matrix multiplications.
> >
> > ---
> >
> > **Q4**. Would ALL-IN’s benefits (especially distributional invariance) be lost if the random projection matrix $C$ is sampled only once and then fixed throughout training and inference?
> >
> > **A4**.  Thank you for the insightful question. Our theoretical guarantees are stated for the stochastic setting where $\mathbf{C}$ is treated as a random variable. In particular, the distributional invariance in Proposition 4.1, $\mathbf{R}^{(0)} \stackrel{d}{=} (\mathbf{X}\mathbf{P})\mathbf{C}$, relies on the sampling of $\mathbf{C}$, and the symmetry-breaking statement in Theorem 4.3 is also formulated with respect to draws of $\mathbf{C}$. When $\mathbf{C}$ is sampled once and then fixed, these guarantees no longer apply in the same formal sense. Nonetheless, inspired by your question, we conducted an additional ablation where we sample $\mathbf{C}$ once and keep it fixed for all steps (ALL-IN (Fixed $C$)). As shown in the table below, fixing $C$ leads to a moderate degradation on all pre-training datasets. A similar pattern holds on downstream tasks, also presented in the Table below. These ablations empirically support the beneficial role of stochastic projections. We added the results to the revised paper. Thank you.
> >
> >
> >
> > | Method                    | ZINC MAE ↓ | MOLESOL RMSE ↓ | MOLHIV ROC-AUC ↑ | MOLTOX21 ROC-AUC ↑ | MNIST ACC ↑ | CIFAR10 ACC ↑ | MODELNET ACC ↑ | CUNEIFORM ACC ↑ | MSRC21 ACC ↑ |
> > |---------------------------|-----------:|---------------:|-----------------:|--------------------:|------------:|--------------:|---------------:|----------------:|-------------:|
> > | ALL-IN (Fixed $\mathbf{C}$)          |    0.1369  |          1.38  |            74.12 |              66.72  |      93.97  |         39.84 |          39.02 |          90.11  |       96.27  |
> > | ALL-IN (as in paper)     |    0.1237  |          1.29  |            74.49 |              68.20  |      95.22  |         40.08 |          39.37 |          91.17  |       98.08  |
> >
> >
> >
> >
> > | Method                     | Cora ACC ↑        | Citeseer ACC ↑     | MUTAG ACC ↑        | PROTEINS ACC ↑      |
> > |----------------------------|-------------------|--------------------|--------------------|----------------------|
> > | ALL-IN (Fixed $\mathbf{C}$)           | 81.93 ± 0.85      | 68.43 ± 0.92       | 91.26 ± 5.59       | 75.86 ± 4.05         |
> > | ALL-IN (as in paper)      | 82.13 ± 0.97      | 69.12 ± 0.89       | 92.90 ± 6.34       | 78.20 ± 3.81         |

---

> ### Author Response · Authors · 2025-11-23
> **Response (Part 3)**
>
> **Q5**. What types of task-relevant information is lost when the features are reduced to their second-order statistics? How would ALL-IN distinguish between two nodes with feature vectors $x$ and $-x$if they contribute identically to the expected covariance matrix?
>
> **A5**. This is an important question. We would like to clarify that while the covariance matrix captures second-order statistics, ALL-IN does not discard the first-order information. This is because the model uses the randomly projected features $\mathbf{R}^{(0)}=\mathbf{XC}$ directly as the initial node embeddings $\mathbf{H}^{(0)}$ (Equation 5). Regarding distinguishability, we note that, for nodes $u$ and $v$ with features $x$ and $-x$, the initial input signals are $\mathbf{R}^{(0)}_u = x \mathbf{C}$ and $\mathbf{R}^{(0)}_v = -x \mathbf{C}$. Since these are distinct, opposite vectors, the GNN starts with a clear distinction between the nodes. Furthermore, we note that within the covariance operator, the nodes are distinguishable: while the auto-covariance terms are identical and yield ($x x^\top$), the cross-covariance terms with any third node $z$ will be proportional to $x z^\top$ for $u$ and $-x z^\top$ for $v$. This means that the rows of the operator $\mathbf{K}$ are distinct, ensuring different message propagation. We have revised Section 3 in the paper to reflect this important discussion. Thank you.
>
> ---
>
> **Q6**. Were covariance operators computed using edge features used for pre-training on datasets like ZINC (which have edge features)? How does simple node-level aggregation of projected edge features avoid losing critical information? Can you show this using an ablation?
>
> **A6.** Thank you for the comment. For datasets with edge features like ZINC, we use the strategy described in Section 3: edge features are randomly projected, aggregated to nodes, and used to compute supplementary covariance operators that are added to the operator set.
> We acknowledge that, mathematically, node-level aggregation can lose information by transforming multiple edge feature vectors into a single aggregated vector per node. As a general-purpose model, this design is a trade-off between information granularity, efficiency, and compatibility with our node-wise covariance framework. Importantly, this aggregated signal is not used in isolation: it is converted into an $n \times n$ edge covariance operator $K_{edge}$, whose entries compare the aggregated edge-feature environments of all node pairs. In parallel, the backbone still utilizes the raw edge features, so the aggregated edge branch acts as a complementary operator rather than the only channel for edge information. To further accommodate your comment, we now add an ablation study that quantifies the contribution of using edge features in ALL-IN using our approach described in Section 3. The empirical effect of this design is validated on ZINC which includes edge features, as shown in the table below.
>
> | Method                      | ZINC MAE ↓ |
> |----------------------------|--------------------------|
> | GIN                        |  0.3870                       |
> | GINE (GIN with edge features) |  0.1630                        |
> | ALL-IN (no edge features)  | 0.2583                       |
> | ALL-IN (with edge features)  | 0.1195                        |
>
> From the Table above, we can see that when we include the edge-based covariance operator, performance is better than when edge features are omitted entirely, and it closely tracks the behavior of  GNNs that use edge features, such as GINE. In contrast, removing edge features yields performance closer to a standard GIN, matching the understanding and findings of graph learning literature, also in our GFM case with ALL-IN. This ablation indicates that the aggregation scheme proposed in Section 3 retains and utilizes edge information for the molecular benchmarks we study. We have added these results and the corresponding discussion to the revised paper. Thank you.
>
> ---
>
> We would like to express our appreciation of your review. It touches important points that we now address, and we remain available to address any additional comments or questions you may have. We hope that you find our responses in order, and that you will consider revising your score.

---

### Official Review · Reviewer_PJTg · 2025-11-02

**Soundness:** 4
**Presentation:** 4
**Contribution:** 4
**Rating:** 10
**Confidence:** 4

**Summary:**

Finding a good and general approach for graph foundation models is a relevant, important and actual topic in representation learning for structured data. The present article develops a really interesting approach in this direction, by proposing a novel way to think about misalignement between attributed graph datasets. The idea is to randomly project the features on the nodes and to compute the node-covariances operators though these projections. This ensures that key properties of the features are kept (e.g., strong relations between the features of some nodes) while providing a representation of these features which is invariant to permutations and more generally orthogonal rotations, of the features, and also which has the same dimension whatever the initial dataset.

The key idea in this work is very clever. The section 3 first details the method using this random projection and the concept of node-covariance operators, and secondly proposes a novel node representation method by using these elements inside a GNN (which encodes the structure, while the random features and node-covariance operators encode the attributes,  possibly supplemented to structural encodings if one wants). Then Section 4 proposes a thorough theoretical analysis of the key properties: invariance of the representation under orthogonal rotations ; cases of distinguishability ; proof that the training provides an upper bound of the training objectives formulated ; some elements on the transferability of the representation. All that are really good elements.

Numerical experiments follow and they are solid. The reader basically will agree to the steps and the conclusions obtained by the authors. The obtained performance are good, both as general purpose representions (Table 1), and for transfer to new datasets (Tables 2 and 3).

**Strengths:**

The strengths of the article are :

1. A very good and novel proposition to build a transferable, general and learnable representation of attributed graphs with consistency across datasets, and transferability. Hence, this work is a very good step toward an efficient graph foundation model.

2. There are key theoretical insights, well proven in the article (or its Appendices) that show the impact and the importance of the various elements.

3. Numerical experiments and conducted on several situations. The results are compared to a variety of state-of-the-art methods for graph learning (some supervised baselines ; some GFMs using LLM ; some GFMs built on top of GNN like the present method). The  complementary studies about the separability (linear or non-linear) of the obtained representation, and the ablation study are good also (for ablation: to study the impact of structural and positional encoding ; the impact of random projections ; the impact of using covariance operator ; of propagating it ; 	and so on). This is really a solid work.

4. The writing of the article is globally exemplar.

**Weaknesses:**

I did not find any weakness in this paper.

**Questions:**

I have some questions or suggestions:

The step of equation (1) could be stressed a little bit more, possibly writing already here some insight about why you propose that. Also, could it be generalised to other types of random projections, or of sketching ?

* For theorem 4.5: I would put the remark in parentheses ("this holds,… ") outside of the theorem. Possibly just underneath.

* In 5.1 : it appears to be only 9 datasets, while D.1 writes 10 (we only have the results for 9 in the paper).

* Table 1: the order of the data is not natural. It would be easier to have it either ordered so that data with the same metric are adjacent, or ordered in the same manner than in the text (ModelNet should then be last). Still, having dataset where smaller is better first (ZINC, MOLSOL), then ROC-AUC (MOLHIV, MOLTOX21) and finally datasets measured with ACC (MOLHIV and the others) would be nice.

* Do you have a comment to compare the proposed ALL-IN to SCORE which appears to be a good competitors. Do you think that ALL-IN can be generally better to transfer in unseen graphs (as in Table 3) or do you currently know no more ?

* page 23, third bullet point: $+ S$ should be $ \oplus S $, shouldn't it ?

**Details Of Ethics Concerns:**

No concerns.

---

> ### Author Response · Authors · 2025-11-23
> **Response (Part 1)**
>
> We are delighted by your exceptionally positive review and strong recommendation, recognizing ALL-IN's novelty and comprehensive theoretical foundation. We agree that resolving feature heterogeneity is a crucial step toward efficient Graph Foundation Models. We have addressed your suggestions regarding style, presentation, and minor corrections below, as well as answered the questions you posed. Thank you.
>
> ---
>
> **Q1**. Corrections and presentation edits.
>
> **A1**. We appreciate your detailed suggestions for refinement and confirm that the following changes have been applied to our revised paper :
> The formal theorem statement for Theorem 4.5 has been improved by moving the explanatory remark regarding convexity ("this holds, for instance...") out of the theorem block and into the accompanying prose immediately underneath. This enhances the clarity and formality of the theorem.
>
> We clarified the text in Section 5.1 to address the ambiguity in the dataset count, confirming that the results are presented for 9 source datasets.
>
> The order of datasets in Table 1 (Performance on pre-training datasets) has been reordered to group datasets by evaluation metric (MAE/RMSE, then ROC-AUC, then ACC) for improved logical flow and readability, making performance comparisons easier for the reader.
>
> We also corrected the typo in the Appendix (page 23, third bullet point) where concatenation was incorrectly written as addition, replacing $\mathbf{X} + \mathbf{S}$ with the correct notation for matrix concatenation, $\mathbf{X} \oplus \mathbf{S}$. Thank you.
>
> ---
>
> **Q2**. The step of equation (1) could be stressed a little bit more, possibly writing already here some insight about why you propose that. Also, could it be generalised to other types of random projections, or of sketching ?
>
>
> **A2**. Thank you for the insightful comment and interesting question. We agree that the initial transformation deserves additional emphasis. We revised the text before Equation (1) ($\mathbf{R}^{(0)} = \mathbf{XC}$) in Section 3, to better highlight its core function: establishing a common, fixed feature dimensionality $\mathbf{h}$ that is independent of the original arbitrary input dimension $d$. This step is foundational for enabling cross-dataset processing. Regarding generalization to other types of random projections: **Yes.** ALL-IN theoretically accommodates other random projections (e.g., Rademacher random projection, sparse sketching) provided they satisfy the following conditions: (1) **Permutation Invariance (Proposition 4.1):** The projection matrix should consist of i.i.d entries. As long as this holds, the distribution of projected features remains invariant to node permutations. (2) **Consistency (Proposition 4.6):** The node-covariance operator should converge to its expected value in probability (this holds for a broad class of random matrices beyond Gaussians). We utilized Gaussian projections to satisfy the stronger condition of invariance to general orthogonal rotations (**Theorem 4.4**). We believe that studying additional projection or sketching approaches is an exciting future research direction. Thank you.

---

> > ### Author Response · Authors · 2025-11-23
> > **Response (Part 2)**
> >
> > **Q3**. SCORE appears to be a strong competitor. Do you think that ALL-IN can be generally better to transfer in unseen graphs (as in Table 3) or do you currently know no more ?
> >
> > **A3**. We acknowledge that SCORE (Wang and Luo, 2024) is a strong competitor that also aims to unify feature spaces, specifically by using Singular Value Decomposition (SVD) on the node feature matrix $\mathbf{X}$. Focusing on the input feature space bridging technique, from a linear algebra perspective, the subspace identified by the left singular vectors from the SVD of the data matrix $\mathbf{X}$ (used by SCORE) is mathematically equivalent to the eigenspace derived from the eigendecomposition of the node-covariance (Gram) matrix (the target of ALL-IN). If both methods were deterministic, they would theoretically yield identical feature spaces and suffer from the same limitations.
> >
> > One fundamental advantage of ALL-IN lies not in the matrix operation itself, but in the stochastic nature of its estimation. SCORE’s use of deterministic SVD means that it preserves feature-based symmetries: if two nodes have different features but occupy symmetric positions in the feature space (automorphisms), the deterministic operation will compress them into identical embeddings, causing a loss of fine-grained node identity. In contrast, ALL-IN estimates the covariance structure using random projections ($\mathbf{R}^{(0)} = \mathbf{X}\mathbf{C}$). As proven in Theorem 4.3 (Distinguishability through $\mathbf{C}$), the introduction of the random matrix $\mathbf{C}$ breaks these symmetries with probability 1. This ensures that nodes with distinct input features are mapped to distinct representations, even if they are structurally symmetric. This stochastic symmetry breaking allows ALL-IN to retain a richer, more discriminative signal than the one obtained from SVD, leading to the superior transfer performance observed in our experiments (e.g., 92.90% vs. 85.33% on MUTAG).
> >
> > ---
> >
> > We would like to conclude by expressing our gratitude for your positive assessment of our work, as well as the insightful and thoughtful feedback, which helped us to improve the quality of our paper. Thank you.

---

### Author Response · Authors · 2025-12-03
**Final Author Comments (Part 1)**

Dear Area Chair,

We thank you for overseeing our submission and for your commitment to the reviewing process. In light of the conclusion of the discussion period that terminated earlier than expected, **preventing all of our Reviewers from participating in the discussion**, we would like to provide a concise summary of the reviews, the rebuttal, and the revisions included in the current version of our submission. We hope this is helpful for your decision, and that the full context will be taken into consideration in the final decision making process. Thank you.

We are also grateful to all four reviewers for their detailed comments. During the rebuttal period, we thoroughly and profoundly addressed every point raised by each Reviewer (theoretical, experimental, and presentation/editorial wise) and incorporated all clarifications and extensive new experiments into the revised manuscript. Unfortunately, because of the OpenReview incident, none of our reviewers had the chance to communicate with us, despite our provision of the specific data and comparisons they requested, which directly and fully address all raised comments. Our revised version is uploaded to OpenReview, with changes marked in blue.

Nonetheless, we believe that **our clarifications and revisions address all the raised comments**, and we believe that had they engaged and the planned reviewing timeline and process would have proceeded as expected, our rebuttal and possible follow-up discussions would have resulted in further positive assessment of our paper reflected in the final scores. We hope that this important detail will be taken into account. **Thank you.**

---


### **1. Overall assessment and scores after rebuttal**
The initial ratings and final ratings (unchanged due to early termination of discussion period) are:

**Reviewer PJTg**
- Soundness: 4 (excellent)
- Presentation: 4 (excellent)
- Contribution: 4 (excellent)
- **Rating: 10**

**Reviewer txCX**
- Soundness: 3 (good)
- Presentation: 3 (good)
- Contribution: 3 (good)
- **Rating: 6**

**Reviewer ptNs**
- Soundness: 3 (good)
- Presentation: 3 (good)
- Contribution: 3 (good)
- **Rating: 4**


**Reviewer 8FFw**
- Soundness: 2 (fair)
- Presentation: 2 (fair)
- Contribution: 2 (fair)
- **Rating: 2**


As mentioned earlier, none of the Reviewers had the chance to communicate with us, but we believe that the scores after a discussion period as is planned for ICLR would have resulted in improved scores, given our detailed responses to each of the Reviewers comments.

---

### **2. Strengths highlighted by the Reviewers**

**Conceptual novelty:** Reviewers endorsed the core idea of bridging feature spaces via covariance operators. **PJTg** described the approach as "very clever" and a "novel proposition to build a transferable, general and learnable representation." **txCX** praised the "clear and compelling principle" that is "simple and architecture-agnostic."

**Theoretical Rigor:** Reviewers valued the formal proofs regarding invariance. **PJTg** highlighted the "thorough theoretical analysis" of orthogonal rotation invariance and distinguishability. **ptNs** acknowledged the method is "supported by theoretical proofs demonstrating that its representations are robust."

**Empirical Performance:** There was a strong consensus on the results. **ptNs** noted the "strong empirical results" and successful transfer to unseen datasets "without retraining." **PJTg** described the experiments as "solid" and covering "a variety of state-of-the-art methods."

---

> ### Author Response · Authors · 2025-12-03
> **Final Author Comments (Part 2)**
>
> ### **3. Main concerns and how they were resolved**
>
> **Computational Complexity (ptNs, txCX):** We clarified that ALL-IN scales **linearly** in the number of edges and nodes when implemented with standard sparse adjacency matrices. We refined and updated the complexity analysis in the manuscript (Appendix C.17) to reflect the efficiency of ALL-IN.
>
> **Baselines and State-of-the-Art (ptNs, 8FFw):** We added extensive evaluations of ALL-IN and comparisons with recent Graph Foundation Models (GFMs) and specialized baselines requested by the Reviewers. In particular, we added the following results to the revised paper:
>
> - **GFMs:** Added comparisons to **GFT**, **SAMGPT**, **OpenGraph**, **AutoGFM**, and **RiemannGFM**.
> - **Heterophily:** Added specialized baselines **H2GCN**, **GloGNN++**, **GPR-GNN**, and **LinkX**.
> - **Result:** ALL-IN consistently outperformed these baselines (e.g., surpassing H2GCN by +13.5% on Squirrel).
>
> **Experimental Scope and Dataset Domains (8FFw):** We expanded the evaluation beyond bioinformatics and citation networks. In particular:
>
> - **New Domains:** Added **3D shape segmentation (ShapeNet)** and **social network classification (IMDB-B)**.
> - **New Tasks:** Added graph regression on **Peptides-struct**.
> - **Result:** ALL-IN matched or outperformed GIN and GPS baselines in these new settings.
>
> **Ablations and Edge Features (ptNs, txCX):** We provided concrete ablation studies (Tables 7, 8, 9, 14, 15, 16, 17) confirming the necessity of stochastic projections, edge-covariance operators, and multi-hop propagations. We demonstrated that removing edge features degrades performance, justifying our aggregation strategy.
>
> **Pre-training Objectives (8FFw)** We clarified the multi-task supervised pre-training objective and provided a direct comparison between supervised and unsupervised pre-training (Table 22), showing comparable performance.
> All of these changes and clarifications are incorporated into the current OpenReview version, marked in blue, and we believe they fully resolve the Reviewers' concerns.
>
> ---
>
> ### **4. Effect of the early termination of the discussion period**
>
> Because our Reviewers were not able to respond to our rebuttal, the current scores under-represent what we think, given our rebuttal and revision, the scores would have looked like in a reviewing procedure as planned by the conference. Concretely:
>
> - **ptNs (Score 4)** rated the paper's fundamentals as "Good" and asked for complexity clarification and baseline comparisons. We provided the linear complexity proof and the exact requested baselines. Given their positive assessment of the paper's "Soundness," we expect that these clarifications and added experiments in our revised paper, would have supported a positive score update, had the discussion continued.
> - **8FFw (Score 2)** asked for clarification on pre-training and evaluation on broader domains. We clarified the pre-training objective and added experiments on 3D shapes and social networks. Their concerns regarding "unclear pretraining" and "limited domains" are now factually and profoundly resolved in the revised paper.
>
> Overall, we feel that our detailed responses to each of the Reviewers’ comments, including a revision that incorporates all the discussions and added results fully address any remaining gap, and that the early termination of the discussion period prevented Reviewers from reflecting this in their scores and possible follow-up discussions.
>
> ---
>
> ### **5. Conclusion**
>
> We believe, based on the feedback from all Reviewers, and especially after our rebuttal and paper revision, that our ALL-IN presents a theoretically grounded and empirically robust solution to input feature heterogeneity, towards Graph Foundation Models.
>
> We respectfully ask you, our Area Chair, that your final decision will take into consideration the context of this year’s incident on OpenReview, the overall positive feedback received from Reviewers, and importantly the substantiated and thorough rebuttal and revised submission that addresses all comments raised by our Reviewers.
>
> We appreciate your attention to these important details, and for your commitment to the reviewing process.
>
> Thank you, and warmest regards,\
> Authors

---

### Meta-Review · Area_Chair_rWbR · 2026-01-07

**Summary:**

This paper proposes ALL-IN, which projects node (and optionally edge) features into a shared random space and builds covariance-based operators to derive representations invariant to graph input feature spaces, which can enable zero-shot transfer across graph datasets with mismatched input features.  Reviewers appreciated the simple and architecture-agnostic principle for bridging feature spaces (txCX) and  clever invariance-motivated theoretical analysis (PJTg), along with some evidence of competitive pretraining performance and transfer to unseen datasets (ptNs).

- the “foundation-model” claim would be stronger with broader, larger-scale and more diverse downstream validation beyond small benchmarks and clearer stress-tests of scalability/memory (txCX, ptNs).  it is less clear whether FMs are really needed for settings with many small graphs.  Authors could make this contribution really strong by expanding the scope of this work to better addressing this point and strengthening motivation and practicality.

- the paper could benefit from better positioning against recent GFM baselines and more consistent ablations/clarifications (ptNs, 8FFw)

- pretraining setup and dataset split rationale were initially unclear and appear somewhat ad hoc (8FFw)

- the paper draws limited connection and intuition in the invariances motivating their method and real graph datasets, which is a missed opportunity for the work (esp in highlighting how these invariances may exist across domains) (AC)

Authors took considerable troubles to add new results to the work which should strengthen it during the rebuttal period.  I appreciate these additions and I am sure future readers will too.  I expect reviewers may have raised some scores.  Ultimately this paper had quite broad score range, but given the work is a notable and thought-inspiring contribution for the GFM community, I recommend to accept.

**Reviewer Concerns:**

See above.

**Reviewer Scores:**

PJTg: 10->10
txCX: 6->6
ptNs: 4->5
8FFw: 2->3

I think the authors did address several concerns raised, but one concern is that these answers (perceivably) raise more questions about whether the positioning of this work should be improved (e.g. to encompass proactive discussion re: other GFMs, desirable criteria of GFMs like scale to large-scale data, exposition of the invariances and whether they can be captured on large-scale data with rich input features with low-rank R matrices, etc.). I encourage authors to iterate on these for the final version.

---

### Decision · Program_Chairs · 2026-01-26

Accept (Poster)